# Physics-Constrained Fine-Tuning of Flow-Matching Models for Generation and Inverse Problems

**Jan Tauberschmidt**[1,2], **Sophie Fellenz**[2], **Sebastian J. Vollmer**[1,2], **Andrew B. Duncan**[3]

[1]DSA, German Research Center for Artificial Intelligence (DFKI)
[2]Department of Computer Science, University of Kaiserslautern–Landau (RPTU)
[3]Department of Mathematics, Imperial College London
{jan.tauberschmidt, sebastian.vollmer}@dfki.de,
fellenz@cs.uni-kl.de, a.duncan@imperial.ac.uk

## Abstract

We present a framework for fine-tuning flow-matching generative models to enforce physical constraints and solve inverse problems in scientific systems. Starting from a model trained on low-fidelity or observational data, we apply a differentiable post-training procedure that minimizes weak-form residuals of governing partial differential equations (PDEs), promoting physical consistency and adherence to boundary conditions without distorting the underlying learned distribution. To infer unknown physical inputs, such as source terms, material parameters, or boundary data, we augment the generative process with a learnable latent parameter predictor and propose a joint optimization strategy. The resulting model produces physically valid field solutions alongside plausible estimates of hidden parameters, effectively addressing ill-posed inverse problems in a data-driven yet physics-aware manner. We validate our method on canonical PDE problems, demonstrating improved satisfaction of physical constraints and accurate recovery of latent coefficients. Further, we confirm cross-domain utility through fine-tuning of natural-image models. Our approach bridges generative modeling and scientific inference, opening new avenues for simulation-augmented discovery and data-efficient modeling of physical systems.

## 1 Introduction

Physical systems with rich spatio-temporal structure can be effectively represented by deep generative models, including diffusion and flow-matching methods (Kerrigan et al., 2024; Erichson et al., 2025; Baldan et al., 2025; Price et al., 2023). Although their dynamics can be highly complex, these systems are often governed by fundamental principles, such as conservation laws, symmetries, and boundary conditions, that constrain the space of admissible solutions. Incorporating such physical structure into generative modeling can improve both sample fidelity and out-of-distribution generalization.

In many scientific domains, including atmospheric and oceanographic modeling, seismic inversion, and medical imaging, we often observe system states without access to the underlying physical parameters that govern them. Crucially, PDE-based constraints are typically parameter-dependent, with residuals that vary according to material properties, source terms, or other latent variables. Prior work has largely focused on simple or global constraints—such as fixed boundaries or symmetries, that apply uniformly across the data distribution. Handling parameter-dependent constraints naively would require training over the joint distribution of solutions and parameters, which is often infeasible because parametric labels are missing, expensive to obtain, or high-dimensional. Addressing this limitation is critical for scientific discovery. Many inverse problems in the natural sciences and engineering require reasoning about unobserved parameters or exploring hypothetical scenarios inaccessible to direct experimentation. A generative model that can enforce parameter-dependent PDE constraints using only observational data would provide a powerful tool for data-efficient sim-

ulation, hypothesis testing, and the discovery of new physical phenomena, helping to bridge the gap between raw observations and mechanistic understanding.

This work proposes a framework for fine-tuning flow-matching generative models to enforce parameter-dependent PDE constraints without requiring joint parameter–solution training data. This work aligns with a growing trend of simulation-augmented machine learning (Karniadakis et al., 2021), where generative models accelerate scientific discovery by efficiently exploring physically plausible solution spaces. Our approach reformulates fine-tuning as a stochastic optimal control problem via Adjoint Matching (Domingo-Enrich et al., 2025), guided by weak-form PDE residuals. By augmenting the model with a latent parameter evolution, we enable joint generation of physically consistent solution–parameter pairs, addressing ill-posed inverse problems. We evaluate our proposed fine-tuning framework on four representative PDE families spanning elliptic diffusion, elasticity, wave propagation, and incompressible flow and show an application to natural images. We demonstrate denoising and conditional generation capabilities, including robustness to noisy data and the ability to infer latent parameters from sparse observations. Visual and quantitative results, including strong reductions in residuals across tasks and robustness to model misspecification, highlight the flexibility of our method for integrating physical constraints into generative modeling.

To sum up, our contributions are as follows:

- POST-TRAINING ENFORCEMENT OF PHYSICAL CONSTRAINTS: We introduce a fine-tuning strategy that tilts the generative distribution toward PDE-consistent samples using weak-form residuals, improving physical validity while preserving diversity.

- ADJOINT-MATCHING FINE-TUNING WITH THEORETICAL GROUNDING: Leveraging the adjoint-matching framework, we recast reward-based fine-tuning as a stochastic control problem, extending flow-matching models to generate latent parameters alongside states, enabling inverse problem inference without paired training data.

- BRIDGING GENERATIVE MODELING AND PHYSICS-INFORMED LEARNING: Our approach connects preference-aligned generation with physics-based inference, enabling simulation-augmented models to generate solutions that respect complex physical laws.

An implementation of our method is available at `https://github.com/jantauberschmidt/PCFT`.

## 2 RELATED WORK

**Physics-Constrained Generative Models** Integrating physical constraints—such as boundary conditions, symmetry invariances, and partial differential equation (PDE) constraints—into machine learning models improves both accuracy and out-of-distribution generalization. Classical approaches, such as Physics-Informed Neural Networks (PINNs, Raissi et al. (2019)), directly regress solutions that satisfy governing equations. While effective for forward or inverse problems, PINNs do not capture distributions over solutions, making them unsuitable for generative tasks that require sampling diverse plausible outcomes.

In the generative setting, the main challenge is ensuring that the physically constrained samples retain the variability of the underlying generative model, avoiding pathological issues such as mode collapse. Bastek et al. (2024) proposes a unified framework for introducing physical constraints into Denoising Diffusion Probabilistic Models (DDPMs, Ho et al. (2020)) at pre-training time, by adding a first-principles physics-residual loss to the diffusion training objective. This loss penalizes violations of governing PDEs (e.g. fluid dynamics equations) so that generated samples inherently satisfy physical laws. The method was empirically shown to reduce residual errors for individual samples significantly, while simultaneously acting as a regularizer against overfitting, thereby improving generalization. To evaluate the physics-residual loss, one needs to compute the expected PDE residual of the final denoised sample conditioned on the current noisy state in the DDPM process. Accurately estimating this expectation requires generating multiple reverse-diffusion trajectories from the same noisy sample, which makes pre-training significantly more expensive. A common alternative is to use Tweedie's formula to approximate the conditional expectation in a single pass, but this shortcut introduces bias, particularly in the final denoising steps.

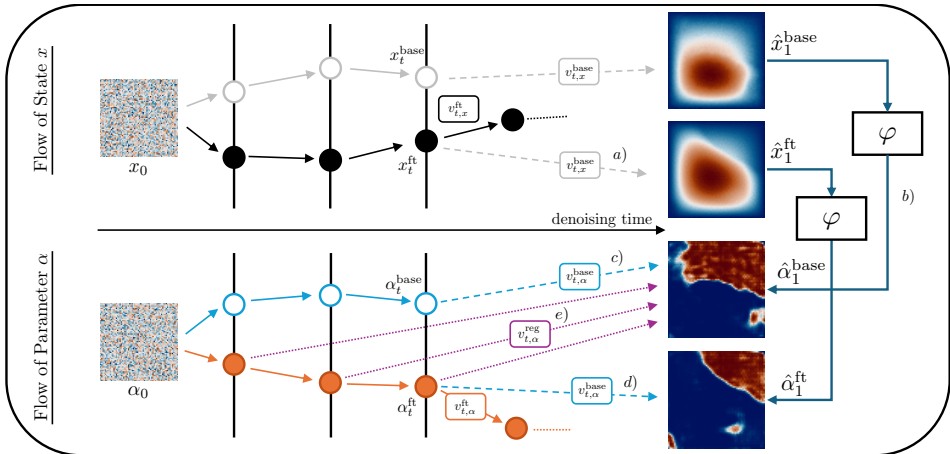

Figure 1: Visual depiction of proposed method. Starting at state $x_t^{\text{base}}$ or $x_t^{\text{ft}}$, we use the base vector field $v_{t,x}^{\text{base}}$ to predict the final sample [a]. Through the inverse predictor $\varphi$, we recover the corresponding predicted parameters $\hat{\alpha}_1^{\text{base}}$ and $\hat{\alpha}_1^{\text{ft}}$ [b]. These estimates can be used as a target for evolving $\alpha_t^{\text{base}}$ [c]) or as a baseline for the fine-tuned evolution of $\alpha_t^{\text{ft}}$ [d]. For purposes of regularization, we further consider $v_{t,\alpha}^{\text{reg}}$, pointing from the current $\alpha_t^{\text{ft}}$ to the predicted final parameter of the base evolution $\hat{\alpha}_1^{\text{base}}$ [e]).

Zhang & Zou (2025) proposes enforcing constraints through a post-hoc distillation stage, where a deterministic student model is trained from a vanilla diffusion model to generate samples in one-step, regularized by a PDE residual loss. In Wang et al. (2025) the authors introduce PhyDA, diffusion-based data assimilation framework that ensures reconstructions obey PDE-based dynamics, specifically for atmospheric science. An autoencoder is used to encode sparse observations into a structured latent prior for the diffusion model, which is trained with an additional physical residual loss.

**Inference- and Post-Training Constraint Enforcement** Various works have proposed approaches to enforce PDE constraints at inference time, often in combination with observational constraints, drawing connections to conditional diffusion models (Dhariwal & Nichol, 2021; Ho & Salimans, 2021). Huang et al. (2024) introduce guidance terms within the denoising update of a score-based diffusion model to steer the denoising process towards solutions which are both consistent with data and underlying PDEs. A related approach was considered by Xu et al. (2025), further introducing an adaptive constraint to mitigate instabilities in early diffusion steps. In Christopher et al. (2024), the authors recast the inference-time sampling of a diffusion process as a constrained optimization problem, each diffusion step is projected to satisfy user-defined constraints or physical principles. This allows strict enforcement of hard constraints (including convex and non-convex constraints, as well as ODE-based physical laws) on the generated data. Lu & Xu (2024) consider the setting where the base diffusion model is trained on cheap, low-fidelity simulations, leveraging a similar approach to generate down-scaled samples via projection.

**Flow-Matching Models for Simulation and Inverse Problems** Flow-matching (FM, Lipman et al. (2023)) has emerged as a flexible generative modeling paradigm for complex physical systems across science, including molecular systems (Hassan et al., 2024), weather (Price et al., 2023) and geology (Zhang et al., 2025) . In the context of physics-constrained generative models Utkarsh et al. (2025) introduces a zero-shot inference framework to enforce hard physical constraints in pre-trained flow models, by repeatedly projecting the generative flow at sampling time. Similarly, Cheng et al. (2024) proposed the ECI algorithm, to adapt a pre-trained flow-matching model so that it exactly satisfies constraints without using analytical gradients. In each iteration of flow sampling, ECI performs: an Extrapolation step (advancing along the learned flow), a Correction step (applying a constraint-enforcement operation), and an Interpolation step (adjusting back towards the model's trajectory). While projection approaches are a compelling strategy for hard constraints, they can be

challenging particularly for local constraints such as boundary conditions, as direct enforcement can introduce discontinuities. The above approach mitigates this by interleaving projections with flow steps, however this relies on the flow's ability to rapidly correct such non-physical artifacts.

Baldan et al. (2025) propose Physics-Based Flow Matching (PBFM), which embeds constraints (PDE or symmetries) directly into the FM loss during training. The approach leverages temporal unrolling to refine noise-free final state predictions and jointly minimizes generative and physics-based losses without manual hyperparameter tuning of their tradeoff. To mitigate conflicts between physical constraints and the data loss, they employ the ConFIG (Liu et al., 2024), which combines the gradients of both losses in a way that ensures that gradient updates always minimize both losses simultaneously.

Related to our approach are the works on generative models for Bayesian inverse problems (Stuart, 2010), where the goal is to infer distributions over latent PDE parameters given partial or noisy observations. Conditional diffusion and flow-matching models can be used to generate samples from conditional distributions and posterior distributions, supporting amortized inference and uncertainty quantification (Song et al., 2021; Utkarsh et al., 2025; Zhang et al., 2023). Conditioning is typically achieved either through explicit parameter inputs or guidance mechanisms during sampling, as in classifier-guided diffusion. While effective when large volumes of paired training data is available, these approaches are less relevant to observational settings where parameters are unobserved. In contrast, our approach connects the observed data to the latent parameters only during post-training, requiring substantially smaller volumes of data.

## 3   METHOD

FM models are trained to learn and sample from a given distribution of data $p_{\text{data}}$. They approximate this distribution by constructing a Markovian transformation from noise to data, such that the time marginals of this transformation match those of a *reference flow* $X_t = \beta_t X_1 + \gamma_t X_0$. Specifically FM models learn a vector field $v_t(x)$ that transports noise to data, via the ODE $dX_t = v_t(X_t)\,dt$. We can optionally inject a noise schedule $\sigma(t)$ along the trajectory to define an equivalent SDE that preserves the same time marginals (Maoutsa et al., 2020),

$$dX_t = \left( v_t(X_t) + \frac{\sigma(t)^2}{2\eta_t}\left( v_t(X_t) - \frac{\dot{\beta}_t}{\beta_t}X_t \right) \right) dt + \sigma(t)\,dB_t \; =: \; b_t(X_t)\,dt + \sigma(t)\,dB_t, \quad (1)$$

where we combine coefficients $\beta_t$ and $\gamma_t$ into $\eta_t = \gamma_t\big(\frac{\dot{\beta}_t}{\beta}\gamma_t - \dot{\gamma}_t\big)$.

Assuming we have access to a FM model which generates samples according to distribution $p(x)$, we seek to adjust this model so as to generate samples from the tilted distribution $p_r(x) \propto e^{\lambda\,r(x)}p(x)$, where $r$ is a reward function and $\lambda$ characterizes the degree of distribution shift induced by fine-tuning. To achieve this, we leverage the adjoint-matching framework of Domingo-Enrich et al. (2025). This work reformulates reward fine-tuning for flow-based generative models as a control problem in which the base generative process given by $v_t^{\text{base}}$ is steered toward high-reward samples via modifying the learned vector field, which we denote as $v_t^{\text{ft}}$ with corresponding drift term $b_t^{\text{ft}}$. Our approach is conceptually related to reward- or preference-based fine-tuning of generative models (Christiano et al., 2017; Sun et al., 2024), where a learned or computed reward steers generation toward desired properties. Here, the reward is defined via PDE residuals, encoding knowledge about underlying dynamics and physical constraints to the solutions space as deviations to differential operators or boundary conditions. Notably, we assume that the distribution generated by the base model $p(x)$ only captures an observed quantity, but does not provide us with corresponding parameters or coefficient fields often needed to evaluate the respective differential operator. In the following, we will present a strategy of jointly recovering unknown parameters and fine-tuning the generation process.

### 3.1   REWARD

A generative model can reproduce the visual characteristics of empirical data while ignoring the physics that governs it, thereby rendering the samples unusable for downstream scientific tasks. To bridge this gap we impose the known governing equations as *soft constraints*, expressed through

differential operators $\mathcal{L}_\alpha x = 0$ with parameters $\alpha$. Throughout, a generated sample $x$ is interpreted as a discretization of a continuous field $x(\xi)$ on a domain $\Omega$. The *strong* PDE residual is defined as

$$\mathcal{R}_{\text{strong}}(x, \alpha) = \left\| \mathcal{L}_\alpha x \right\|_{L^2(\Omega)}^2.$$

In practice, strong residuals involve high-order derivatives that make the optimization landscape unstable. We therefore adopt *weak-form residuals* of the form $\langle \mathcal{L}_\alpha x, \psi \rangle_{L^2(\Omega)}$ for suitably chosen test functions $\psi \in \Psi$, which are numerically more stable under noisy or misspecified data. Repeated applications of integration-by-parts can transfer derivatives from $x$ to $\psi$. The set $\Psi$ is composed of compactly supported local polynomial kernels. For each evaluation we draw $N_{\text{test}}$ such functions; their centers and length-scales are sampled at random. A mollifier enforces $\psi|_{\partial\Omega} = 0$, justifying the integration by parts. The resulting residual is

$$\mathcal{R}_{\text{weak}}(x, \alpha) = \frac{1}{N_{\text{test}}} \sum_{i=1}^{N_{\text{test}}} \left| \langle \mathcal{L}_\alpha x, \psi^{(i)} \rangle_{L^2(\Omega)} \right|^2.$$

These randomly sampled local test functions act as stochastic probes of PDE violations, providing a low-variance, data-efficient learning signal. A more detailed description of the test functions used can be found in Appendix D.3. Note that the residual might be augmented by adding soft constraints for boundary conditions.

## 3.2 JOINT EVOLUTION

Fine-tuning is nontrivial in our setting because we must infer latent physical parameters jointly with the generated solutions. On fully denoised samples, we can train an inverse predictor, i.e., $\varphi(x_1) = \alpha_1$, such that the weak PDE residual is minimized. As a naïve approach, this already induces a joint distribution over $(x_1, \alpha_1)$ via the push-forward through $\varphi$. However, we advocate a more principled formulation that evolves *both* $x$ and $\alpha$ along vector fields, enabling joint sampling of parameters and solutions, as well as a controlled regularization of fine-tuning through the Adjoint Matching framework as outlined below. In the fine-tuning model, this can be achieved by directly learning the vector field $v_{t,\alpha}^{\text{ft}}$ jointly with $v_{t,x}^{\text{ft}}$ by augmenting the neural architecture. Since no ground-truth flow of $\alpha$ for the base model is available, at each state $(x_t, \alpha_t)$ we define a *surrogate base flow* using the inverse predictor $\varphi$. Specifically, we consider the one-step estimates

$$\hat{x}_1 = x_t + (1-t) v_t^{\text{base}}(x_t), \qquad \hat{\alpha}_1 = \varphi(\hat{x}_1).$$

The direction from the current state $\alpha_t$ to the predicted final parameter $\hat{\alpha}_1$ serves as a base vector field which we use to evolve $\alpha$, i.e. $v_{t,\alpha}^{\text{base}}(\alpha_t) = (\hat{\alpha}_1 - \alpha_t)/(1-t)$ inducing corresponding drift $b_{t,\alpha}^{\text{base}}$. This *surrogate base flow*, starting at a noise sample $\alpha_0^{\text{base}} \sim \mathcal{N}(0, I)$, emulates a denoising process of the recovered parameter. We denote by $\alpha^{\text{base}}$ the parameter aligned with the base trajectory $x^{\text{base}}$. While the evolution of $\alpha^{\text{base}}$ does not influence the trajectory of $x^{\text{base}}$, the inferred vector field can be used to effectively regularize the generation of the fine-tuned model. Similarly, to regularize towards the parameter recovered under the base model, we introduce an additional field $v_{t,\alpha}^{\text{reg}}(\alpha_t^{\text{ft}}) = (\hat{\alpha}_1^{\text{base}} - \alpha_t^{\text{ft}})/(1-t)$. This vector field points from the current parameter estimate of the fine-tuned trajectory $\alpha_t^{\text{ft}}$ to the recovered parameter under the base model $\hat{\alpha}_1^{\text{base}}$. The field is used to pull the fine-tuned dynamics towards final samples associated with parameters similar to those of the base trajectory. The introduced vector fields are visualized in Fig. 1.

## 3.3 ADJOINT MATCHING

Considering an augmented state variable of the joint evolution $\tilde{X}_t = (X_t^T, \alpha_t^T)^T$, we cast fine-tuning as a stochastic optimal control problem:

$$\min_{\tilde{u}} \mathbb{E}\left[ \int_0^1 \left( \frac{1}{2} \left\| \tilde{u}_t(\tilde{X}_t) \right\|^2 + f(\tilde{X}_t) \right) dt + g(\tilde{X}_1) \right]$$

$$\text{s.t.} \quad d\tilde{X}_t = \left( \tilde{b}_t^{\text{base}}(\tilde{X}_t) + \sigma(t)\, \tilde{u}_t(\tilde{X}_t) \right) dt + \sigma(t)\, d\tilde{B}_t \tag{2}$$

with control $\tilde{u}_t(\tilde{X}_t)$, running state cost $f(\tilde{X}_t)$, and terminal cost $g(\tilde{X}_1)$. In this formulation, fine-tuning amounts to a point-wise modification of the base drift through application of control $\tilde{u}$, i.e.

$$\tilde{b}_t^{\text{ft}}(\tilde{X}_t) = \tilde{b}_t^{\text{base}}(\tilde{X}_t) + \sigma(t)\, \tilde{u}_t(\tilde{X}_t).$$

In Domingo-Enrich et al. (2025), Adjoint Matching is introduced as a technique with lower variance and computational cost than standard adjoint methods. The method is based on a *Lean Adjoint* state, which is initialized as

$$\tilde{a}_1^T = \tilde{\lambda} \nabla_{\tilde{x}} g(\tilde{X}_1) = \left( \lambda_x \nabla_x g(X_1, \alpha_1), \, \lambda_\alpha \nabla_\alpha g(X_1, \alpha_1) \right)$$

and evolves backward in time according to

$$\frac{d}{dt} \tilde{a}_t = - \left( \nabla_{\tilde{x}} \tilde{b}_t^{\text{base}}(\tilde{X}_t)^T \, \tilde{a}_t + \nabla_{\tilde{x}} f(\tilde{X}_t)^T \right) = - \begin{pmatrix} J_{xx}^T & J_{\alpha x}^T \\ J_{x\alpha}^T & J_{\alpha\alpha}^T \end{pmatrix} \begin{pmatrix} a_{t,x} \\ a_{t,\alpha} \end{pmatrix} - \begin{pmatrix} \nabla_x f(X_t, \alpha_t)^T \\ \nabla_\alpha f(X_t, \alpha_t)^T \end{pmatrix} \quad (3)$$

where the block-Jacobian is evaluated along the base drift for $X$ and $\alpha$, which means that $J_{ij} = \nabla_j b_{t,i}^{\text{base}}(X_t, \alpha_t)$ for $i, j \in \{x, \alpha\}$. The hyperparameters $\lambda_x$ and $\lambda_\alpha$ can be used to regulate the extent to which the fine-tuned distribution departs from the base distribution. The *Adjoint Matching* objective can then be formulated as a consistency loss:

$$\begin{aligned}
\mathcal{L}(\tilde{u}; \tilde{X}) &= \tfrac{1}{2} \int_0^1 \left\| \tilde{u}_t(\tilde{X}_t) + \sigma(t) \, \tilde{a}_t \right\|^2 dt \\
&= \tfrac{1}{2} \int_0^1 \left( \left\| u_{t,x}(X_t, \alpha_t) + \sigma(t) \, a_{t,x} \right\|^2 + \left\| u_{t,\alpha}(X_t, \alpha_t) + \sigma(t) \, a_{t,\alpha} \right\|^2 \right) dt.
\end{aligned} \quad (4)$$

It can be shown (Domingo-Enrich et al., 2025) that with $f = 0$, this objective is consistent with the tilted target distribution for reward $r = -g$, if optimized with a *memoryless* noise schedule. This schedule ensures sufficient mixing during generation such that the final sample $X_1$ is independent of $X_0$. To stabilize fine-tuning we introduce a scaled variant of the memoryless noise schedule. Instead of using the canonical choice $\sigma^2(t) = 2\eta_t$ identified by Domingo-Enrich et al. (2025), we adopt

$$\sigma^2(t) = (1 - \kappa) \, 2\eta_t, \qquad 0 \le \kappa < 1,$$

which retains the theoretical memoryless property (see Lemma 1 in Appendix D.4) while attenuating the magnitude of the noise variance. The introduction of the scaling factor $0 \le \kappa < 1$ constitutes a simple but novel extension of the adjoint-matching framework. Whereas prior work highlighted a unique schedule, our analysis shows that a family of scaled schedules remains consistent with the memoryless condition. This additional degree of freedom acts as a *numerical stabilisation knob*, mitigating blow-ups near $t \to 0$ without losing theoretical consistency. Further, it offers a *control–fidelity trade-off* by regulating the amount of exploration. In practice, this flexibility allows practitioners to adapt fine-tuning to the conditioning of the PDE residuals and the stability of the solver, a feature not available in the original formulation.

Equation 2 is optimized by iteratively sampling trajectories with the fine-tuned model while following a memoryless noise schedule, numerically computing the lean adjoint states by solving the ODE in Equation 3, and taking a gradient descent step to minimize the loss in Equation 4. Note that gradients are only computed through the control $\tilde{u}_t$ and not through the adjoint, reducing the optimization target to a simple regression loss. We state the full training algorithm and implementation details in Appendix D.5.

Adjoint Matching steers the generator toward the reward-tilted distribution, thereby reshaping the entire output distribution rather than correcting individual trajectories. However, when fine-tuning observational data or under system misspecification, we might be interested in retaining sample-specific detail. Empirically we find that this can be effectively encoded by imposing similarity of the inferred coefficients between base and fine-tuned model. Therefore, we add a running state cost

$$f(\alpha) = \lambda_f \left\| v_{t,\,\alpha}^{\text{ft}}(\alpha) - v_{t,\,\alpha}^{\text{reg}}(\alpha) \right\|^2$$

which penalizes deviations of the fine-tuned $\alpha$-drift from the direction pointing toward the base estimate $\hat{\alpha}_1^{\text{base}}$. The hyper-parameter $\lambda_f$ controls a smooth trade-off: $\lambda_f = 0$ recovers pure Adjoint Matching, while larger $\lambda_f$ progressively anchors the final parameters $\alpha_1$ obtained under the fine-tuned model to their base-model counterparts, thus retaining trajectory-level detail.

## 4 EXPERIMENTS

We evaluate across five settings: four PDE systems (including boundary and system misspecification, and observational noise) and a natural-image model. Unlike latent-space fine-tuning for

images, our PDE models operate directly in pixel space. High-variance noise during sampling can drive off-manifold trajectories and perturb PDE residuals, motivating $\kappa > 0$ for these models. For base Flow Matching backbones, we use U-FNO (Wen et al., 2022) for PDEs and the DiT-based latent FM of Dao et al. (2023) for images. In all experiments we first sample from the base generator and pre-train the inverse predictor $\varphi$ to recover $\alpha$ by minimizing the (PDE) residual, then fine-tune. Following Domingo-Enrich et al. (2025), fine-tuning is initialized from the base weights. We augment capacity to condition $v_{t,x}^{\text{ft}}$ on $\alpha_t$ and add a separate head for $v_{t,\alpha}^{\text{ft}}$. Fine-tuning uses a memoryless noise schedule, while all reported results are generated without injected noise ($\sigma(t) = 0$). Implementation details appear in App. D.2.

**Comparisons, ablations, and metrics.** Our proposed method converts a single-variable flow into a joint generative model (Sec. 3.2). We compare against: (i) a *Base AM* variant (vanilla Adjoint Matching) where $\varphi$ is frozen and used only to compute residuals, (ii) a *Base AM+$\varphi$* variant where $\varphi$ continues to train but the flow over $\alpha$ is not modeled jointly, and (iii) *PBFM* (Baldan et al., 2025), augmented with our pre-trained $\varphi$ to enable residual evaluation. Details on the comparison methods can be found in App. E.2. All evaluations use 256 samples, generated from shared seeds across methods. We report weak and strong residuals, $R_{\text{weak}}$ and $R_{\text{strong}}$, scaled by the mean residual of a fixed reference set. The reference set $\mathcal{D}_{\text{ref}}$ is a synthetic, clean dataset generated under the target PDE specification assumed during fine-tuning (no noise, modified BCs, lossless Helmholtz, or unforced Stokes respectively). We also report Maximum Mean Discrepancy (MMD) based distributional similarities for states and parameters ($\text{MMD}_x$, $\text{MMD}_\alpha$) computed against this dataset (details in App. E.1). While the main text shows representative results, the complete set of experimental evaluations is provided in App. F.

## 4.1 DARCY FLOW

Consider a square domain $\Omega = [0, 1]^2$ where a permeability $\alpha(\xi)$ and forcing $f(\xi)$ induce a pressure field $x(\xi)$ governed by $-\nabla \cdot (\alpha(\xi)\nabla x(\xi)) - f(\xi) = 0$ with zero Dirichlet boundary conditions and constant $f$. We draw $\alpha$ from a discretized Gaussian process and corrupt pressures with observation noise before training the base FM. Dataset details are in App. B. Figure 2 compares three Darcy samples generated from the *same* noise seed $x_0$: the base draw, fine-tuning with our regularization (here $\lambda_f = 1.0$), and fine-tuning without regularization. The base pressure $x^{\text{base}}$ is visibly contaminated by high-frequency noise, and the inverse predictor $\varphi$ correspondingly yields a scattered, artifact-ridden permeability map $\alpha^{\text{base}}$. With regularization enabled, fine-tuning attenuates noise in the pressure $x^{\text{ft}}$ while remaining close to $\alpha^{\text{base}}$. Because $\alpha^{\text{base}}$ is itself fragmented, some artifacts persist. In contrast, disabling regularization produces a fully denoised pressure and a markedly more coherent $\alpha^{\text{ft}}$, but at the expected expense of erasing sample-specific details present in the base realization.

| Base | Fine-tuned w/ regularization | Fine-tuned w/o regularization |
|---|---|---|

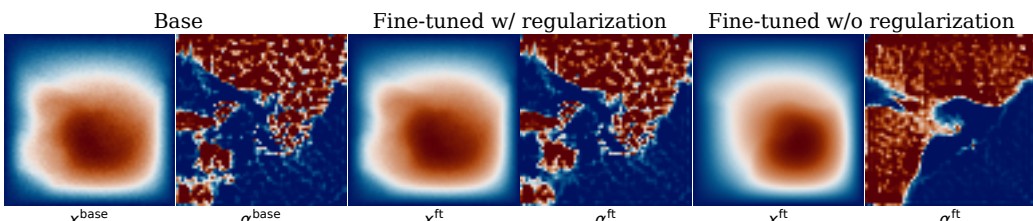

$x^{\text{base}}$ $\quad$ $\alpha^{\text{base}}$ $\quad\quad$ $x^{\text{ft}}$ $\quad$ $\alpha^{\text{ft}}$ $\quad\quad$ $x^{\text{ft}}$ $\quad$ $\alpha^{\text{ft}}$

Figure 2: Darcy denoising (qualitative). Base vs. fine-tuned outputs for a fixed seed. Regularization ($\lambda_f = 1.0$) denoises while staying close to the base sample. Removing it denoises more aggressively at the cost of fidelity to the base realization. Additional non-curated samples in App. F.3.1. Color maps throughout this work taken from Crameri et al. (2020).

We quantify the controllable trade-offs in Fig. 3. Panel (a) increases $\lambda_x = \lambda_\alpha$ at $\lambda_f = 0$, which reduces the PDE residual while also reducing diversity in the inferred permeabilities (measured via the complement of the mean pairwise SSIM; see App. E.1). Panel (b) fixes $\lambda_x = \lambda_\alpha = 20K$ and varies $\lambda_f$, reporting $\text{MMD}_x$ between the fine-tuned samples and the base dataset. As expected, stronger regularization preserves distributional fidelity (lower MMD) but yields higher residuals.

These ablations illustrate how practitioners can target residual reduction or distributional fidelity by tuning $(\lambda_x, \lambda_\alpha, \lambda_f)$.

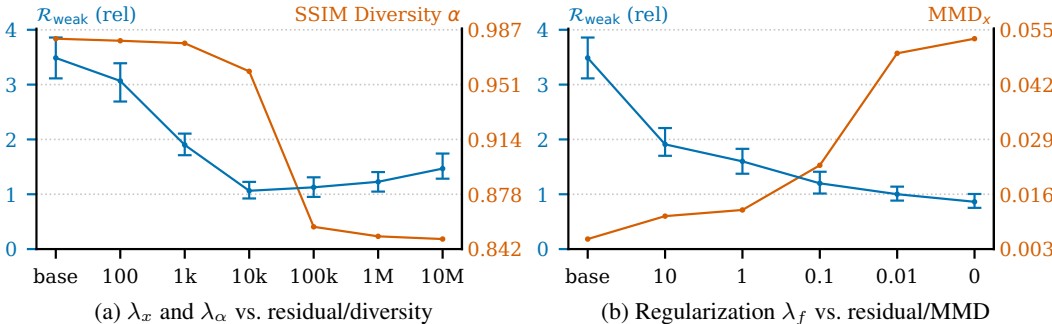

(a) $\lambda_x$ and $\lambda_\alpha$ vs. residual/diversity

(b) Regularization $\lambda_f$ vs. residual/MMD

Figure 3: Darcy ablations. (a) Increasing $\lambda_x = \lambda_\alpha$ with $\lambda_f = 0$ lowers the PDE residual but reduces diversity in the inferred parameters (reported via SSIM-based diversity). (b) Sweeping $\lambda_f$ trades PDE residuals against fidelity to the base distribution ($\text{MMD}_x$). Each point averages 256 samples with shared noise seeds across settings.

Computationally, adaptation is lightweight: fine-tuning on noisy Darcy requires only 20 gradient steps (hyperparameters in App. E.3) and completes in under 15 minutes on a single NVIDIA L40S, after which sampling proceeds at base-model cost with no inference-time adjustments.

## 4.2 GUIDANCE ON SPARSE OBSERVATIONS

In many realistic settings dense observations of a state variable are available for pre-training a generative model, whereas only a few measurements of the latent parameter can be collected. To sample from the posterior of parameter–state pairs that respect such sparse evidence we steer the generative process through *guidance*. Huang et al. (2024) demonstrate guided sampling towards sparse observations from a model that was pre-trained on the joint parameter-state distribution. Our approach applies a similar guidance mechanism, however, to a model that was pre-trained on noisy state observations alone. We state details on the guiding mechanism in E.4. Figure 4 shows that the guided sampler adheres to sparse measurements while preserving realistic variability in the generated samples. Additional results for different amounts of conditioning observations are given in Appendix F.3.5.

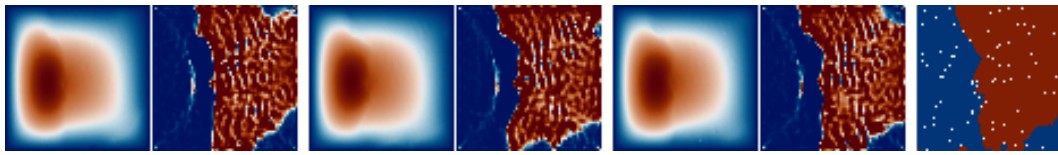

Figure 4: Three samples through guidance towards sparse observations (white markers in right panel) of the permeability, showing a plausible conditional distribution.

## 4.3 LINEAR ELASTICITY

We consider plane–strain linear elasticity on $\Omega = [0,1]^2$ with spatially varying Young's modulus $\alpha(\xi)$ and fixed Poisson ratio. Boundaries are Dirichlet: left/right clamped, top/bottom receive inward sinusoidal *normal* displacements with zero tangential slip. During fine-tuning, we impose a modified lower-boundary amplitude to induce controlled misspecification (see App. B) and include an MSE boundary penalty in the weak residual. We report quantitative BC results as the MSE at the boundary in Table 4.3 and a qualitative comparison is provided in App. F. Our method attains low weak/strong residuals while keeping distributional shift modest; PBFM and FM+ECI drift distributionally or present high residuals (full details and non-curated samples in App. E.5, F.3.2).

| Model | BC error (MSE) ↓ | $\mathcal{R}_{\text{weak}}$ (rel) ↓ | $\mathcal{R}_{\text{strong}}$ (rel) ↓ | $\text{MMD}_x$ ↓ | $\text{MMD}_\alpha$ ↓ |
|---|---|---|---|---|---|
| FM | $6.98 \times 10^{-5}$ ($\pm 0.53$) | $1.59 \times 10^1$ ($\pm 0.37$) | $1.83 \times 10^1$ ($\pm 0.66$) | 0.24 | 0.05 |
| PBFM | $2.32 \times 10^{-5}$ ($\pm 0.87$) | $6.32 \times 10^0$ ($\pm 0.82$) | $4.22 \times 10^0$ ($\pm 0.26$) | 0.92 | 0.54 |
| FM+ECI | 0.0 | $1.01 \times 10^3$ ($\pm 0.13$) | $2.49 \times 10^2$ ($\pm 0.32$) | 1.16 | 0.36 |
| Ours | $1.71 \times 10^{-6}$ ($\pm 0.50$) | $6.15 \times 10^0$ ($\pm 0.77$) | $3.79 \times 10^0$ ($\pm 0.87$) | 0.15 | 0.12 |

Table 1: Linear elasticity under BC misspecification. Quantitative boundary-condition (BC) error, relative weak/strong residuals, and distributional metrics. Our method achieves low residuals with limited distributional shift.

## 4.4 HELMHOLTZ

We consider time–harmonic wave propagation governed by the heterogeneous Helmholtz equation $-\Delta u - (1 - i \tan \delta) \kappa(x)^2 u = s$ on $\Omega = [0,1]^2$ with Robin boundary conditions. Training data use a small damping term ($\tan \delta > 0$) producing complex attenuated fields, while fine-tuning assumes the idealized lossless model ($\tan \delta = 0$), inducing a controlled model mismatch.

| Model | Criterion | $\mathcal{R}_{\text{weak}}$ (rel) ↓ | $\mathcal{R}_{\text{strong}}$ (rel) ↓ | $\text{MMD}_x$ ↓ | $\text{MMD}_\alpha$ ↓ |
|---|---|---|---|---|---|
| FM | – | $1.5 \times 10^1$ ($\pm 0.59$) | $2.55 \times 10^1$ ($\pm 0.55$) | 0.18 | 0.03 |
| PBFM | – | $8.33 \times 10^0$ ($\pm 3.04$) | $1.22 \times 10^1$ ($\pm 0.33$) | 0.09 | 0.03 |
| Base AM | $\mathcal{R}_{\text{weak}}$ | $4.9 \times 10^0$ ($\pm 1.85$) | $1.34 \times 10^1$ ($\pm 0.32$) | 0.15 | 0.04 |
| Base AM | $\text{MMD}_x$ | $5.64 \times 10^0$ ($\pm 2.09$) | $1.59 \times 10^1$ ($\pm 0.33$) | 0.13 | 0.04 |
| Base AM + $\varphi$ | $\mathcal{R}_{\text{weak}}$ | $4.99 \times 10^0$ ($\pm 2.12$) | $1.16 \times 10^1$ ($\pm 0.33$) | 0.13 | 0.05 |
| Base AM + $\varphi$ | $\text{MMD}_x$ | $5.46 \times 10^0$ ($\pm 1.94$) | $1.59 \times 10^1$ ($\pm 0.33$) | 0.12 | 0.04 |
| AM | $\mathcal{R}_{\text{weak}}$ | $4.3 \times 10^0$ ($\pm 1.29$) | $1.14 \times 10^1$ ($\pm 0.29$) | 0.07 | 0.04 |
| AM | $\text{MMD}_x$ | $4.32 \times 10^0$ ($\pm 1.43$) | $1.05 \times 10^1$ ($\pm 0.30$) | 0.06 | 0.04 |

Table 2: Helmholtz: residuals and distribution (representative configs). Normalized weak/strong residuals and MMD metrics. We include AM variants and a PBFM-style baseline.

Table 2 reports representative configurations for each method, selected as either the setting with the lowest weak residual ($\mathcal{R}_{\text{weak}}$) or the lowest $\text{MMD}_x$. Full results are provided in App. F. The base FM model shows the largest weak and strong residuals due to the damped–vs.–lossless mismatch. PBFM substantially reduces both residuals relative to FM and, notably, also lowers $\text{MMD}_x$ and preserves $\text{MMD}_\alpha$. The AM ablations (Base AM and Base AM+$\varphi$) further reduce the weak residuals into the range $4.9 \times 10^0$–$5.6 \times 10^0$, with strong residuals similar to those of PBFM, but they incur a moderate increase in $\text{MMD}_x$ and $\text{MMD}_\alpha$ compared to PBFM. Our full joint AM model achieves the lowest residuals overall (weak residuals down to $4.3 \times 10^0$ and strong residuals near $1.05 \times 10^1$) while simultaneously attaining the lowest $\text{MMD}_x$ among all methods and maintaining $\text{MMD}_\alpha$ comparable to the ablations. This indicates that the joint flow most effectively resolves the misspecification while preserving distributional fidelity.

## 4.5 STOKES LID-DRIVEN CAVITY

We consider steady incompressible flow in the Stokes regime (linear, low–Reynolds-number proxy) governed by $-\nabla \cdot (\nu(x)\nabla u) + \nabla p = f$, $\nabla \cdot u = 0$ with no-slip walls and a smooth moving lid. The dataset uses nonzero Kolmogorov forcing $f \neq 0$, while fine-tuning assumes $f \equiv 0$, creating a systematic model mismatch. Figure 5 reports the residual–distribution trade-offs for the Stokes lid-driven cavity. We show only the Base AM variants and our joint model: the base FM model exhibits extremely large residuals ($3.05 \times 10^2 \pm 3.16$) and is omitted for clarity, while PBFM fails to converge to meaningful velocity–pressure fields (strong residuals $1.15 \times 10^1 \pm 0.05$; see App. F). In contrast to Darcy and Helmholtz, the attainable weak residuals of all remaining variants are similar ($R_{\text{weak}} \approx 4$–15). However, the joint model reaches *substantially lower* parameter-distribution discrepancies, achieving $\text{MMD}_\alpha \approx 0.07$–0.13, whereas both ablations remain around 0.22–0.28. Overall, although residual levels are similar across AM variants, only the joint model can enter the low–MMD regime—particularly for $\text{MMD}_\alpha$. This highlights the joint flow's greater flexibility in achieving high-fidelity parameter distributions.

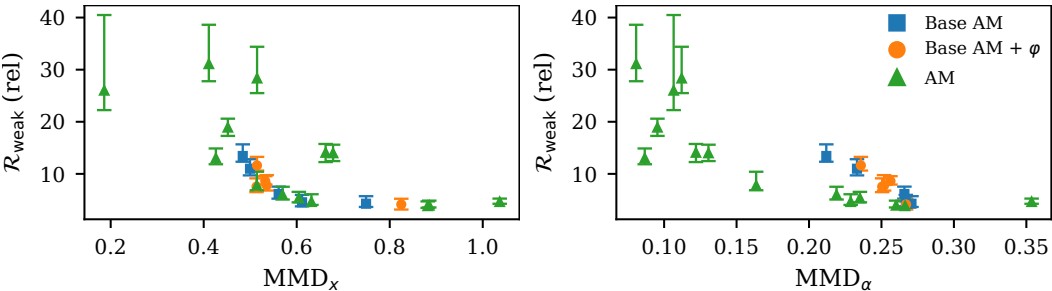

Figure 5: Stokes lid-driven cavity: residual–distribution trade-offs. Weak residuals $R_{\text{weak}}$ versus (a) $\text{MMD}_x$ and (b) $\text{MMD}_\alpha$. Across all variants, attainable residuals are comparable, but the joint model (green) reaches *much lower* parameter-distribution discrepancies ($\text{MMD}_\alpha \approx 0.07$–$0.13$) than the Base AM (blue) and Base AM+$\varphi$ (orange) ablations, which remain around $0.22$–$0.28$.

## 4.6 NATURAL IMAGES: PARAMETRIC COLOR TRANSFORMATION

To demonstrate cross-domain utility, we apply our method to natural images by introducing a parametric recoloring pathway: analogous to the hidden PDE parameter, $\alpha$ here specifies a polynomial color transform that operates outside the latent space, enabling exploration of image appearances not well supported by the base distribution. We use a class-conditional Latent Flow Matching (LFM) model (Dao et al., 2023) pre-trained on ImageNet-1k (Deng et al., 2009) and optimize PickScore (Kirstain et al., 2023) with a globally fixed prompt. As a concrete example, we fine-tune on the class *macaw* with the prompt "close-up Pop Art of a macaw parrot," yielding the samples in Fig. 6. Joint fine-tuning with recoloring produces markedly more vibrant palettes and, crucially, *joint* adjustments (e.g., background textures that the recoloring exploits). Details about the recoloring parametrization are given in Appendix E.7 and further non-curated samples are provided in Appendix F.3.6.

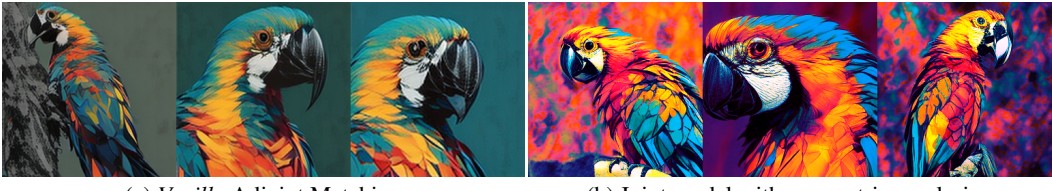

(a) *Vanilla* Adjoint Matching          (b) Joint model with parametric recoloring

Figure 6: Fine-tuning of LFM model on *macaw* class using prompt "close-up Pop Art of a macaw parrot", comparing *vanilla* Adjoint Matching with our joint approach.

## 5 CONCLUSION

We have introduced a framework for post-training fine-tuning of flow-matching generative models to enforce physical constraints and jointly infer latent physical parameters informing the constraints. Through a novel architecture, combined with the combination of weak-form PDE residuals with an adjoint-matching scheme our method can produce samples that adhere to complex constraints without significantly affecting the sample diversity. Experiments across PDE problems demonstrate the potential of this method to reduce residuals and enable joint solution–parameter generation, supporting its promise for physics-aware generative modeling. Future steps include adaptive approaches to optimizing trade-off between constraint enforcement and generative diversity, and extending the framework to more complex and multi-physics systems, including coupled PDEs and stochastic or chaotic dynamics. We would also explore how this methodology can be leveraged for uncertainty quantification and propagation, and downstream tasks such as optimal sensor placement and scientific discovery workflows.

## REPRODUCIBILITY STATEMENT

We report datasets, model backbones, training schedules, loss definitions, evaluation metrics, and the key hyperparameters required to reproduce our results in the main text and appendix. Remaining implementation choices are documented in the released configuration files. We fixed random seeds where applicable and specify hardware/software versions.

## ACKNOWLEDGEMENTS

This work was supported by funding from the German Federal Ministry for Education and Research (Bundesministerium für Bildung und Forschung, BMBF) under grants 16IS24071A / 16IS24071B and 01IW23005. SF additionally acknowledges support by the DFG through FOR 5359 (ID 459419731), TRR 375 (ID 511263698), and SPP 2331 (441958259, 553345933, 466468799), and by the Carl-Zeiss Foundation through the initiatives AI-Care and Process Engineering 4.0. We thank the anonymous reviewers for their comments.

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

## A    USE OF LARGE LANGUAGE MODELS

We employed large language models to polish the manuscript (wording, grammar, and synonyms) and to assist with implementation details, including plotting scripts and code rewriting/refactoring. The research questions, problem formulation, algorithmic design and experimental methodology, however, were conceived by the authors. All LLM-produced text and code were reviewed, adapted, and verified by the authors prior to inclusion.

## B    DATASET DETAILS

In this section we detail the datasets used throughout our study. Our guiding principle was to select scenarios in which the underlying parameter fields contain sharp discontinuities, thereby inducing rich, non-linear behavior in the associated state variables and making the inverse problem decidedly nontrivial. Although the Darcy-flow benchmark follows the setup of Li et al. (2020), we regenerate the data so that the sample count, grid resolution, and ground-truth parameters can be controlled precisely. Complete scripts for producing all data will be released to facilitate transparency and reproducibility.

### B.1    DARCY FLOW DATASET

The dataset comprises 20,000 independent samples, each a pair $(a, u)$ on the unit square $\Omega = [0, 1]^2$, where $a : \Omega \to \mathbb{R}_{>0}$ is the permeability and $u : \Omega \to \mathbb{R}$ is the steady-state pressure solving

$$-\nabla \cdot \big(a(\xi) \nabla u(\xi)\big) = f(\xi), \qquad \xi = (\xi_1, \xi_2) \in \Omega,$$

with homogeneous Dirichlet boundary conditions $u|_{\partial \Omega} = 0$ and constant forcing $f \equiv 1$.

**Discretization.** We use a uniform $64 \times 64$ nodal grid with spacing $\Delta x = 1/(64-1)$ and the standard five-point finite-difference scheme. Interface permeabilities are formed by two-point arithmetic averaging. Dirichlet values are imposed strongly, yielding a sparse SPD linear system that is solved with a direct sparse solver.

**Permeability sampling.** We draw a zero-mean Gaussian random field $a_{\mathrm{raw}}$ via a cosine-basis Karhunen–Loève synthesis on $\Omega$ associated with the Matérn-type covariance operator

$$\mathcal{C} = (\tau^2 - \Delta)^{-\alpha}, \qquad \alpha = 2, \ \ \tau = 3,$$

i.e., using the DCT-II basis (Neumann eigenfunctions of $-\Delta$), setting the DC mode to zero to enforce exact zero mean, scaling by the spectrum of $\mathcal{C}$, and applying an orthonormal inverse DCT to obtain a grid realization.
To model sharp contrasts, we threshold the Gaussian field into a piecewise-constant permeability,

$$a(\xi) = \begin{cases} 12, & a_{\mathrm{raw}}(\xi) \geq 0, \\ 3, & a_{\mathrm{raw}}(\xi) < 0. \end{cases}$$

**Observational noise.** In experiments with noisy observations, we corrupt the pressure with additive Gaussian noise

$$\tilde{u} = u + \sigma\,\varepsilon, \qquad \varepsilon \sim \mathcal{N}(0, I), \quad \sigma = 10^{-4}.$$

## B.2 Linear Elasticity Dataset

The dataset contains $N = 10{,}000$ independent samples on the unit square $\Omega = [0,1]^2$. Each sample is a pair $(E, u)$ where $E : \Omega \to \mathbb{R}_{>0}$ denotes the spatially varying Young's modulus and $u : \Omega \to \mathbb{R}^2$ is the plane–strain displacement field under fixed Poisson ratio $\nu = 0.30$. The boundary loading is deterministic and identical across samples: the left/right edges are clamped, and sinusoidal normal displacements are prescribed on the top and bottom edges with zero tangential slip (amplitudes $A_{\mathrm{top}} = A_{\mathrm{bot}} = 0.10$). For each $E$, $u$ solves the static linear elasticity equations (no body force)

$$-\nabla \cdot \sigma(\xi) = 0, \qquad \sigma = \lambda(E, \nu)\,\mathrm{tr}(\varepsilon)\,I + 2\,\mu(E, \nu)\,\varepsilon, \qquad \varepsilon = \tfrac{1}{2}\big(\nabla u + \nabla u^\top\big), \quad \xi \in \Omega,$$

with Lamé parameters $\lambda(E, \nu) = \frac{\nu E}{(1+\nu)(1-2\nu)}$ and $\mu(E, \nu) = \frac{E}{2(1+\nu)}$, and Dirichlet boundary conditions

$$u(\xi_1{=}0, \xi_2) = 0, \quad u(\xi_1{=}1, \xi_2) = 0$$

$$u_x(\xi_1, \xi_2{=}0) = 0, \ \ u_y(\xi_1, \xi_2{=}0) = -A_{\mathrm{bot}} \sin(\pi\,\xi_1)$$

$$u_x(\xi_1, \xi_2{=}1) = 0, \ \ u_y(\xi_1, \xi_2{=}1) = +A_{\mathrm{top}} \sin(\pi\,\xi_1).$$

During fine-tuning, we set $A_{\mathrm{bot}} = 0.075$ to enforce adaptation of the solutions, simulating a misspecification between observed data and the assumed model.

**Discretization.** We use a uniform $64 \times 64$ grid on $\Omega$ and standard second–order finite differences in the small–strain regime. Dirichlet data are imposed strongly. The discrete equilibrium equations are advanced by a stable explicit time–marching (damped gradient) scheme for the static problem until the global residual norm falls below $10^{-6}$, or a cap of $2 \times 10^4$ iterations is reached.

**Coefficient field sampling (piecewise-constant Voronoi medium).** Heterogeneous modulus fields $E$ are obtained from a Voronoi tessellation constructed by drawing a fixed number of sites uniformly in $\Omega$, assigning to each Voronoi cell a modulus sampled log-uniformly within $[1.0, 10.0]$, and rasterizing the resulting partition to the computational grid by nearest-site labeling. This produces piecewise-constant $E$ with sharp jumps that emulate multi-phase media. To control interface smoothness, a separable Gaussian blur with standard deviation $\sigma_{\mathrm{blur}} = 1.0$ (in grid units) is applied to the rasterized field.

### B.3 HELMHOLTZ WAVE PROPAGATION DATASET

The dataset consists of $N = 10{,}000$ independent samples on the unit square $\Omega = [0,1]^2$. Each sample is a pair $(c, u)$, where $c : \Omega \to \mathbb{R}_{>0}$ denotes the spatially varying sound speed and $u : \Omega \to \mathbb{C}$ is the time–harmonic acoustic pressure. For a fixed angular frequency $\omega > 0$, the field $u$ solves the damped Helmholtz equation

$$-\Delta u(\xi) - \left(1 - i \tan \delta\right) \kappa(\xi)^2 \, u(\xi) \;=\; s(\xi), \qquad \kappa(\xi) \;=\; \frac{\omega}{c(\xi)}, \qquad \xi \in \Omega,$$

subject to homogeneous Robin boundary conditions

$$\partial_n u(\xi) + i \, \gamma \, u(\xi) = 0, \qquad \xi \in \partial\Omega,$$

with fixed admittance parameter $\gamma > 0$ and loss tangent $\tan \delta \geq 0$.

**Discretization.** We use a uniform $64 \times 64$ nodal grid on $\Omega$ with spacing $\Delta x = 1/(64 - 1)$. The Laplacian $-\Delta$ is discretized by the standard five-point finite-difference stencil in the interior and homogeneous Neumann conditions in the normal direction on all sides, implemented via ghost-point elimination. The Robin condition $\partial_n u + i\gamma u = 0$ is enforced by adding purely imaginary diagonal contributions at boundary nodes to the stiffness matrix, consistent with the underlying Neumann-type discretization. The heterogeneous reaction term $-(1 - i \tan \delta)\kappa^2 u$ is assembled pointwise on the grid. This yields a sparse complex-valued linear system, which is solved for each sample using a direct sparse solver.

**Sound-speed sampling.** The sound speed $c$ is derived from a Gaussian random field constructed in the same manner as for the Darcy dataset, but interpreted as a wave speed. We first draw a zero-mean Gaussian field $g$ on the grid using a cosine-basis Karhunen–Loève synthesis associated with the Matérn-type covariance operator

$$\mathcal{C} \;=\; (\tau^2 - \Delta)^{-\alpha}, \qquad \alpha = 4, \;\; \tau = 6,$$

using the DCT-II basis on $\Omega$, zeroing the DC mode, scaling by the spectrum of $\mathcal{C}$, and applying the inverse DCT to obtain $g$. To model sharp material contrasts, we map $g$ to a two-level sound speed,

$$c(\xi) \;=\; \begin{cases} c_{\max}, & g(\xi) \geq 0, \\ c_{\min}, & g(\xi) < 0, \end{cases}$$

with $c_{\min} = 0.8$ and $c_{\max} = 1.2$ in the reported experiments.

**Source and damping.** The source term $s$ is a fixed, smooth Gaussian bump centered in $\Omega$,

$$s(\xi) \;=\; \exp\!\left(-\tfrac{\|\xi - (0.5,\,0.5)\|_2^2}{2\sigma^2}\right), \quad \sigma = 0.05,$$

shared across all samples. We use a fixed angular frequency $\omega = 20$ and Robin parameter $\gamma = 0.03$. Data is generated with a small but nonzero loss tangent $\tan \delta = 0.02$, leading to complex-valued damped solutions.

### B.4 STOKES LID-DRIVEN CAVITY DATASET

The dataset consists of $N = 10{,}000$ independent samples on the unit square $\Omega = [0,1]^2$. Each sample is a pair $(\nu, u)$, where $\nu : \Omega \to \mathbb{R}_{>0}$ denotes the spatially varying kinematic viscosity and $u : \Omega \to \mathbb{R}^2$ is the steady incompressible velocity field with associated pressure $p : \Omega \to \mathbb{R}$. For each $\nu$, $(u, p)$ solve the variable-coefficient Stokes system

$$-\nabla \cdot \left(\nu(\xi) \, \nabla u(\xi)\right) + \nabla p(\xi) \;=\; f(\xi), \qquad \nabla \cdot u(\xi) \;=\; 0, \qquad \xi \in \Omega,$$

with a smooth lid-driven cavity boundary condition on the top wall

$$u(\xi_1, \xi_2{=}1) \;=\; \left(U_{\text{lid}} \sin^2(\pi\, \xi_1),\, 0\right),$$

and no-slip conditions on the remaining walls

$$u(\xi_1, \xi_2{=}0) = 0, \qquad u(\xi_1{=}0, \xi_2) = 0, \qquad u(\xi_1{=}1, \xi_2) = 0.$$

To excite richer flow structures, we add a Kolmogorov-type body force

$$f(\xi_1, \xi_2) \;=\; F_0\left(\sin(\pi\, \xi_2),\, 0\right),$$

with fixed amplitude $F_0 > 0$.

**Discretization.** The Stokes system is discretized using FEniCS (Baratta et al., 2023; Scroggs et al., 2022b;a; Alnaes et al., 2014) on a conforming triangular mesh obtained from a $(64-1) \times (64-1)$ grid of squares, each split into two triangles. We employ Taylor–Hood elements (vector-valued $P_2$ for $u$, scalar $P_1$ for $p$) on the mixed space $V \times Q$. The weak form reads

$$\int_\Omega \nu \, \nabla u : \nabla v \, dx - \int_\Omega p \, \nabla \cdot v \, dx - \int_\Omega q \, \nabla \cdot u \, dx \; = \; \int_\Omega f \cdot v \, dx,$$

for all test functions $(v, q) \in V \times Q$, with the lid and no-slip boundary conditions imposed strongly. For each sample, the resulting saddle-point system is solved with a direct solver, and the solution $(u, p)$ is sampled on a uniform $64 \times 64$ grid on $\Omega$; the pressure is post-processed to have zero spatial mean in order to fix the gauge.

**Viscosity field sampling.** The viscosity $\nu$ is obtained from a Gaussian random field constructed in the same manner as for the Darcy and Helmholtz datasets. We first draw a zero-mean Gaussian field $g$ on the grid using a cosine-basis Karhunen–Loève synthesis associated with the Matérn-type covariance operator

$$\mathcal{C} \; = \; (\tau^2 - \Delta)^{-\alpha}, \qquad \alpha = 4, \;\; \tau = 6,$$

using the DCT-II basis on $\Omega$, zeroing the DC mode, scaling by the spectrum of $\mathcal{C}$, and applying the inverse DCT to obtain $g$. To model sharp viscosity contrasts, we map $g$ to a two-level field,

$$\nu(\xi) \; = \; \begin{cases} \nu_{\max}, & g(\xi) \geq 0, \\ \nu_{\min}, & g(\xi) < 0, \end{cases}$$

with $\nu_{\min} = 0.02$ and $\nu_{\max} = 0.5$ in the reported experiments.

**Lid and body-force parameters.** In all samples we fix the lid velocity scale to $U_{\text{lid}} = 1.0$ and the Kolmogorov forcing amplitude to $F_0 = 2.0$.

## C  PRE-TRAINING OF FLOW MATCHING MODELS

We adopt the vanilla Flow–Matching (FM) procedure of Lipman et al. (2023). Based on the reference flow

$$X_t = \beta_t X_1 + \gamma_t X_0$$

with $\beta_t = t$, and $\gamma_t = 1 - t$, we can define the conditional vector field as training targets for a parametric model $v_\theta(x_t, t)$. Given an end-point $x_1 \sim p_{\text{data}}$, the conditional vector field is available as

$$v_t(x|x_1) = \frac{1}{1-t}(x_1 - x).$$

This leads to the simplest form of Flow Matching objectives:

$$\mathcal{L}_{\text{FM}}^{\text{OT}} = \mathbb{E}\|v_\theta(X_t, t) - (X_1 - X_0)\|^2$$

where $X_0 \sim \mathcal{N}(0, I)$, $X_1 \sim p_{\text{data}}$ and $t \sim U[0, 1]$.

**Network Backbone.** For PDE data, the mapping $v_\theta$ is realised with a U-FNO (Wen et al., 2022), which combines Fourier Neural Operator (FNO, Li et al. (2020)) with U–Net (Ronneberger et al., 2015) layers. The FNO layers have an inductive bias towards low-frequency solutions and are therefore particularly suited for modeling PDE data, while the U–Net layers help to refine outputs and produce discontinuities.

Departing from the original design, we use the U–Net skip-connection structure in all layers. Also, we employ a more powerful time-conditioning in the U–Net layers by using FiLM-style Adaptive Group Normalization (AdaGN) time conditioning, i.e., predicting per-channel scale and shift from a sinusoidal time embedding and applying them after GroupNorm in each residual block. This follows the feature-wise linear modulation idea of Perez et al. (2018) and its diffusion U–Net instantiations in Dhariwal & Nichol (2021); Rombach et al. (2022).

We prepend the physical input with fixed sinusoidal encodings for absolute spatial coordinates and the normalized time stamp so the spectral backbone receives explicit space-time context. Padding

inside the U-FNO (for the spectral convolution layers) is reflective, which has empirically worked better than replicative. Before training, data is standardised to zero mean and unit variance.

Table 3 compiles the hyperparameters of the U-FNO architecture used for the base FM models in our experiments.

Table 3: U–FNO (2D) backbone hyperparameters used in our experiments.

| Hyperparameter | Value | Description |
|---|---|---|
| Number of layers | 4 | Count of spectral and U–Net operator blocks. |
| Padding mode | reflect | Boundary padding for convolutions/lifts. |
| Input channels | 1 - 3 | Channels of input field $x$. |
| Output channels | 1 - 3 | Channels of output field. |
| Time embedding dim | 32 | Dimensionality of time conditioning. |
| Spatial embedding dim | 32 | Dimensionality of coordinate embedding. |
| Lifting width | 256 | Channels in input lifting/projection. |
| Fourier modes (kx, ky) | [32, 32] | Retained spectral modes per axis. |
| Spectral block width | 32 | Channel width within FNO layers. |
| U–Net base widths | 32 | Stage-wise channel widths. |
| Embedding width (U–Net) | 64 | Channels for conditioning embeddings. |
| Attention stages | [] | Stages with self-attention (empty $\Rightarrow$ none). |
| Attention heads | [] | Multi-head count if attention is enabled. |
| Total number of parameters | $\approx$ 19M | |

On natural image data, we use a pre-trained Latent Flow Matching Model (Dao et al., 2023) based on a DiT (Peebles & Xie, 2023) backbone.

**Optimization.** We train the FM backbone with AdamW (Loshchilov & Hutter, 2017), using a linear warmup of the learning rate from $0$ to the base value over the first $p_{\mathrm{wu}}$ fraction of training steps, followed by a constant learning rate thereafter. For evaluation stability, we maintain an exponential moving average (EMA) of the network parameters $\theta^{\mathrm{EMA}} \leftarrow \beta\,\theta^{\mathrm{EMA}} + (1 - \beta)\,\theta$, a practice rooted in Polyak averaging (Polyak & Juditsky, 1992) and widely adopted in modern deep models (e.g., Tarvainen & Valpola, 2017).

Table 4: Flow Matching (FM) training hyperparameters and schedule.

| Hyperparameter | | Description |
|---|---|---|
| Batch size | 128 | Minibatch size per step. |
| Base learning rate | 1e-4 | AdamW step size. |
| Warmup fraction $p_{\mathrm{wu}}$ | 0.01 | Fraction of total steps used for linear LR warmup. |
| Epochs | 300 | Full passes over the dataset. |
| Optimizer | AdamW | Applied to FM backbone parameters. |
| Weight decay | 0.01 | AdamW L2/decoupled decay coefficient. |
| EMA decay | 0.9998 | Exponential moving average of weights for evaluation. |
| Schedule after warmup | constant | LR held constant after warmup. |

The FM training configuration is summarized in Table 4. For the Darcy dataset (20k samples), training takes around 12 hours on a single NVIDIA RTXA6000 GPU. For the smaller datasets of 10k samples but with more channels (Elasticity, Helmholtz, Stokes), training takes around 7 hours using the same configuration.

## D METHOD: IMPLEMENTATION DETAILS

In this section, we provide further details on the implementation of our fine-tuning method. This includes neural network architectures, the specific design of test functions for the computation of the weak residuals, numerical heuristics, and finally the full training algorithm for fine-tuning.
All relevant code is implemented using Python 3.12.3, specifically, neural architectures are implemented in PyTorch Paszke et al. (2019) version 2.7.

### D.1 INVERSE PREDICTOR

We parametrise the inverse map $\varphi$ with a two-layer U-FNO, mirroring the spectral–spatial bias of the forward backbone. However, we increase the capacity of the U–Net components and use attention at the two lowest resolutions (stage indices 2 and 3). Since the inverse predictor only operates at $t = 1$, we drop the temporal but keep the spatial conditioning. Exact parameters are stated in Table 5.

Table 5: U-FNO Inverse Predictor.

| Hyperparameter | Value |
|---|---|
| Number of layers | 2 |
| Padding mode | reflect |
| Input channels | 1 - 3 |
| Time embedding dimension | 0 |
| Spatial embedding dimension | 32 |
| Output channels | 1 |
| Lifting width | 256 |
| Fourier modes $(k_x, k_y)$ | [32, 32] |
| Spectral block width | 32 |
| U–Net base widths | [64, 64, 96, 128] |
| U–Net embedding width | 64 |
| Attention stages | [2, 3] |
| Attention heads | 4 |

### D.2 ARCHITECTURE MODIFICATIONS

Fine-tuning augments the base U–FNO map $x \mapsto v_{t,x}^{\text{base}}(x, t)$ to a joint, $\alpha$-conditioned vector field

$$(v_{t,x}^{\text{ft}}, v_{t,\alpha}^{\text{ft}}) = v^{\text{ft}}(x, \alpha, t),$$

implemented as residual corrections around the U–FNO core. Given the padded input $x_{\text{pad}}$ (with time/space embeddings) and conditioning fields $\alpha$ and $v_{t,\alpha}^{\text{base}}$, the base feature stack $x_*$ is produced by the original spectral+skip+U–Net blocks. A first *correction head* (a U–Net) takes $[x_*, v_{t,x}^{\text{base}}, \alpha]$ and outputs an additive refinement, yielding

$$v_{t,x}^{\text{ft}} = v_{t,x}^{\text{base}} + \mathcal{U}_x\big(x_*, v_{t,x}^{\text{base}}, \alpha, t\big).$$

After unpadding, a *pixel-wise correction* (lightweight channel-wise MLP) further adjusts $v_{t,x}^{\text{ft}}$ using local features and 2D positional channels,

$$v_{t,x}^{\text{ft}} \leftarrow v_{t,x}^{\text{ft}} + \mathcal{M}_x\big(\text{pos}(x_*, t), v_{t,x}^{\text{ft}}\big),$$

which provides extra capacity for rapid adaptation without altering the global operator. For the parameter dynamics, we adopt a strictly residual strategy that conditions on both $\alpha$ and the baseline field:

$$v_{t,\alpha}^{\text{ft}} = v_{t,\alpha}^{\text{base}} + \mathcal{U}_\alpha\big(x_{\text{pad}}, \alpha, v_{t,\alpha}^{\text{base}}, t\big),$$

where $\mathcal{U}_\alpha$ is a second U–Net. All correction heads are zero-initialized at their final projection layers, so that at the start of fine-tuning $v^{\text{ft}} \equiv (v_{t,x}^{\text{base}}, v_{t,\alpha}^{\text{base}})$ and departures are learned smoothly. Table 6 lists the parameters of the correction U–Nets.

Table 6: Parameterization of the fine-tuning U–Net heads $\mathcal{U}_x$ and $\mathcal{U}_\alpha$.

| Hyperparameter | Value |
|---|---|
| U–Net base widths | [64, 64, 96, 128] |
| Embedding width (time/aux) | 64 |
| Hidden lift (internal width) | 256 |
| Self-attention stages | [2, 3] |
| Attention heads | 4 |

Overall, the modifications to the base architecture add around 6M parameters to the model, resulting in a total $\approx$25M parameters.

### D.3 WEAK RESIDUALS AND TEST FUNCTIONS

**Darcy Flow.** A pressure field $u$ that solves our Darcy flow equations fulfills $-\nabla\cdot(a\nabla u) - f = \mathcal{L}_a u = 0$ on $\Omega \subset \mathbb{R}^d$ with homogeneous Dirichlet data $u|_{\partial\Omega} = 0$. For any $\psi \in C_0^1(\Omega)$ we compute the L$^2$ inner product by multiplying with $\psi$ and integrating:

$$\langle \mathcal{L}_a u, \psi \rangle_{L^2(\Omega)} = \int_\Omega \left( -\nabla\cdot(a\nabla u) - f \right) \psi \, d\xi$$

One application of the divergence theorem yields

$$\begin{aligned}
\langle \mathcal{L}_a u, \psi \rangle_{L^2(\Omega)} &= -\int_{\partial\Omega} (a\nabla u \cdot n)\, \psi \, d\xi \\
&\quad + \int_\Omega (a\, \nabla u \cdot \nabla \psi) - f\, \psi \, d\xi \\
&= \int_\Omega (a\, \nabla u \cdot \nabla \psi) - f\, \psi \, d\xi,
\end{aligned}$$

where the boundary term vanishes because of $\psi|_{\partial\Omega} = 0$. This expression only contains a first-order derivative of $u$ and can be used to compute in the computation of the weak residual. To obtain a dimensionless, coefficient-scaled quantity comparable across locations, we normalize by the local mean permeability over the support of $\psi$,

$$\bar{a}_\psi := \frac{1}{|\operatorname{supp}\psi|} \int_{\operatorname{supp}\psi} a(\xi)\, d\xi, \qquad \widetilde{\mathcal{R}}[\psi] := \frac{\mathcal{R}[\psi]}{\bar{a}_\psi}.$$

In practice, $\operatorname{supp}\psi$ is the compact patch where $\psi$ (or its mollified variant) is nonzero, so that $\bar{a}_\psi$ captures the local coefficient scale used to normalize the residual.

**Linear Elasticity.** For any compactly supported vector test $\psi \in C_0^1(\Omega; \mathbb{R}^2)$, the weak residual is

$$\mathcal{R}[\psi] := \langle \mathcal{L}_E u, \psi \rangle_{L^2(\Omega)} = \int_\Omega \left( -\nabla\cdot\sigma(u; E, \nu) \right)\cdot\psi \, d\xi.$$

A single integration by parts yields

$$\mathcal{R}[\psi] = -\int_{\partial\Omega} (\sigma n)\cdot\psi \, dS + \int_\Omega \sigma : \nabla\psi \, d\xi = \int_\Omega \sigma : \nabla\psi \, d\xi,$$

since $\psi|_{\partial\Omega} = 0$. Here "$\sigma : \nabla\psi$" is the Frobenius product and we denote the stress components by $\sigma_{xx}, \sigma_{xy}(= \sigma_{yx}), \sigma_{yy}$. To reuse the same scalar test generator for both momentum equations, we restrict to tests that share a single scalar profile in both components. Concretely, we take

$$\psi^{(x)} = (\psi, 0), \qquad \psi^{(y)} = (0, \psi), \qquad \psi \in C_0^1(\Omega).$$

With this restriction, residuals can be computed component-wise

$$\langle \mathcal{L}_E u, \psi^{(x)} \rangle = \int_\Omega \big( \sigma_{xx} \, \partial_{\xi_1} \psi + \sigma_{xy} \, \partial_{\xi_2} \psi \big) \, d\xi, \qquad \langle \mathcal{L}_E u, \psi^{(y)} \rangle = \int_\Omega \big( \sigma_{xy} \, \partial_{\xi_1} \psi + \sigma_{yy} \, \partial_{\xi_2} \psi \big) \, d\xi.$$

To obtain a dimensionless, coefficient-scaled quantity, we normalize by the local mean modulus over the support of $\psi$,

$$\overline{E}_\psi := \frac{1}{|\operatorname{supp} \psi|} \int_{\operatorname{supp} \psi} E(\xi) \, d\xi, \qquad \widetilde{\mathcal{R}}^{(x)}[\psi] := \frac{\langle \mathcal{L}_E u, \psi^{(x)} \rangle}{\overline{E}_\psi}, \quad \widetilde{\mathcal{R}}^{(y)}[\psi] := \frac{\langle \mathcal{L}_E u, \psi^{(y)} \rangle}{\overline{E}_\psi}.$$

For a family of tests $\{\psi_k\}_{k=1}^N$, the scalar residual used in experiments is the $\ell_2$ aggregation over components followed by averaging over tests. Therefore, the weak residual for the elasticity experiment is

$$\mathcal{R}_{\mathrm{weak}} := \frac{1}{N} \sum_{k=1}^N \Big( \big( \widetilde{\mathcal{R}}^{(x)}[\psi_k] \big)^2 + \big( \widetilde{\mathcal{R}}^{(y)}[\psi_k] \big)^2 \Big).$$

**Helmholtz.** For the Helmholtz experiments we consider complex-valued time–harmonic pressures $u : \Omega \to \mathbb{C}$ satisfying

$$\mathcal{L}_c u := -\Delta u(\xi) - \big( 1 - i \tan \delta \big) \kappa(\xi)^2 u(\xi) - s(\xi) = 0, \qquad \kappa(\xi) = \frac{\omega}{c(\xi)},$$

where $c(\xi)$ is the sound speed, $\omega > 0$ the angular frequency, $\tan \delta \geq 0$ the loss tangent, and $s(\xi)$ a fixed source. Writing $u = u_R + i u_I$ with real and imaginary parts $u_R, u_I : \Omega \to \mathbb{R}$, we split the operator into its real and imaginary components

$$\mathcal{L}_c u = \big( \mathcal{L}_c^{(R)} u_R + \mathcal{C}^{(R)} u_I - s \big) + i \big( \mathcal{L}_c^{(I)} u_I + \mathcal{C}^{(I)} u_R \big),$$

with

$$\mathcal{L}_c^{(R)} u_R := -\Delta u_R - \kappa^2 u_R, \qquad \mathcal{L}_c^{(I)} u_I := -\Delta u_I - \kappa^2 u_I,$$
$$\mathcal{C}^{(R)} u_I := \alpha_I u_I, \qquad \mathcal{C}^{(I)} u_R := -\alpha_I u_R,$$

where $\alpha_R := \kappa^2$ and $\alpha_I := -\tan \delta \, \kappa^2$. For any scalar test function $\psi \in C_0^1(\Omega)$, we define real and imaginary weak residuals as

$$\mathcal{R}^{(R)}[\psi] := \big\langle \mathcal{L}_c^{(R)} u_R + \mathcal{C}^{(R)} u_I - s, \, \psi \big\rangle_{L^2(\Omega)}, \qquad \mathcal{R}^{(I)}[\psi] := \big\langle \mathcal{L}_c^{(I)} u_I + \mathcal{C}^{(I)} u_R, \, \psi \big\rangle_{L^2(\Omega)}.$$

A single integration by parts in the Laplacian terms gives

$$\mathcal{R}^{(R)}[\psi] = -\int_{\partial\Omega} (\nabla u_R \cdot n) \, \psi \, dS + \int_\Omega \big( \nabla u_R \cdot \nabla \psi - \alpha_R u_R \psi + \alpha_I u_I \psi - s \, \psi \big) \, d\xi,$$

$$\mathcal{R}^{(I)}[\psi] = -\int_{\partial\Omega} (\nabla u_I \cdot n) \, \psi \, dS + \int_\Omega \big( \nabla u_I \cdot \nabla \psi - \alpha_R u_I \psi - \alpha_I u_R \psi \big) \, d\xi.$$

Since $\psi$ is compactly supported in the interior, the boundary contributions vanish and we use only the volume terms in the residual. As in the elasticity case, we reuse the same scalar profile $\psi$ for both channels (real and imaginary parts) and evaluate $\mathcal{R}^{(R)}[\psi]$ and $\mathcal{R}^{(I)}[\psi]$ on identical compact patches to increase computational efficiency.

To obtain a dimensionless quantity that is comparable across locations and coefficient scales, we normalize by a local Helmholtz energy associated with the bilinear form

$$E_\psi(u) := \int_{\operatorname{supp} \psi} \Big( |\nabla u_R|^2 + |\nabla u_I|^2 + \kappa(\xi)^2 \big( u_R(\xi)^2 + u_I(\xi)^2 \big) \Big) \, d\xi,$$

which combines gradient and reaction contributions of both channels. The normalized residuals are

$$\widetilde{\mathcal{R}}^{(R)}[\psi] := \frac{\mathcal{R}^{(R)}[\psi]}{\sqrt{E_\psi(u)}}, \qquad \widetilde{\mathcal{R}}^{(I)}[\psi] := \frac{\mathcal{R}^{(I)}[\psi]}{\sqrt{E_\psi(u)}}.$$

Given a family of scalar tests $\{\psi_k\}_{k=1}^N$, the scalar weak residual used in experiments is obtained by aggregating real and imaginary components and averaging over tests,

$$\mathcal{R}_{\mathrm{weak}} := \frac{1}{N} \sum_{k=1}^N \Big( \big( \widetilde{\mathcal{R}}^{(R)}[\psi_k] \big)^2 + \big( \widetilde{\mathcal{R}}^{(I)}[\psi_k] \big)^2 \Big).$$

**Stokes lid-driven cavity.** For the Stokes experiments, each sample contains a velocity–pressure pair $(u, p)$ solving the variable–viscosity incompressible Stokes system

$$\mathcal{L}_\nu(u, p) := \left(-\nabla \cdot (\nu \nabla u) + \nabla p, \ \nabla \cdot u\right) = (0, 0) \qquad \text{in } \Omega,$$

with $\nu : \Omega \to \mathbb{R}_{>0}$ denoting the viscosity field. Writing $u = (u_x, u_y)$, the operator decomposes into

$$\mathcal{L}_\nu^{(x)}(u, p) = -\nabla \cdot (\nu \nabla u_x) + \partial_{\xi_1} p, \qquad \mathcal{L}_\nu^{(y)}(u, p) = -\nabla \cdot (\nu \nabla u_y) + \partial_{\xi_2} p,$$

$$\mathcal{L}_\nu^{(\mathrm{div})}(u, p) = \partial_{\xi_1} u_x + \partial_{\xi_2} u_y.$$

For any scalar test function $\psi \in C_0^1(\Omega)$, we form momentum and incompressibility residuals by testing each of these three equations with the same scalar profile,

$$\mathcal{R}^{(x)}[\psi] := \left\langle \mathcal{L}_\nu^{(x)}(u, p), \psi \right\rangle_{L^2(\Omega)}, \qquad \mathcal{R}^{(y)}[\psi] := \left\langle \mathcal{L}_\nu^{(y)}(u, p), \psi \right\rangle_{L^2(\Omega)},$$

$$\mathcal{R}^{(\mathrm{div})}[\psi] := \left\langle \mathcal{L}_\nu^{(\mathrm{div})}(u, p), \psi \right\rangle_{L^2(\Omega)}.$$

As in the Darcy and elasticity settings, one integration by parts isolates only first derivatives of the solution inside the volume integral. Since $\psi$ has compact support inside $\Omega$, all boundary terms vanish and we obtain the local weak forms

$$\mathcal{R}^{(x)}[\psi] = \int_\Omega \left( \nu \, \nabla u_x \cdot \nabla \psi + (\partial_{\xi_1} p) \, \psi \right) d\xi,$$

$$\mathcal{R}^{(y)}[\psi] = \int_\Omega \left( \nu \, \nabla u_y \cdot \nabla \psi + (\partial_{\xi_2} p) \, \psi \right) d\xi,$$

$$\mathcal{R}^{(\mathrm{div})}[\psi] = \int_\Omega \left( \partial_{\xi_1} u_x + \partial_{\xi_2} u_y \right) \psi \, d\xi.$$

To obtain a dimensionless, coefficient–scaled residual comparable across locations, we normalize by the local viscosity–weighted kinetic energy over the support of $\psi$,

$$E_\psi(u) := \int_{\mathrm{supp}\,\psi} \left( \nu(\xi) \, \|\nabla u(\xi)\|_F^2 + \|u(\xi)\|_2^2 \right) d\xi,$$

where $\|\nabla u\|_F^2 = |\partial_{\xi_1} u_x|^2 + |\partial_{\xi_2} u_x|^2 + |\partial_{\xi_1} u_y|^2 + |\partial_{\xi_2} u_y|^2$. The normalized weak residuals are

$$\widetilde{\mathcal{R}}^{(x)}[\psi] := \frac{\mathcal{R}^{(x)}[\psi]}{\sqrt{E_\psi(u)}}, \qquad \widetilde{\mathcal{R}}^{(y)}[\psi] := \frac{\mathcal{R}^{(y)}[\psi]}{\sqrt{E_\psi(u)}}, \qquad \widetilde{\mathcal{R}}^{(\mathrm{div})}[\psi] := \frac{\mathcal{R}^{(\mathrm{div})}[\psi]}{\sqrt{E_\psi(u)}}.$$

Finally, for a collection of scalar tests $\{\psi_k\}_{k=1}^N$, the scalar weak residual used in experiments aggregates momentum and incompressibility contributions and averages across tests:

$$\mathcal{R}_{\mathrm{weak}} := \frac{1}{N} \sum_{k=1}^N \left( \left(\widetilde{\mathcal{R}}^{(x)}[\psi_k]\right)^2 + \left(\widetilde{\mathcal{R}}^{(y)}[\psi_k]\right)^2 + \left(\widetilde{\mathcal{R}}^{(\mathrm{div})}[\psi_k]\right)^2 \right).$$

**Wendland–wavelet test family.** For estimating the weak residual accurately and to provide a strong learning signal, we need to sample sufficiently many test functions. Evaluating them on the entire computational grid would be prohibitively costly. We therefore consider test functions which are locally supported, such that we can restrict computations to smaller patches. Wendland polynomials are a natural candidate meeting these requirements since they are compactly supported within unit radius and allow for efficient gradient computation. Here, we will describe the test functions in detail.

Each test function is drawn from a radially anisotropic family.

$$\psi_{c,\sigma,b}(x) = \underbrace{\left(1 - r(x)\right)_+^4 \left(4r(x) + 1\right)}_{\text{Wendland } C^2} \underbrace{\left(1 - 64\,b\,r(x)^4\right)}_{\text{optional wavelet}},$$

where $r(x) = \sqrt{\sum_j (x_j - c_j)^2/\sigma_j^2}$. Length-scales $\sigma_j$ are uniformly sampled from the range $[\sigma_{\min} \Delta_j, \sigma_{\max} \Delta_j]$. By multiplying $\sigma_{\min}$ and $\sigma_{\max}$ with the grid spacing $\Delta_j$ of axis $j$, we obtain a parametrization that is intuitive to tune since the length-scales of the test functions are given in *pixel* units. Instead of also sampling center points $c$ uniformly and independently from the full domain $\Omega$, we instead start from the grid coordinates of the data, considering each grid point as one center. We found that this way of ensuring spatial coverage improves training efficiency. To still retain stochasticity, we apply i.i.d jitter to each center point. $b \sim \mathrm{Ber}(p)$ randomly toggles a high-frequency wavelet factor that provides additional variability within the test functions and proved especially effective on noisy data.

To enforce $\psi_i|_{\partial\Omega} = 0$, we multiply every test function by a *bridge mollifier*

$$m(\xi) = \big((\xi - \xi_{\min})(\xi_{\max} - \xi)\big)/(\xi_{\max} - \xi_{\min})^2,$$

applied per axis. This legitimizes the application of integration-by-parts in the derivation of the weak forms. At training time we sample a set of test functions $\{\psi^{(i)}\}_{i=1}^{N_{\text{test}}}$ independently per residual evaluation and define the loss as

$$R_{\text{weak}}(x, \alpha) = \frac{1}{N_{\text{test}}} \sum_i |\langle \mathcal{L}_\alpha x, \psi^{(i)} \rangle_{L^2}|^2.$$

### D.4 ADJOINT MATCHING DETAILS

As mentioned in the main paper, we introduce a scaling coefficient $\kappa$ that allows us to attenuate the magnitude of the noise variance. In this section, we provide a justification for using $\kappa > 0$ and lay out numerical heuristics used in the implementation.

**Memoryless Noise.** Domingo-Enrich et al. (2025) define a generative process with noise schedule $\sigma^2(t)$ to be memoryless, if and only if $\sigma^2(t) = 2\eta_t + \chi(t)$ with $\chi : [0,1] \to \mathbb{R}$ chosen such that

$$\forall t \in (0,1) \quad \lim_{t' \to 0} \beta_{t'} \, \exp\left(-\int_{t'}^t \frac{\chi(s)}{2\gamma_s^2} \, ds\right) = 0.$$

Specifically, they refer to $\sigma(t) = \sqrt{2\eta_t}$ as the memoryless noise schedule. In our setting of $\beta_t = t$ and $\gamma_t = 1 - t$, the memoryless noise schedule can be simplified to

$$\sigma(t) = \sqrt{\frac{2(1-t)}{t}}.$$

**Lemma 1** (Scaling of memoryless noise). *Consider a generative process as in 1 with $\beta_t = t$ and $\gamma_t = 1 - t$. For $0 \le \kappa < 1$, the schedule $\sigma^2(t) = (1 - \kappa)\, 2\eta_t$ is memoryless.*

*Proof.* First, we consider the integral term. The desired $\sigma^2(t) = (1 - \kappa)\, 2\eta_t$ implies that

$$\chi(t) = -2\kappa\eta_t = -2\gamma_t^2 \, \frac{\kappa}{t(1-t)}.$$

With this, we can simplify the integral term:

$$-\int_{t'}^t \frac{\chi(s)}{2\gamma_s^2} \, ds = \kappa \int_{t'}^t \frac{1}{s(1-s)} \, ds = \kappa \int_{t'}^t \frac{1}{s} + \frac{1}{1-s} \, ds$$

$$= \kappa \left[\log(s) - \log(1-s)\right]_{t'}^t$$

$$= \kappa \left(\log \frac{t}{1-t} - \log \frac{t'}{1-t'}\right).$$

Thus, for an arbitrary fixed $t \in (0, 1)$, it holds that

$$\beta_{t'} \exp\left(-\int_{t'}^{t} \frac{\chi(s)}{2\gamma_s^2}\, ds\right) = t' \exp\left(\kappa\left(\log\frac{t}{1-t} - \log\frac{t'}{1-t'}\right)\right)$$

$$= t'\left(\frac{t}{1-t}\right)^{\kappa}\left(\frac{t'}{1-t'}\right)^{-\kappa}$$

$$= \underbrace{\left(\frac{t}{1-t}\right)^{\kappa}}_{\text{const}} \underbrace{t'^{\,1-\kappa}}_{\to 0} \underbrace{(1-t')^{\kappa}}_{\text{bounded}} \xrightarrow[t'\to 0]{} 0.$$

$\square$

Therefore, scaling down the noise schedule by a factor $1 - \kappa$ is justified and consistent with the theory provided in Domingo-Enrich et al. (2025).

**Heuristics.** Still, the term $\eta_t$ causes numerical problems for $t = 0$. Furthermore, it forces the control $u$ to be close to zero for $t$ close to 1. Following the original paper, we instead use

$$\eta_t = \frac{1-t+h}{t+h},$$

where we choose $h$ as the step size of our numerical ODE/SDE solver. This resolves infinite values and allows for faster fine-tuning by letting the fine-tuned model deviate further from the base model close to $t = 1$.

While $\kappa$ is an effective tool to mitigate residual noise in final samples, increasing slows down training. As another way of improving sample quality without adding computational cost, we consider non-uniform time grids when sampling memoryless rollouts. We observe that the most critical phases of sampling are at the beginning, where noise magnitudes are the highest, and at the end, where final denoising happens. Therefore, we tilt the uniform grid towards the endpoints: Let $S \in \mathbb{N}$ be the number of steps and define the uniform grid $t_i = i/S$ for $i = 0, \ldots, S$. We tilt this grid toward the endpoints by first applying a cosine–ease mapping

$$g(t) = \tfrac{1}{2}\left(1 - \cos(\pi t)\right), \qquad t \in [0, 1],$$

which has $g(0) = 0$, $g(1) = 1$. For a mixing parameter $q \in [0, 1]$, the tilted times are the convex combination

$$\tilde{t}_i = (1-q)\, t_i + q\, g(t_i), \qquad i = 0, \ldots, S.$$

$\{\tilde{t}_i\}$ is strictly increasing and distributes grid points more densely near $t = 0$ and $t = 1$ for $q > 0$. For PDE experiments, we use $q = 0.9$. This heuristic was not needed for latent-space models.

**Loss Computation.** As in the original paper, we do not compute the Adjoint Matching loss (Equation 4) for all simulated time steps, since the gradient signal for successive time steps is similar. Note that for solving the lean adjoint ODE, we do not need to compute gradients through the FM model, therefore we can compute the lean adjoint states efficiently but save computational resources when computing the Adjoint Matching loss. Again, the last steps in the sampling process are most important for empirical performance. For that reason, we also compute the loss for a fraction of last steps $K_{\text{last}}$. Additionally, we sample $K$ steps from the remaining time steps.

To ensure stable learning, we apply a clipping function to the loss to exclude noisy high-magnitude gradients from training. Empirically, the values provided in Domingo-Enrich et al. (2025) work well in our setting, i.e. we set the loss clipping threshold (LCT) as $\text{LCT}_x = 1.6\,\lambda_x^2$ and $\text{LCT}_\alpha = 1.6\,\lambda_\alpha^2$ respectively.

### D.5 Full Training Algorithm

Algorithm 1 details the complete optimization loop used in all experiments. Starting from the pre-trained base flow $v^{\text{base}}$ we attach two residual heads that (i) condition the state flow $v_{t,x}^{\text{ft}}$ on the latent

parameter $\alpha$ and (ii) predict the parameter flow $v^{\text{ft}}_{t,\alpha}$. Their respective output layers are initialized to zero. The inverse predictor $\varphi$ is first pre-trained on base samples and then frozen, providing surrogate target flows for $\alpha$. Each epoch rolls both the base and fine-tuned trajectories on the tilted time grid, solves the lean-adjoint equation, and updates only the fine-tune parameters $\theta$ through the Adjoint-Matching loss.

---

**Algorithm 1** Adjoint Matching with Joint State–Parameter Evolution

---

**Notation:** $v^{\text{base}}_x, v^{\text{ft}}_x, v^{\text{ft}}_\alpha$ denote *functions* (neural vector fields). Evaluations at time $t$ are written in bold, e.g. $\mathbf{v}^{\text{base}}_{x,t} := v^{\text{base}}_x(x_t, t)$.

**Input:** base state field $v^{\text{base}}_x$; initialize $v^{\text{ft}} \leftarrow v^{\text{base}}_x$ and add heads for $v^{\text{ft}}_x, v^{\text{ft}}_\alpha$.

1: **Pretrain** $\varphi$ on $x_1$ samples from $v^{\text{base}}_x$ by minimizing $\mathcal{R}_{\text{weak}}(x_1, \varphi(x_1))$.
2: **Freeze** $v^{\text{base}}_x$ and $\varphi$; let $\theta$ be the trainable parameters of $v^{\text{ft}}$.
3: **for** epoch $= 1, \ldots$ **do**
4:      Sample $x_0 \sim \mathcal{N}(0, I)$ and $\alpha_0 \sim \mathcal{N}(0, I)$; $\bar{T} \leftarrow \text{GET\_TILTED\_TIME}(T)$.
5:
6:      **for** $i = 0$ to $L - 1$ **do**
7:          $t \leftarrow t_i, t^+ \leftarrow t_{i+1}, h \leftarrow t^+ - t$.
8:          **Base state velocity:**
9:              $\mathbf{v}^{\text{base}}_{x,t} = v^{\text{base}}_x(x_t, t)$.
10:          **Surrogate terminal parameter:**
11:              $\hat{x}_1 = x^{\text{ft}}_t + (1 - t)\, \mathbf{v}^{\text{base}}_{x,t}$
12:              $\hat{\alpha}^{\text{ft}}_1 = \varphi(\hat{x}_1)$.
13:          **Base parameter velocity (surrogate):**
14:              $\mathbf{v}^{\text{base}}_{\alpha,t} = (\hat{\alpha}^{\text{ft}}_1 - \alpha_t)/(1 - t)$.
15:          **Fine-tuned velocities:**
16:              $\mathbf{v}^{\text{ft}}_{x,t} = v^{\text{ft}}_x(x_t, \alpha_t, \mathbf{v}^{\text{base}}_{\alpha,t}, t)$
17:              $\mathbf{v}^{\text{ft}}_{\alpha,t} = v^{\text{ft}}_\alpha(x_t, \alpha_t, \mathbf{v}^{\text{base}}_{\alpha,t}, t)$.
18:          **Forward steps (with noise $\sigma(t)$):**
19:              $x^{\text{base}}_{t^+} = \text{STEP}(x^{\text{base}}_t, \mathbf{v}^{\text{base}}_{x,t}, \sigma(t), h)$
20:              $x^{\text{ft}}_{t^+} = \text{STEP}(x^{\text{ft}}_t, \mathbf{v}^{\text{ft}}_{x,t}, \sigma(t), h)$
21:              $\alpha^{\text{ft}}_{t^+} = \text{STEP}(\alpha^{\text{ft}}_t, \mathbf{v}^{\text{ft}}_{\alpha,t}, \sigma(t), h)$.
22:      **end for**
23:
24:      $(a_{x,t}, a_{\alpha,t})_{t \in \bar{T}} \leftarrow \text{SOLVELEANADJOINT}(\{x^{\text{base}}_t\}, \{x^{\text{ft}}_t\}, \{\alpha^{\text{ft}}_t\})$.
25:
26:      **Subsample time indices:** form $\mathcal{I} \subset \{0, \ldots, L\}$ according to $K$ and $L_{\text{last}}$
27:
28:      **Per-step drifts:** for each $t_i \in \mathcal{I}$ compute
29:          $\mathbf{b}^{\text{ft}}_{x,t_i} = \mathbf{v}^{\text{ft}}_{x,t_i} + \frac{\sigma(t_i)^2}{2\,\eta(t_i)}\big(\mathbf{v}^{\text{ft}}_{x,t_i} - \frac{1}{t_i+\varepsilon} x^{\text{ft}}_{t_i}\big);$      $\mathbf{b}^{\text{base}}_{x,t_i} = \mathbf{v}^{\text{base}}_{x,t_i} + \frac{\sigma(t_i)^2}{2\,\eta(t_i)}\big(\mathbf{v}^{\text{base}}_{x,t_i} - \frac{1}{t_i+\varepsilon} x^{\text{base}}_{t_i}\big).$
30:          $\mathbf{b}^{\text{ft}}_{\alpha,t_i} = \mathbf{v}^{\text{ft}}_{\alpha,t_i} + \frac{\sigma(t_i)^2}{2\,\eta(t_i)}\big(\mathbf{v}^{\text{ft}}_{\alpha,t_i} - \frac{1}{t_i+\varepsilon} \alpha^{\text{ft}}_{t_i}\big);$      $\mathbf{b}^{\text{base}}_{\alpha,t_i} = \mathbf{v}^{\text{base}}_{\alpha,t_i} + \frac{\sigma(t_i)^2}{2\,\eta(t_i)}\big(\mathbf{v}^{\text{base}}_{\alpha,t_i} - \frac{1}{t_i+\varepsilon} \alpha^{\text{ft}}_{t_i}\big).$
31:
32:      **Per-step losses (with clipping):** for each $t_i \in \mathcal{I}$ set
33:          $\ell_x(t_i) = \min\big\{ [\sigma(t_i)^{-1}(\mathbf{b}^{\text{ft}}_{x,t_i} - \mathbf{b}^{\text{base}}_{x,t_i}) + \sigma(t_i)\, a_{x,t_i}]^2, \text{ LCT}_x \big\},$
34:          $\ell_\alpha(t_i) = \min\big\{ [\sigma(t_i)^{-1}(\mathbf{b}^{\text{ft}}_{\alpha,t_i} - \mathbf{b}^{\text{base}}_{\alpha,t_i}) + \sigma(t_i)\, a_{\alpha,t_i}]^2, \text{ LCT}_\alpha \big\},$
35:          (squaring and clipping applied elementwise, then summed over spatial dimensions)
36:
37:      **Adjoint–matching objective:**
38:          $\mathcal{L} = \frac{1}{|\mathcal{I}|} \sum_{t_i \in \mathcal{I}} \big(\ell_x(t_i) + \ell_\alpha(t_i)\big).$
39:          $\theta \leftarrow \text{GRADIENT\_DESCENT}(\theta, \mathcal{L})$
40: **end for**
41: **return** $v^{\text{ft}}, \varphi$

---

# E   DETAILS: EXPERIMENTS

Here we describe the fine-tuning configuration of the conducted experiments. Specifics about the base model FM training can be found in Appendix C. In all experiments, we use AdamW as the optimizer with a weight decay of $0.01$ and all fine-tuning is conducted on a single NVIDIA L40S GPU. As reported in the main paper, fine-tuning for Darcy takes less than one minute per epoch. For the other, computationally more expensive experiments, one epoch of fine-tuning takes around one minute (1-2 hours for 100 epochs).

## E.1   METRICS

**SSIM diversity.**   In panel (a) of Figure 3, we report a SSIM-based (Wang et al., 2004) metric for diversity, which is implemented as follows:

Given a batch $\{\alpha_i\}_{i=1}^B$ of single-channel parameter maps scaled to $[0, 1]$, we define diversity as the mean complement of the pairwise structural similarity index (SSIM) :

$$\mathcal{D}_{\text{SSIM}}(\{\alpha_i\}_{i=1}^B) \;=\; \frac{1}{\binom{B}{2}} \sum_{1 \leq i < j \leq B} \Big(1 - \text{SSIM}(\alpha_i, \alpha_j)\Big), \qquad \alpha_i \in [0, 1]^{H \times W}.$$

With $\text{SSIM} \in [0, 1]$ (data range $= 1$), $\mathcal{D}_{\text{SSIM}} \in [0, 1]$ and larger values indicate greater sample diversity.

**Maximum Mean Discrepancy (MMD).**   To quantify distributional differences between generated samples and the reference dataset $\mathcal{D}_{\text{ref}}$, we use the squared Maximum Mean Discrepancy ($\text{MMD}^2$) with an RBF kernel (Gretton et al., 2012). Given two batches $\{x_i\}_{i=1}^{N_1} \subset \mathbb{R}^D$ and $\{y_j\}_{j=1}^{N_2} \subset \mathbb{R}^D$, let

$$k_\sigma(u, v) = \exp\Big(-\tfrac{1}{2\sigma^2}\|u - v\|_2^2\Big), \qquad \gamma = \tfrac{1}{2\sigma^2}.$$

The empirical $\text{MMD}^2$ is

$$\text{MMD}^2(x, y) = \frac{1}{N_1^2} \sum_{i,i'} k_\sigma(x_i, x_{i'}) + \frac{1}{N_2^2} \sum_{j,j'} k_\sigma(y_j, y_{j'}) - \frac{2}{N_1 N_2} \sum_{i,j} k_\sigma(x_i, y_j).$$

In all experiments we use the median heuristic (Gretton et al., 2012), which sets the kernel bandwidth $\sigma^2$ to the median of the pairwise squared distances across the pooled samples $z = \{x_i\} \cup \{y_j\}$. In practice we subsample up to 1000 points to compute

$$\sigma^2 = \text{median}\big\{\|z_a - z_b\|_2^2 : a < b\big\} + \varepsilon, \qquad \varepsilon = 10^{-6},$$

which yields a stable and scale-adaptive kernel width (Garreau et al., 2017).

## E.2   ABLATIONS AND BASELINES

**Single Adjoint Matching.**   To isolate the effect of modeling a joint flow over $(x, \alpha)$, we include two ablations based on the vanilla Adjoint Matching (AM) objective of Domingo-Enrich et al. (2025). Both variants optimize the standard lean-adjoint control loss over the state trajectory $\{x_t\}$, using the same noise schedule, ODE discretization, and step subsampling as our full method.

*Base AM* freezes the inverse predictor $\varphi$. It is used only to compute PDE residuals but receives no gradient updates, and no joint flow over $\alpha$ is modeled.

*Base AM+$\varphi$* keeps $\varphi$ trainable and backpropagates through the residual term $r$, but still does not model a coupled flow over $(x, \alpha)$. This variant tests whether updating the inverse predictor alone can stabilize residuals without introducing a surrogate base flow.

**PBFM.**   As an alternative baseline we adapt Physics-Based Flow Matching (PBFM) (Baldan et al., 2025) to our setting. Since PBFM requires paired $(x, \alpha)$ information during training, we augment the method with our pre-trained inverse predictor $\varphi$ to estimate $\alpha$ and enable residual computation. Training follows the original ConFIG (Liu et al., 2024) update rule: at each iteration we compute the

FM loss and the residual loss, extract their gradients, and form a conflict-free update via ConFIG. Our implementation mirrors the official code except for minor modifications to support complex-valued weights (arising from FNO backbones). Due to memory constraints, we have to use much lower batch sizes than with vanilla Flow Matching (16 instead of 128). We chose the number of epochs such that the total number of update steps is twice that of our base FM models.

PBFM serves as a comparison against a training-time physics-regularized baseline, whereas our method enforces physics *post-training* (data-free). Note that all our PDE settings deliberately introduce a mismatch between the physics assumed during fine-tuning and the physics underlying the training data. Such misspecification is inherently challenging for training-time methods like PBFM which naturally places them at a disadvantage.

### E.3 DARCY

For the Darcy experiments, we use the hyperparameters listed in Table 7. Note that one epoch amounts to exactly one gradient descent step, meaning that we only fine-tune for 20 total steps. The parameters $\lambda_x, \lambda_\alpha$ and $\lambda_f$ are varied in the experiments, see Figure 3 in the main text.

Table 7: Darcy fine-tuning hyperparameters.

| Hyperparameter | Value |
|---|---|
| Time–tilting factor $q$ | 0.9 |
| $\lambda_x$ | varying |
| $\lambda_\alpha$ | varying |
| $\lambda_f$ | varying |
| $K_{\text{last}}$ | 20 |
| $K$ | 20 |
| Noise scaling $\kappa$ | 0.9 |
| Sampling steps (per trajectory) | 100 |
| Training epochs | 20 |
| Learning rate | 0.00002 |
| Batch size | 15 |

### E.4 GUIDANCE

For generating the guidance results, we use the same model as in the Darcy experiments to further highlight that we can infer functional joint distributions when starting from noisy observations. Instead of the usual Euler stepping, here we use a Heun sampler following Huang et al. (2024). While this is more expensive, since we need to differentiate through two forward passes of our model to obtain the guidance gradient, it empirically improved faithfulness to the sparse observations significantly. Note, however, that we only guide on sparse observations and not on PDE residuals. The full sampling procedure is shown in Algorithm 2. In our experiments, we use 100 steps for sampling and guidance scales of $\gamma_x = \gamma_\alpha = 0.75$.

### E.5 ELASTICITY

Fine-tuning in the elasticity experiment is more challenging than the Darcy denoising experiment, which is why we increase the number of fine-tuning epochs to 100. Other parameters are the same as in the Darcy experiments.

We compare our fine-tuning approach with the inference-time correction method ECI presented in Cheng et al. (2024). Our implementation of the sampling correction method is given in Algorithm 3. For ECI, we set $M = 5$ and use 1000 steps when sampling, compared to 100 steps used when sampling from the fine-tuned model. The reported residual heat maps in the main paper show one test function per grid point, where the magnitude indicates local error without aggregating across test functions.

---

**Algorithm 2** Guided Heun sampler with sparse observations

---

**Require:** initial state $x_0$; initial parameter $\alpha_0$; observed targets $\alpha_{\text{obs}}$; index set $\mathcal{I}$; steps $S$; guidance scales $\gamma_x, \gamma_\alpha$; base fields $v_t^x, v_t^\alpha$; fine-tuned joint field $v_t^{\text{ft}}$
**Ensure:** trajectories $\{x^{(i)}\}_{i=0}^S$ and $\{\alpha^{(i)}\}_{i=0}^S$

1: $x^{(0)} \leftarrow x_0$
2: $\alpha^{(0)} \leftarrow \alpha_0$
3: $t_i \leftarrow i/S$ for $i = 0, \ldots, S$
4: $h \leftarrow 1/S$

5: **for** $i = 0$ **to** $S - 1$ **do**
6:     **Predictor (Euler):**
7:     $v^x \leftarrow v_{t_i}^x\big(x^{(i)}\big)$
8:     $v_{\text{base}}^\alpha \leftarrow v_{t_i}^\alpha\big(x^{(i)}, \alpha^{(i)}, v^x\big)$
9:     $(\tilde{v}^x, \tilde{v}^\alpha) \leftarrow v_t^{\text{ft}}\big(x^{(i)}, \alpha^{(i)}, v_{\text{base}}^\alpha\big)$
10:     $\hat{x} \leftarrow x^{(i)} + h\,\tilde{v}^x$
11:     $\hat{\alpha} \leftarrow \alpha^{(i)} + h\,\tilde{v}^\alpha$

12:     **if** $i < S - 1$ **then**
13:         **Corrector (Heun):**
14:         $v_+^x \leftarrow v_{t_{i+1}}^x(\hat{x})$
15:         $v_{\text{base},+}^\alpha \leftarrow v_{t_{i+1}}^\alpha(\hat{x}, \hat{\alpha}, v_+^x)$
16:         $(\tilde{v}_+^x, \tilde{v}_+^\alpha) \leftarrow v_{t+1}^{\text{ft}}(\hat{x}, \hat{\alpha}, v_{\text{base},+}^\alpha)$
17:         **One-step terminal extrapolation:**
18:         $\hat{\alpha}_1 \leftarrow \hat{\alpha} + (1 - t_{i+1})\,\tilde{v}_+^\alpha$
19:         **Sparse-observation loss:**
20:         $L \leftarrow \dfrac{1}{|\mathcal{I}|} \sum_{j \in \mathcal{I}} \big\|\alpha_{\text{obs}}[j] - \hat{\alpha}_1[j]\big\|_2^2$
21:         **Heun average update:**
22:         $x^{(i+1)} \leftarrow x^{(i)} + \dfrac{h}{2}\big(\tilde{v}^x + \tilde{v}_+^x\big)$
23:         $\alpha^{(i+1)} \leftarrow \alpha^{(i)} + \dfrac{h}{2}\big(\tilde{v}^\alpha + \tilde{v}_+^\alpha\big)$
24:         **Decaying guidance:**
25:         $s \leftarrow \sqrt{1 - i/S}$
26:         $x^{(i+1)} \leftarrow x^{(i+1)} - s\,\gamma_x\,\nabla_{x^{(i)}} L$
27:         $\alpha^{(i+1)} \leftarrow \alpha^{(i+1)} - s\,\gamma_\alpha\,\nabla_{\alpha^{(i)}} L$
28:     **else**
29:         **Last step (no correction/guidance):**
30:         $x^{(i+1)} \leftarrow \hat{x}$
31:         $\alpha^{(i+1)} \leftarrow \hat{\alpha}$
32:     **end if**
33: **end for**
34: **return** $\{x^{(i)}\}_{i=0}^S$, $\{\alpha^{(i)}\}_{i=0}^S$

---

### E.6   HELMHOLTZ AND STOKES LID-DRIVEN CAVITY

For both Helmholtz and the Stokes lid-driven cavity problems, we use the same configuration as in the elasticity experiment (see 8), training for 100 epochs. We only adjust the number of input/output channels.

### E.7   NATURAL IMAGES: RECOLORING

For natural-image experiments, we adopt the class-conditional Latent Flow Matching (LFM) model of Dao et al. (2023) trained on ImageNet-1k (Deng et al., 2009) with a DiT backbone (Peebles & Xie, 2023). We fix an ImageNet class label $y$ to condition the generator and hold a global text

Table 8: Elasticity fine-tuning hyperparameters.

| Hyperparameter | Value |
|---|---|
| Time–tilting factor $q$ | 0.9 |
| $\lambda_x$ | varying |
| $\lambda_\alpha$ | varying |
| $\lambda_f$ | varying |
| $K_{\text{last}}$ | 20 |
| $K$ | 20 |
| Noise scaling $\kappa$ | 0.9 |
| Sampling steps (per trajectory) | 100 |
| Training epochs | 100 |
| Learning rate | 0.00002 |
| Batch size | 15 |

---

**Algorithm 3** ECI-style sampling with boundary correction

---

**Require:** initial state $x_0$; steps $S$; inner correction iterations $M$; model drift $v_t(\cdot)$; correction operator $\mathcal{C}(\cdot)$; noise sampler $\text{Noise}(\cdot)$
**Ensure:** trajectory $\{x^{(i)}\}_{i=0}^S$
 1: $x^{(0)} \leftarrow x_0$
 2: $t_i \leftarrow i/S$ for $i = 0, \ldots, S$

 3: **for** $i = 0$ **to** $S - 1$ **do**
 4:     $t \leftarrow t_i$
 5:     $t^{\text{next}} \leftarrow t_{i+1}$

 6:     **Inner ECI corrections at fixed $t$:**
 7:     $\tilde{x} \leftarrow x^{(i)}$
 8:     **for** $j = 1$ **to** $M$ **do**
 9:         $v \leftarrow v_t(\tilde{x})$
10:         $x_{\text{os}} \leftarrow \tilde{x} + (1 - t)\, v$                                    *one-step Euler update*
11:         $x_{\text{corr}} \leftarrow \mathcal{C}(x_{\text{os}})$                              *apply boundary correction*
12:         $\eta \leftarrow \text{Noise}(\text{shape of } x_0)$
13:         $\tilde{x} \leftarrow (1 - t)\, \eta + t\, x_{\text{corr}}$                          *stochastic convex blending*
14:     **end for**

15:     **Final correction and roll-forward to $t^{\text{next}}$:**
16:     $v \leftarrow v_t(\tilde{x})$
17:     $x_{\text{os}} \leftarrow \tilde{x} + (1 - t)\, v$
18:     $x_{\text{corr}} \leftarrow \mathcal{C}(x_{\text{os}})$
19:     $\eta \leftarrow \text{Noise}(\text{shape of } x_0)$
20:     $x^{(i+1)} \leftarrow (1 - t^{\text{next}})\, \eta + t^{\text{next}}\, x_{\text{corr}}$
21: **end for**
22: **return** $\{x^{(i)}\}_{i=0}^S$

---

prompt $c$ to define the fine-tuning direction. As a reward we use *PickScore v1*, a CLIP-H/14–based preference scorer trained on the Pick-a-Pic dataset, that evaluates image–text compatibility via cosine similarity in the CLIP embedding space (Kirstain et al., 2023; Radford et al., 2021; Cherti et al., 2023; Schuhmann et al., 2022), implemented with TRANSFORMERS (Wolf et al., 2020).

Concretely, we maximize $R\Big(T\big(D(z),\ \alpha\big),\ c\Big)$, where D is the latent-space decoder and $T(\cdot, \alpha)$ is a parametric per-pixel recoloring with coefficients $\alpha$. The recoloring operates in logit space to avoid

saturation: with $D(z) = x : \Omega \to (0, 1)^3$,

$$x_\varepsilon(\xi) = \text{clip}\big(x(\xi), \varepsilon, 1 - \varepsilon\big), \qquad \ell(\xi) = \log \frac{x_\varepsilon(\xi)}{1 - x_\varepsilon(\xi)} \in \mathbb{R}^3,$$

we apply a residual $r(\xi; \alpha) = \alpha \, \Phi_D^{\text{bias}}\big(x(\xi)\big)$ built from RGB monomials up to total degree $D$ and map back via $y(\xi; \alpha) = \sigma\big(\ell(\xi) + r(\xi; \alpha)\big)$. Equivalently, for channel $c \in \{R, G, B\}$,

$$y_c(\xi; \alpha) = \sigma\Big(\ell_c(\xi) + \sum_{m=1}^{M} \alpha_{cm} \, \phi_m(x(\xi)) + \alpha_{c,0}\Big), \quad \phi_m \in \Phi_D.$$

This residual parameterization is identity at initialization ($\alpha = 0$) and provides a low-dimensional, channel-coupled appearance pathway that can disentangle *content* from *presentation* or reach colorings underrepresented by the base LFM. It is related to CNN-predicted polynomial color transforms (quadratic in Chai et al. (2020)), whereas we use cubic polynomials and learn solely via the reward.

The parameter predictor $\varphi$ here maps from latent feature tensors $z$ to parameters $\alpha$. First, a compact scene descriptor is extracted by passing $z$ through convolutional transformations, aggregating information across multiple spatial scales via downsampling and global pooling, and enriching it with low-order channel statistics of $z$ (e.g., moments). The concatenated descriptor is projected into a fixed embedding space, refined by a lightweight pre–LayerNorm MLP with a residual connection, and finally mapped by a linear head to the recoloring coefficients $\alpha$.

Building on the inverse predictor, we augment the base DiT backbone with two lightweight residual heads that couple image dynamics and color–parameter evolution. First, the *image path* predicts the base drift $v_{t,x}^{\text{base}}(x, t)$. Then, we form a compact $\alpha$-token by flattening the current color parameters $\alpha$ and mapping them through a small MLP. This token is broadcast to a $k_\alpha$-channel feature map and concatenated with $(x, v_{t,x}^{\text{base}})$ into a shallow U–Net "correction" that outputs an additive refinement, yielding

$$v_{t,x}^{\text{ft}} = v_{t,x}^{\text{base}} + \mathcal{U}_x\big(x, v_{t,x}^{\text{base}}, \text{map}(\alpha), t\big).$$

Second, the *parameter path* updates the polynomial recoloring coefficients by a residual on top of the baseline parameter field $v_{t,\alpha}^{\text{base}}$. Here, a dedicated $\alpha$-projection module mirrors the inverse predictor: multi-scale pooled conv features are fused—at the *token level*—with tokens from $\alpha$ and $v_{t,\alpha}^{\text{base}}$ via a small fusion MLP, AdaLN/FiLM modulation of a head token, and a light SE rescaling of conv features. The head then predicts a delta added to the baseline,

$$v_{t,\alpha}^{\text{ft}} = v_{t,\alpha}^{\text{base}} + \mathcal{U}_\alpha\big(x, \alpha, v_{t,\alpha}^{\text{base}}\big).$$

All correction heads are zero-initialized at their final projections, so fine-tuning starts from the base behavior and departs smoothly as training progresses. Table 9 lists the parameters of the correction U–Nets. In total, the modified architecture adds $\approx$9M parameters to the $\approx$130M parameters of the base DiT backbone.

Table 9: Correction head hyperparameters (image and parameter paths).

| Hyperparameter | Value |
|---|---|
| $k_\alpha$ | 16 |
| U–Net base widths | [96, 128, 160, 192] |
| U–Net embedding | 96 |
| U–Net hidden lift | 256 |
| Attention stages | [2, 3] |
| Attention heads | 8 |

With the LFM model, we use 40 steps when sampling and use the same training specifications as the original Adjoint Matching paper (Domingo-Enrich et al., 2025). Again, we first pretrain the predictor $\varphi$, and then perform joint fine-tuning with our Adjoint Matching formulation. Fine-tuning is performed for 100 epochs with a batch size of 15.

# F ADDITIONAL RESULTS

In this section, we provide full experimental results and samples from the trained models.

## F.1 FULL EXPERIMENTAL RESULTS

### F.1.1 DARCY

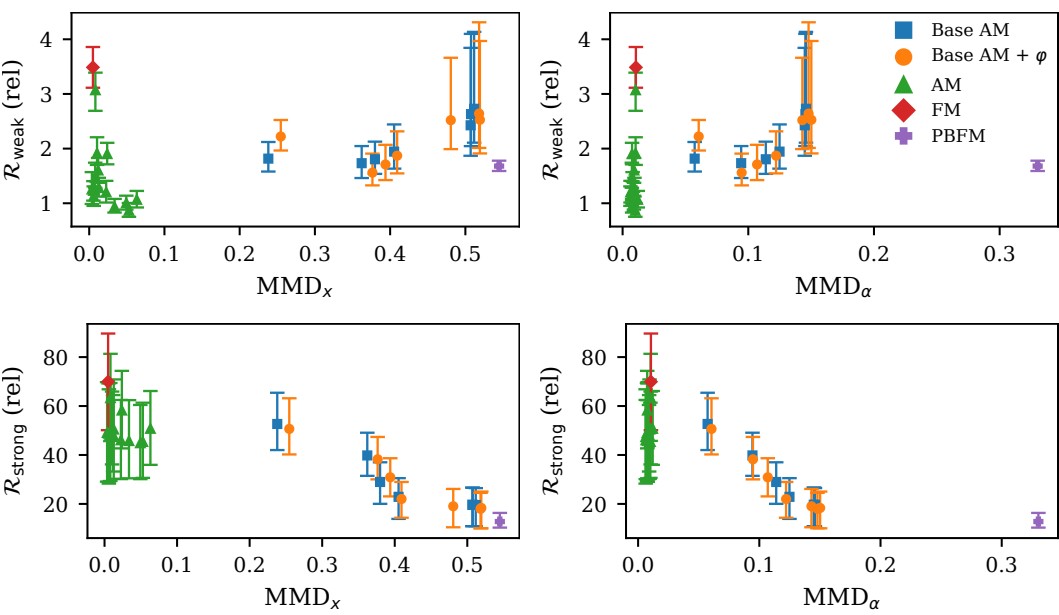

Figure 7: Darcy dataset: scatter plots showing the relationship between PDE residuals (weak and strong) and distributional discrepancies ($\text{MMD}_x$, $\text{MMD}_\alpha$) across all model variants and hyperparameter configurations. Each point corresponds to one configuration; lower values indicate better physics consistency or distributional fidelity.

Table 10: Darcy dataset: full residual and distributional metrics for all methods and hyperparameter settings

| Model | $\lambda_x$ | $\lambda_f$ | $R_{\text{weak}} \downarrow$ | $R_{\text{strong}} \downarrow$ | $\text{MMD}_x \downarrow$ | $\text{MMD}_\alpha \downarrow$ |
|---|---|---|---|---|---|---|
| FM | – | – | $3.56 \times 10^0 \ (\pm 0.62)$ | $7.11 \times 10^1 \ (\pm 3.20)$ | 0.005 | 0.011 |
| PBFM | – | – | $1.69 \times 10^0 \ (\pm 0.14)$ | $1.64 \times 10^1 \ (\pm 1.66)$ | 0.545 | 0.330 |
| Base AM | 1k | – | $1.89 \times 10^0 \ (\pm 0.40)$ | $5.55 \times 10^1 \ (\pm 2.04)$ | 0.238 | 0.057 |
| Base AM | 5k | – | $1.84 \times 10^0 \ (\pm 0.55)$ | $4.22 \times 10^1 \ (\pm 1.63)$ | 0.362 | 0.094 |
| Base AM | 10k | – | $1.95 \times 10^0 \ (\pm 0.81)$ | $2.95 \times 10^1 \ (\pm 1.14)$ | 0.380 | 0.114 |
| Base AM | 20k | – | $2.26 \times 10^0 \ (\pm 1.38)$ | $2.34 \times 10^1 \ (\pm 1.10)$ | 0.405 | 0.125 |
| Base AM | 100k | – | $3.44 \times 10^0 \ (\pm 2.98)$ | $2.04 \times 10^1 \ (\pm 1.05)$ | 0.507 | 0.145 |
| Base AM | 1M | – | $3.68 \times 10^0 \ (\pm 3.17)$ | $2.03 \times 10^1 \ (\pm 1.04)$ | 0.508 | 0.145 |
| Base AM | 100M | – | $3.72 \times 10^0 \ (\pm 3.05)$ | $2.02 \times 10^1 \ (\pm 1.03)$ | 0.512 | 0.146 |
| Base AM + $\varphi$ | 1k | – | $2.29 \times 10^0 \ (\pm 0.42)$ | $5.32 \times 10^1 \ (\pm 1.89)$ | 0.255 | 0.061 |
| Base AM + $\varphi$ | 5k | – | $1.68 \times 10^0 \ (\pm 0.51)$ | $4.01 \times 10^1 \ (\pm 1.46)$ | 0.376 | 0.095 |
| Base AM + $\varphi$ | 10k | – | $1.85 \times 10^0 \ (\pm 0.66)$ | $3.2 \times 10^1 \ (\pm 1.16)$ | 0.394 | 0.107 |
| Base AM + $\varphi$ | 20k | – | $2.15 \times 10^0 \ (\pm 1.18)$ | $2.26 \times 10^1 \ (\pm 1.00)$ | 0.409 | 0.122 |
| Base AM + $\varphi$ | 100k | – | $3.36 \times 10^0 \ (\pm 2.61)$ | $1.98 \times 10^1 \ (\pm 1.03)$ | 0.481 | 0.143 |
| Base AM + $\varphi$ | 1M | – | $3.82 \times 10^0 \ (\pm 3.27)$ | $1.87 \times 10^1 \ (\pm 0.99)$ | 0.518 | 0.148 |
| Base AM + $\varphi$ | 100M | – | $3.58 \times 10^0 \ (\pm 3.06)$ | $1.9 \times 10^1 \ (\pm 1.01)$ | 0.519 | 0.150 |

Table 10 – *continued from previous page*

| Model | $\lambda_x$ | $\lambda_f$ | $R_{\text{weak}} \downarrow$ | $R_{\text{strong}} \downarrow$ | $\text{MMD}_x \downarrow$ | $\text{MMD}_\alpha \downarrow$ |
|---|---|---|---|---|---|---|
| AM | 1k | 0.0 | $1.93 \times 10^0$ ($\pm$ 0.32) | $5.96 \times 10^1$ ($\pm$ 2.60) | 0.024 | 0.008 |
| AM | 10k | 0.0 | $1.11 \times 10^0$ ($\pm$ 0.27) | $5.23 \times 10^1$ ($\pm$ 2.18) | 0.063 | 0.012 |
| AM | 20k | 0.0 | $9.15 \times 10^{-1}$ ($\pm$ 2.29) | $4.71 \times 10^1$ ($\pm$ 2.13) | 0.053 | 0.010 |
| AM | 20k | 10 | $1.97 \times 10^0$ ($\pm$ 0.38) | $5.13 \times 10^1$ ($\pm$ 2.15) | 0.010 | 0.010 |
| AM | 20k | 1 | $1.64 \times 10^0$ ($\pm$ 0.34) | $4.94 \times 10^1$ ($\pm$ 2.20) | 0.012 | 0.009 |
| AM | 20k | $10^{-1}$ | $1.23 \times 10^0$ ($\pm$ 0.27) | $4.73 \times 10^1$ ($\pm$ 2.21) | 0.023 | 0.006 |
| AM | 20k | $10^{-2}$ | $1.05 \times 10^0$ ($\pm$ 0.24) | $4.63 \times 10^1$ ($\pm$ 2.08) | 0.049 | 0.009 |
| AM | 20k | $10^{-5}$ | $9.78 \times 10^{-1}$ ($\pm$ 2.18) | $4.74 \times 10^1$ ($\pm$ 2.19) | 0.034 | 0.008 |
| AM | 100k | 0.0 | $1.15 \times 10^0$ ($\pm$ 0.28) | $4.9 \times 10^1$ ($\pm$ 2.58) | 0.006 | 0.006 |
| AM | 1M | 0.0 | $1.26 \times 10^0$ ($\pm$ 0.32) | $5.08 \times 10^1$ ($\pm$ 2.72) | 0.006 | 0.008 |
| AM | 10M | 0.0 | $1.54 \times 10^0$ ($\pm$ 0.35) | $5.09 \times 10^1$ ($\pm$ 2.69) | 0.008 | 0.007 |
| AM | 100M | 0.0 | $1.32 \times 10^0$ ($\pm$ 0.31) | $5.23 \times 10^1$ ($\pm$ 2.77) | 0.013 | 0.009 |

### F.1.2 ELASTICITY

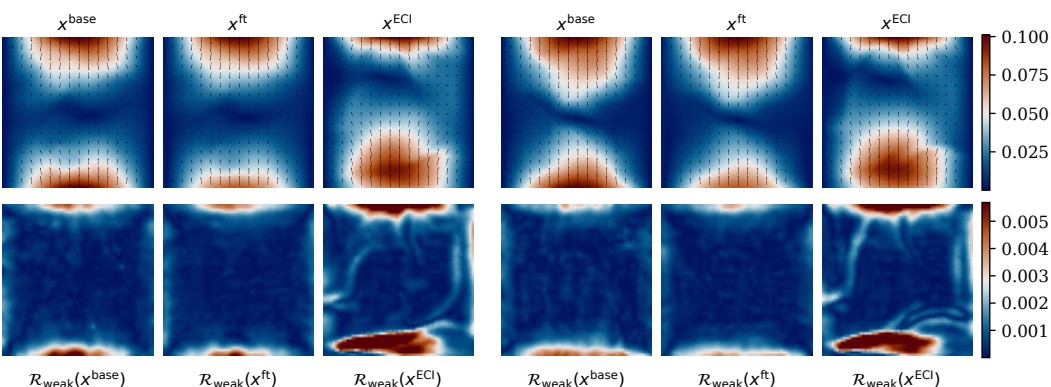

Figure 8: Fine-tuning towards boundary modification, comparing our approach with ECI. Samples for two random seeds shared across models. Top row: displacement fields. Bottom row: corresponding weak residual heatmaps.

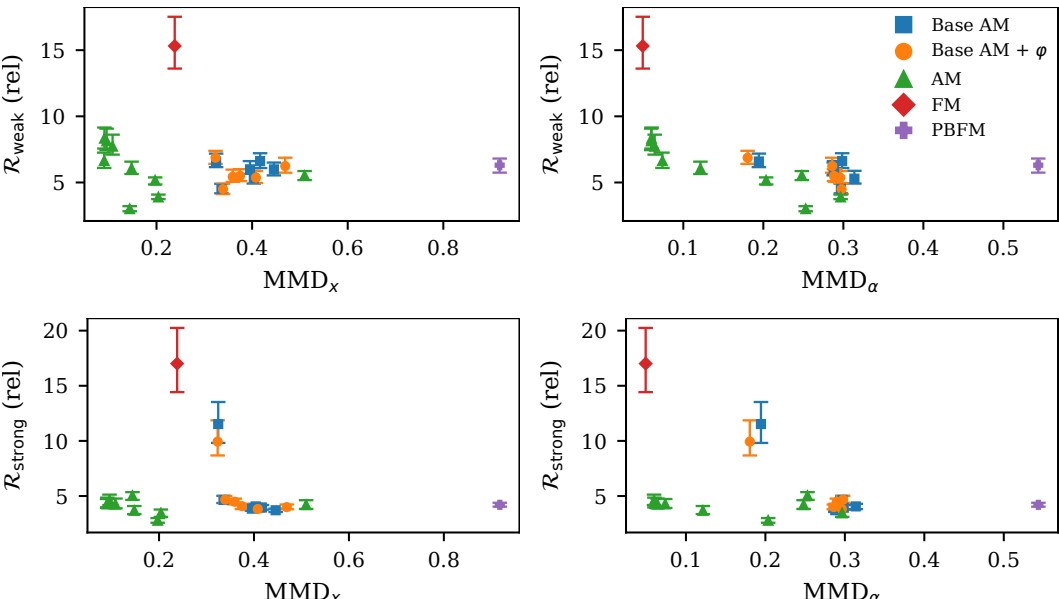

Figure 9: Elasticity dataset: scatter plots showing the relationship between PDE residuals (weak and strong) and distributional discrepancies ($\text{MMD}_x$, $\text{MMD}_\alpha$) across all model variants and hyperparameter configurations. Each point corresponds to one configuration; lower values indicate better physics consistency or distributional fidelity.

Table 11: Elasticity dataset: full residual and distributional metrics for all methods and hyperparameter settings

| Model | $\lambda_x$ | $\lambda_f$ | $R_{\text{weak}} \downarrow$ | $R_{\text{strong}} \downarrow$ | $\text{MMD}_x \downarrow$ | $\text{MMD}_\alpha \downarrow$ |
|---|---|---|---|---|---|---|
| FM | – | – | $1.59 \times 10^1$ ($\pm$ 0.37) | $1.83 \times 10^1$ ($\pm$ 0.66) | 0.238 | 0.049 |
| PBFM | – | – | $6.32 \times 10^0$ ($\pm$ 0.82) | $4.22 \times 10^0$ ($\pm$ 0.26) | 0.918 | 0.544 |
| Base AM | 1k | – | $6.78 \times 10^0$ ($\pm$ 1.09) | $1.21 \times 10^1$ ($\pm$ 0.34) | 0.325 | 0.194 |
| Base AM | 10k | – | $4.62 \times 10^0$ ($\pm$ 0.85) | $4.89 \times 10^0$ ($\pm$ 1.29) | 0.336 | 0.297 |

*continued on next page*

Table 11 – *continued from previous page*

| Model | $\lambda_x$ | $\lambda_f$ | $R_{\text{weak}} \downarrow$ | $R_{\text{strong}} \downarrow$ | $\text{MMD}_x \downarrow$ | $\text{MMD}_\alpha \downarrow$ |
|---|---|---|---|---|---|---|
| Base AM | 50k | – | $5.47 \times 10^0$ ($\pm 0.85$) | $4.16 \times 10^0$ ($\pm 0.87$) | 0.404 | 0.314 |
| Base AM | 100k | – | $6.18 \times 10^0$ ($\pm 0.75$) | $4.0 \times 10^0$ ($\pm 0.98$) | 0.396 | 0.288 |
| Base AM | 500k | – | $6.68 \times 10^0$ ($\pm 0.79$) | $4.04 \times 10^0$ ($\pm 0.76$) | 0.417 | 0.299 |
| Base AM | 1M | – | $6.08 \times 10^0$ ($\pm 0.74$) | $3.76 \times 10^0$ ($\pm 0.54$) | 0.446 | 0.288 |
| Base AM + $\varphi$ | 1k | – | $6.99 \times 10^0$ ($\pm 0.99$) | $1.06 \times 10^1$ ($\pm 0.30$) | 0.323 | 0.180 |
| Base AM + $\varphi$ | 10k | – | $4.58 \times 10^0$ ($\pm 0.70$) | $4.91 \times 10^0$ ($\pm 1.31$) | 0.339 | 0.298 |
| Base AM + $\varphi$ | 50k | – | $5.55 \times 10^0$ ($\pm 0.72$) | $4.58 \times 10^0$ ($\pm 0.94$) | 0.359 | 0.290 |
| Base AM + $\varphi$ | 100k | – | $5.65 \times 10^0$ ($\pm 0.82$) | $4.16 \times 10^0$ ($\pm 0.93$) | 0.375 | 0.288 |
| Base AM + $\varphi$ | 500k | – | $5.47 \times 10^0$ ($\pm 0.72$) | $3.92 \times 10^0$ ($\pm 0.77$) | 0.408 | 0.296 |
| Base AM + $\varphi$ | 1M | – | $6.39 \times 10^0$ ($\pm 0.86$) | $4.09 \times 10^0$ ($\pm 0.68$) | 0.469 | 0.286 |
| AM | 5k | 1 | $5.57 \times 10^0$ ($\pm 0.60$) | $4.34 \times 10^0$ ($\pm 0.95$) | 0.510 | 0.248 |
| AM | 20k | 1k | $8.33 \times 10^0$ ($\pm 1.34$) | $4.78 \times 10^0$ ($\pm 1.15$) | 0.096 | 0.060 |
| AM | 30k | 1k | $8.44 \times 10^0$ ($\pm 1.27$) | $4.48 \times 10^0$ ($\pm 1.08$) | 0.091 | 0.060 |
| AM | 50k | 1k | $7.85 \times 10^0$ ($\pm 1.17$) | $4.4 \times 10^0$ ($\pm 1.08$) | 0.108 | 0.064 |
| AM | 100k | 0.0 | $3.04 \times 10^0$ ($\pm 0.31$) | $5.06 \times 10^0$ ($\pm 0.66$) | 0.144 | 0.253 |
| AM | 100k | 1k | $6.78 \times 10^0$ ($\pm 1.00$) | $4.43 \times 10^0$ ($\pm 1.02$) | 0.091 | 0.074 |
| AM | 500k | 1k | $6.15 \times 10^0$ ($\pm 0.77$) | $3.79 \times 10^0$ ($\pm 0.87$) | 0.148 | 0.121 |
| AM | 1M | 0.0 | $3.93 \times 10^0$ ($\pm 0.27$) | $3.49 \times 10^0$ ($\pm 0.48$) | 0.204 | 0.297 |
| AM | 1M | 1k | $5.14 \times 10^0$ ($\pm 0.42$) | $2.85 \times 10^0$ ($\pm 0.58$) | 0.197 | 0.203 |

### F.1.3 HELMHOLTZ

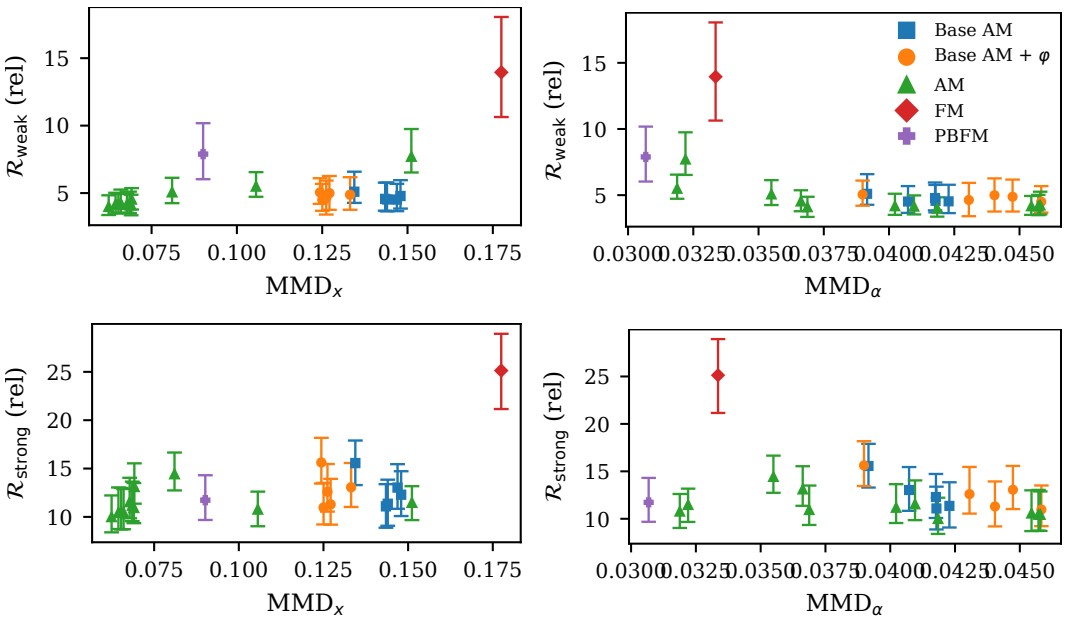

Figure 10: Helmholtz dataset: scatter plots showing the relationship between PDE residuals (weak and strong) and distributional discrepancies ($\mathrm{MMD}_x$, $\mathrm{MMD}_\alpha$) across all model variants and hyperparameter configurations. Each point corresponds to one configuration; lower values indicate better physics consistency or distributional fidelity.

Table 12: Helmholtz dataset: full residual and distributional metrics for all methods and hyperparameter settings

| Model | $\lambda_x$ | $\lambda_f$ | $R_{\text{weak}} \downarrow$ | $R_{\text{strong}} \downarrow$ | $\mathrm{MMD}_x \downarrow$ | $\mathrm{MMD}_\alpha \downarrow$ |
|---|---|---|---|---|---|---|
| FM | – | – | $1.5 \times 10^1$ ($\pm 0.59$) | $2.55 \times 10^1$ ($\pm 0.55$) | 0.177 | 0.033 |
| PBFM | – | – | $8.33 \times 10^0$ ($\pm 3.04$) | $1.22 \times 10^1$ ($\pm 0.33$) | 0.090 | 0.031 |
| Base AM | 5k | – | $5.64 \times 10^0$ ($\pm 2.09$) | $1.59 \times 10^1$ ($\pm 0.33$) | 0.134 | 0.039 |
| Base AM | 50k | – | $4.9 \times 10^0$ ($\pm 1.85$) | $1.34 \times 10^1$ ($\pm 0.32$) | 0.147 | 0.041 |
| Base AM | 100k | – | $5.19 \times 10^0$ ($\pm 2.01$) | $1.27 \times 10^1$ ($\pm 0.32$) | 0.148 | 0.042 |
| Base AM | 1M | – | $4.95 \times 10^0$ ($\pm 1.81$) | $1.17 \times 10^1$ ($\pm 0.33$) | 0.144 | 0.042 |
| Base AM | 100M | – | $5.01 \times 10^0$ ($\pm 2.00$) | $1.14 \times 10^1$ ($\pm 0.32$) | 0.143 | 0.042 |
| Base AM + $\varphi$ | 5k | – | $5.46 \times 10^0$ ($\pm 1.94$) | $1.59 \times 10^1$ ($\pm 0.33$) | 0.124 | 0.039 |
| Base AM + $\varphi$ | 50k | – | $5.35 \times 10^0$ ($\pm 2.31$) | $1.35 \times 10^1$ ($\pm 0.34$) | 0.133 | 0.045 |
| Base AM + $\varphi$ | 100k | – | $5.02 \times 10^0$ ($\pm 2.17$) | $1.32 \times 10^1$ ($\pm 0.33$) | 0.126 | 0.043 |
| Base AM + $\varphi$ | 1M | – | $4.99 \times 10^0$ ($\pm 2.12$) | $1.16 \times 10^1$ ($\pm 0.33$) | 0.125 | 0.046 |
| Base AM + $\varphi$ | 100M | – | $5.41 \times 10^0$ ($\pm 2.20$) | $1.17 \times 10^1$ ($\pm 0.34$) | 0.127 | 0.044 |
| AM | 5k | 1 | $5.32 \times 10^0$ ($\pm 1.49$) | $1.48 \times 10^1$ ($\pm 0.29$) | 0.081 | 0.035 |
| AM | 10k | 0.0 | $4.71 \times 10^0$ ($\pm 1.26$) | $1.35 \times 10^1$ ($\pm 0.28$) | 0.069 | 0.037 |
| AM | 100k | 0.0 | $4.3 \times 10^0$ ($\pm 1.29$) | $1.14 \times 10^1$ ($\pm 0.29$) | 0.069 | 0.037 |
| AM | 100k | 0p1 | $4.49 \times 10^0$ ($\pm 1.46$) | $1.17 \times 10^1$ ($\pm 0.29$) | 0.069 | 0.040 |
| AM | 100k | 1 | $4.42 \times 10^0$ ($\pm 1.35$) | $1.21 \times 10^1$ ($\pm 0.30$) | 0.068 | 0.041 |
| AM | 100k | 1k | $8.36 \times 10^0$ ($\pm 2.65$) | $1.16 \times 10^1$ ($\pm 0.25$) | 0.151 | 0.032 |
| AM | 500k | 0.0 | $4.56 \times 10^0$ ($\pm 1.47$) | $1.09 \times 10^1$ ($\pm 0.31$) | 0.066 | 0.046 |
| AM | 1M | 0.0 | $4.43 \times 10^0$ ($\pm 1.43$) | $1.1 \times 10^1$ ($\pm 0.30$) | 0.064 | 0.046 |
| AM | 1M | 1k | $5.83 \times 10^0$ ($\pm 1.63$) | $1.1 \times 10^1$ ($\pm 0.27$) | 0.106 | 0.032 |
| AM | 100M | 0.0 | $4.43 \times 10^0$ ($\pm 1.43$) | $1.1 \times 10^1$ ($\pm 0.30$) | 0.065 | 0.045 |

Table 12 – *continued from previous page*

| Model | $\lambda_x$ | $\lambda_f$ | $R_{\text{weak}} \downarrow$ | $R_{\text{strong}} \downarrow$ | $\text{MMD}_x \downarrow$ | $\text{MMD}_\alpha \downarrow$ |
|---|---|---|---|---|---|---|
| AM | 100M | 1 | $4.32 \times 10^0 \ (\pm 1.43)$ | $1.05 \times 10^1 \ (\pm 0.30)$ | 0.062 | 0.042 |

## F.2 STOKES LID-DRIVEN CAVITY

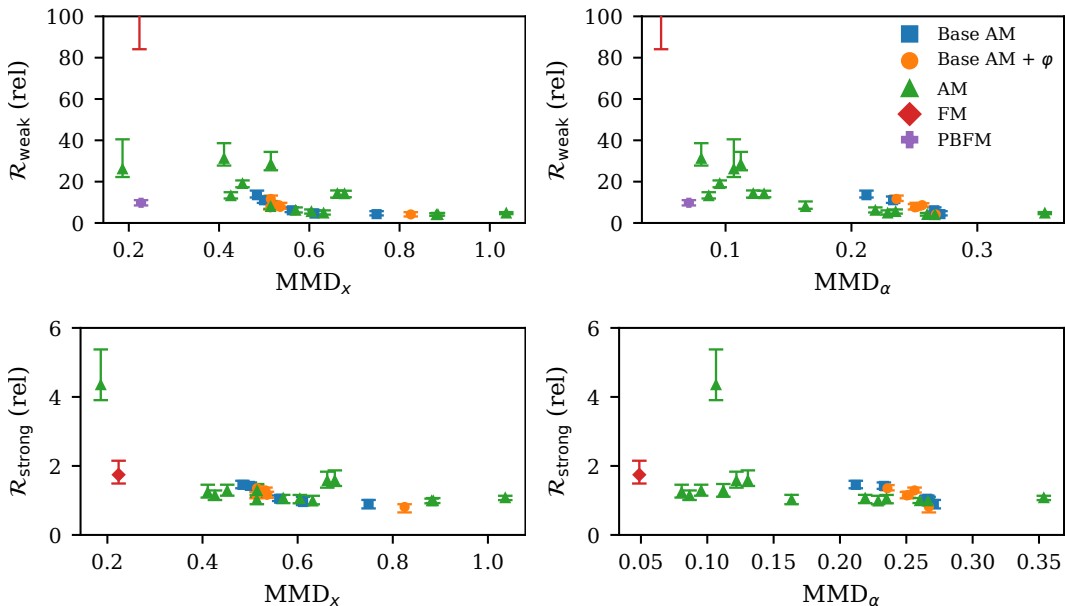

Figure 11: Stokes dataset: scatter plots showing the relationship between PDE residuals (weak and strong) and distributional discrepancies ($\text{MMD}_x$, $\text{MMD}_\alpha$) across all model variants and hyperparameter configurations. Each point corresponds to a single configuration; lower values indicate better physics consistency or distributional fidelity. PBFM is omitted in the strong-residual panels because its training did not converge to physically meaningful samples (yielding strong residuals several orders of magnitude larger than the other methods.

Table 13: Stokes dataset: full residual and distributional metrics for all methods and hyperparameter settings.

| Model | $\lambda_x$ | $\lambda_f$ | $R_{\text{weak}}\downarrow$ | $R_{\text{strong}}\downarrow$ | $\text{MMD}_x\downarrow$ | $\text{MMD}_\alpha\downarrow$ |
|---|---|---|---|---|---|---|
| FM | – | – | $3.05\times10^2$ ($\pm$ 3.16) | $1.81\times10^0$ ($\pm$ 0.44) | 0.224 | 0.049 |
| PBFM | – | – | $9.94\times10^0$ ($\pm$ 2.17) | $1.15\times10^1$ ($\pm$ 0.05) | 0.227 | 0.071 |
| Base AM | 5k | – | $1.42\times10^1$ ($\pm$ 0.28) | $1.47\times10^0$ ($\pm$ 0.17) | 0.484 | 0.212 |
| Base AM | 10k | – | $1.16\times10^1$ ($\pm$ 0.27) | $1.43\times10^0$ ($\pm$ 0.13) | 0.500 | 0.233 |
| Base AM | 100k | – | $6.88\times10^0$ ($\pm$ 2.47) | $1.07\times10^0$ ($\pm$ 0.14) | 0.561 | 0.266 |
| Base AM | 1M | – | $5.3\times10^0$ ($\pm$ 2.23) | $9.94\times10^{-1}$ ($\pm$ 1.59) | 0.611 | 0.267 |
| Base AM | 100M | – | $5.05\times10^0$ ($\pm$ 2.16) | $9.04\times10^{-1}$ ($\pm$ 1.77) | 0.749 | 0.271 |
| Base AM + $\varphi$ | 5k | – | $1.22\times10^1$ ($\pm$ 0.22) | $1.37\times10^0$ ($\pm$ 0.12) | 0.515 | 0.236 |
| Base AM + $\varphi$ | 10k | – | $9.0\times10^0$ ($\pm$ 1.55) | $1.3\times10^0$ ($\pm$ 0.12) | 0.532 | 0.256 |
| Base AM + $\varphi$ | 100k | – | $8.11\times10^0$ ($\pm$ 2.28) | $1.15\times10^0$ ($\pm$ 0.13) | 0.514 | 0.250 |
| Base AM + $\varphi$ | 1M | – | $8.55\times10^0$ ($\pm$ 2.52) | $1.16\times10^0$ ($\pm$ 0.13) | 0.535 | 0.252 |
| Base AM + $\varphi$ | 100M | – | $4.64\times10^0$ ($\pm$ 2.04) | $7.89\times10^{-1}$ ($\pm$ 1.59) | 0.825 | 0.267 |
| AM | 10k | 0.0 | $1.41\times10^1$ ($\pm$ 0.93) | $1.16\times10^0$ ($\pm$ 0.20) | 0.426 | 0.087 |
| AM | 10k | 1 | $1.98\times10^1$ ($\pm$ 0.77) | $1.3\times10^0$ ($\pm$ 0.24) | 0.452 | 0.095 |
| AM | 50k | 0.0 | $9.14\times10^0$ ($\pm$ 4.80) | $1.05\times10^0$ ($\pm$ 0.20) | 0.514 | 0.164 |
| AM | 100k | 0.0 | $7.0\times10^0$ ($\pm$ 3.51) | $1.07\times10^0$ ($\pm$ 0.20) | 0.570 | 0.219 |
| AM | 100k | 1k | $3.43\times10^1$ ($\pm$ 1.80) | $4.87\times10^0$ ($\pm$ 1.29) | 0.186 | 0.107 |
| AM | 500k | 0.0 | $6.26\times10^0$ ($\pm$ 3.15) | $1.07\times10^0$ ($\pm$ 0.21) | 0.605 | 0.235 |
| AM | 1M | 0.0 | $5.8\times10^0$ ($\pm$ 3.28) | $1.03\times10^0$ ($\pm$ 0.20) | 0.632 | 0.229 |
| AM | 1M | 1k | $4.11\times10^1$ ($\pm$ 2.73) | $1.3\times10^0$ ($\pm$ 0.30) | 0.411 | 0.081 |
| AM | 100M | 0.0 | $4.72\times10^0$ ($\pm$ 2.84) | $1.01\times10^0$ ($\pm$ 0.15) | 0.882 | 0.266 |

Table 13 – *continued from previous page*

| Model | $\lambda_x$ | $\lambda_f$ | $R_{\text{weak}} \downarrow$ | $R_{\text{strong}} \downarrow$ | $\text{MMD}_x \downarrow$ | $\text{MMD}_\alpha \downarrow$ |
|-------|-------------|-------------|------------------------------|--------------------------------|---------------------------|--------------------------------|
| AM | 100M | 100k | $3.67 \times 10^1$ ($\pm 2.47$) | $1.35 \times 10^0$ ($\pm 0.34$) | 0.515 | 0.112 |
| AM | 100M | 10k | $1.45 \times 10^1$ ($\pm 0.34$) | $1.64 \times 10^0$ ($\pm 0.31$) | 0.678 | 0.131 |
| AM | 100M | 1 | $4.9 \times 10^0$ ($\pm 3.43$) | $1.02 \times 10^0$ ($\pm 0.16$) | 0.885 | 0.260 |
| AM | 100M | 1k | $5.27 \times 10^0$ ($\pm 2.36$) | $1.07 \times 10^0$ ($\pm 0.12$) | 1.036 | 0.354 |

### F.2.1 EFFECT OF FORCING MISSPECIFICATION

In the Stokes lid–driven cavity experiment, the base Flow Matching (FM) model is trained on data generated with a nonzero Kolmogorov-type forcing term with amplitude $F_0 = 2.0$. In the main experiment reported in the paper, however, fine-tuning is performed under the *assumption of no forcing* ($F_0 = 0.0$). This represents a severe form of model–physics misspecification: the fine-tuning objective assumes a qualitatively different flow regime from the data on which the FM model was trained.

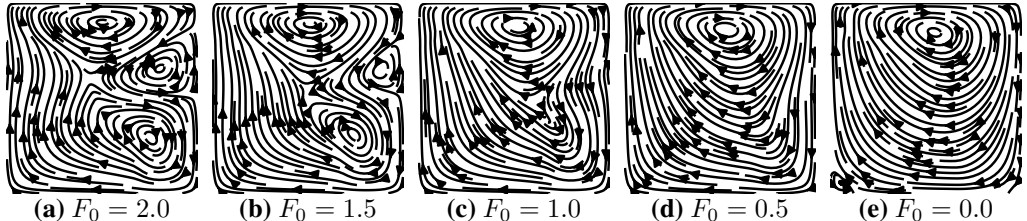

**(a)** $F_0 = 2.0$     **(b)** $F_0 = 1.5$     **(c)** $F_0 = 1.0$     **(d)** $F_0 = 0.5$     **(e)** $F_0 = 0.0$

Figure 12: Representative stationary velocity fields for the same viscosity but different forcing amplitudes $F_0$. With $F_0 = 0$, the flow is driven solely by the moving lid and exhibits a single dominant recirculating vortex. As the forcing strength increases, additional sub-vortices emerge and the flow develops progressively more complex interior structures.

Figure 12 illustrates this mismatch. Samples with $F_0 = 2.0$ exhibit complex flow structure with multiple vortices driven by the interior forcing. In contrast, the flow for $F_0 = 0.0$ contains essentially a single large recirculating vortex induced only by the moving lid. These differences reflect a fundamental change in the underlying PDE solution manifold rather than a minor perturbation. Fine-tuning the FM model to match such a drastically different flow regime therefore constitutes an inherently challenging, strongly out-of-distribution adaptation problem.

To study this effect more systematically, we perform additional fine-tuning runs using different assumed forcing amplitudes $\tilde{F}_0 \in \{0.0, 0.5, 1.0, 1.5, 2.0\}$, while always starting from the same FM model trained on data with $F_0 = 2.0$. All other hyperparameters are kept fixed ($\lambda_x = \lambda_\alpha = 100k$, $\lambda_f = 0.0$). For each $\tilde{F}_0$, we compute the weak PDE residuals after fine-tuning and report them relative to the mean weak residual of *ground-truth* data generated with the same forcing amplitude $\tilde{F}_0$. Because both pre-training of the inverse predictor and the fine-tuning itself use the same assumed forcing $\tilde{F}_0$, we can also report baseline values for the FM model by applying the corresponding inverse predictor to its samples. Figure 13 presents the resulting residuals.

Across all levels of assumed forcing, fine-tuning consistently reduces the weak residuals by at least an order of magnitude compared to the corresponding FM baseline. As expected, the remaining residuals strongly depend on the degree of misspecification: when $\tilde{F}_0$ is close to the true value (2.0), the relative residuals after tuning approach 1.0, indicating consistency with the ground-truth distribution. Under severe mismatch (e.g. $\tilde{F}_0 = 0.0$), the residuals remain higher despite significant improvement relative to the FM baseline, which is consistent with the demand for structurally significant modification.

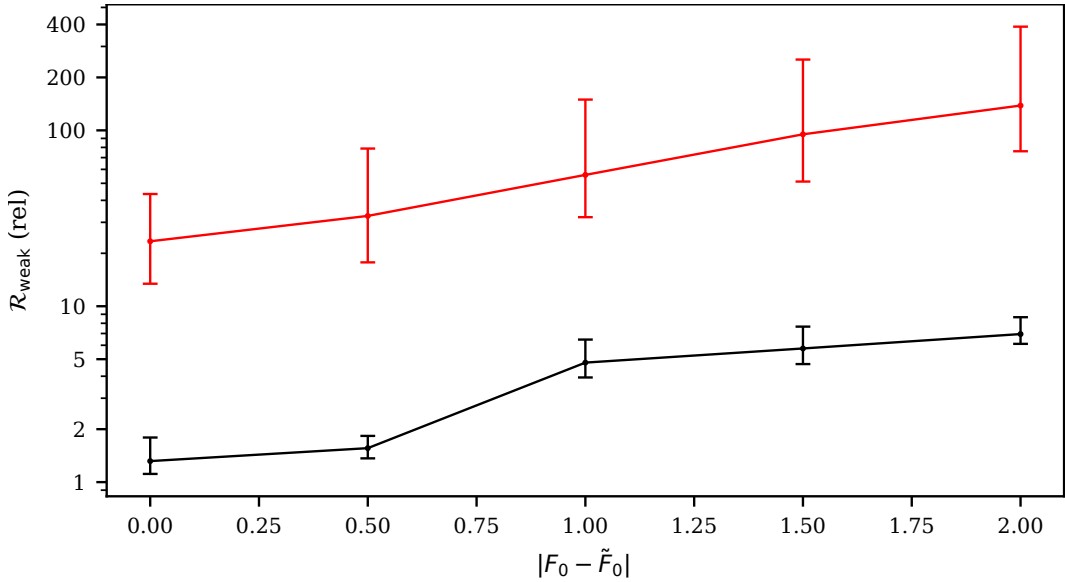

Figure 13: Weak residuals of the base FM model (red) and the fine-tuned model (black) as a function of the assumed forcing amplitude. Reduced misspecification leads to substantially lower post-tuning residuals, approaching the ground-truth residual level when the assumed forcing matches the true physical value.

### F.3 NON-CURATED SAMPLES

### F.3.1 DARCY

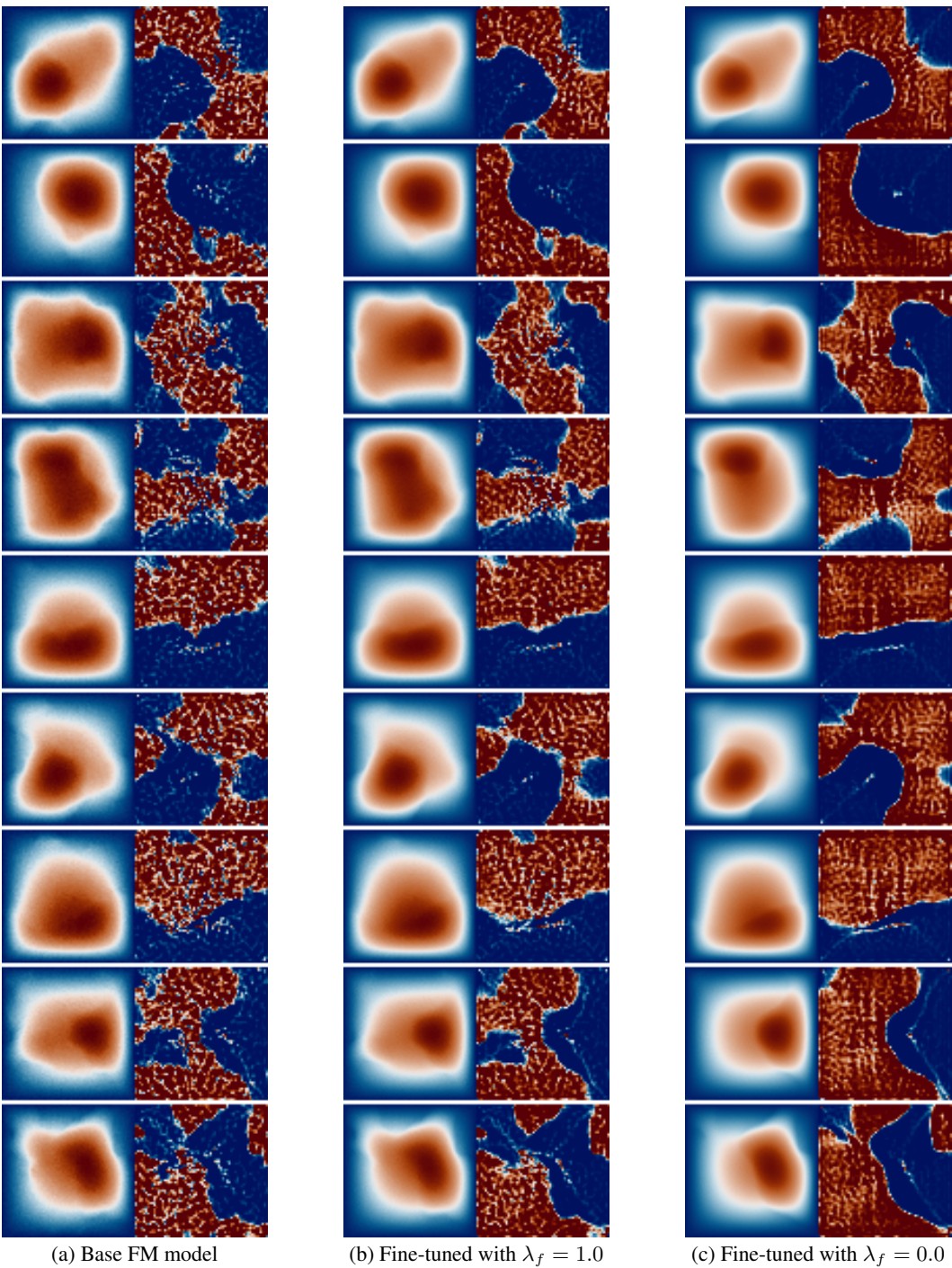

(a) Base FM model    (b) Fine-tuned with $\lambda_f = 1.0$    (c) Fine-tuned with $\lambda_f = 0.0$

Figure 14: Darcy flow: non-curated samples of pressure distributions (left columns) and recovered permeability fields (right columns). Each row was generated using the same initial noise across the three models. Color scales are normalized per row for the pressure distributions. For the base model, permeabilities are obtained with the pre-trained inverse predictor.

### F.3.2 ELASTICITY

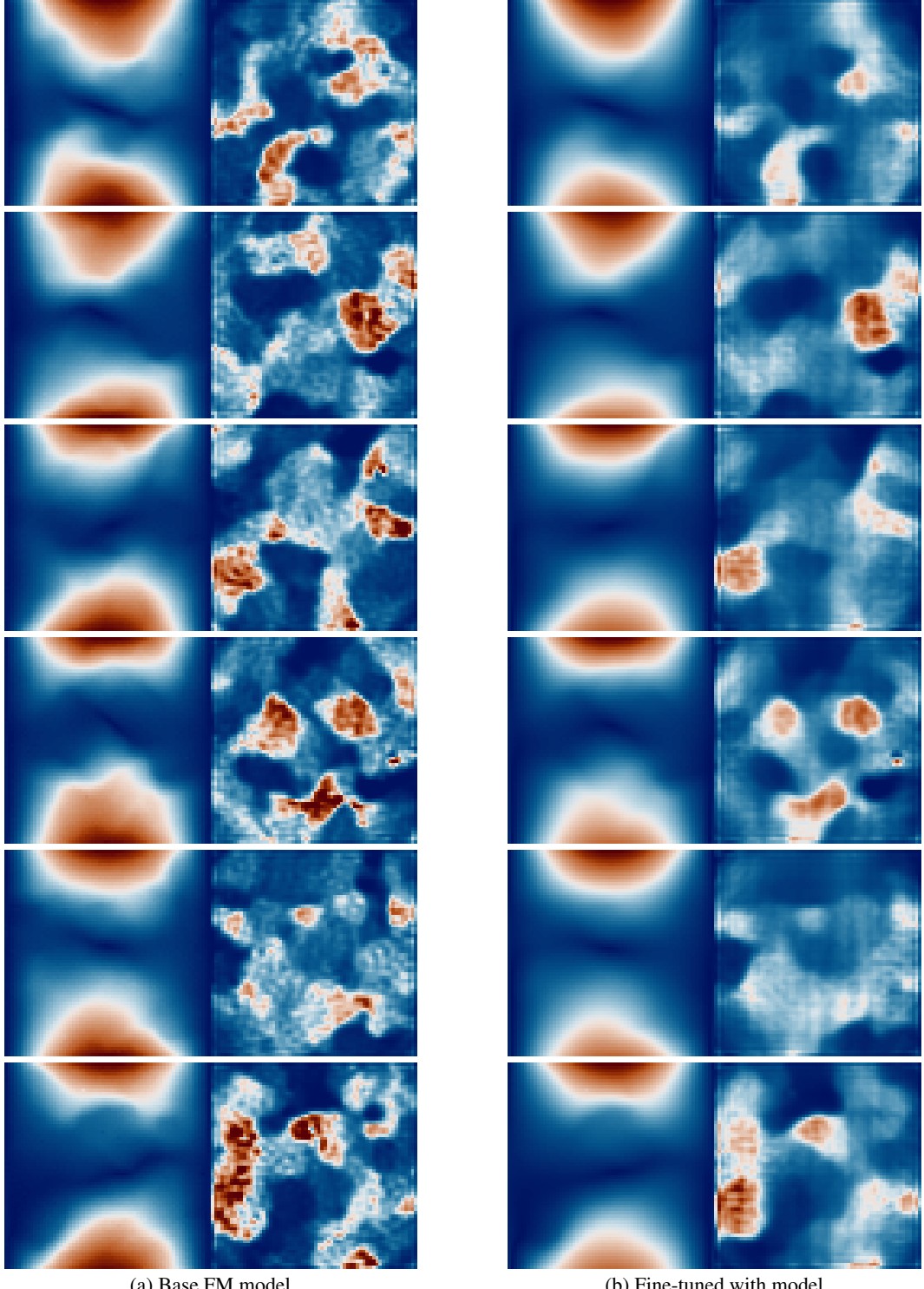

(a) Base FM model  (b) Fine-tuned with model

Figure 15: Non-curated samples from the elasticity experiment, where fine-tuning has to scale down the lower boundary.

### F.3.3   HELMHOLTZ

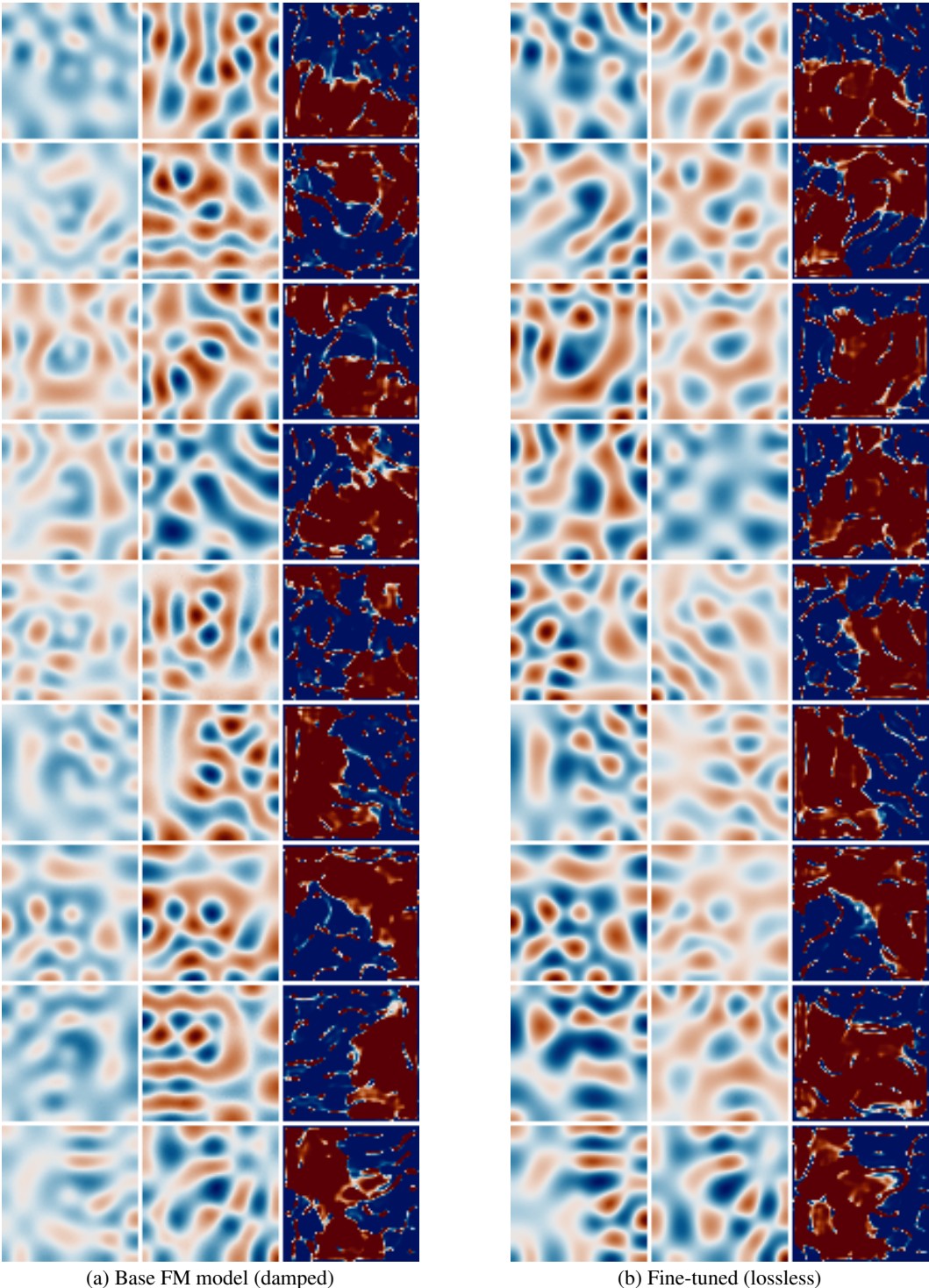

(a) Base FM model (damped)    (b) Fine-tuned (lossless)

Figure 16: Non-curated Helmholtz samples. In both panels, each row shows: (i) the real part of the generated field, (ii) its imaginary part, and (iii) the recovered wave-speed parameter $c$.

### F.3.4 STOKES

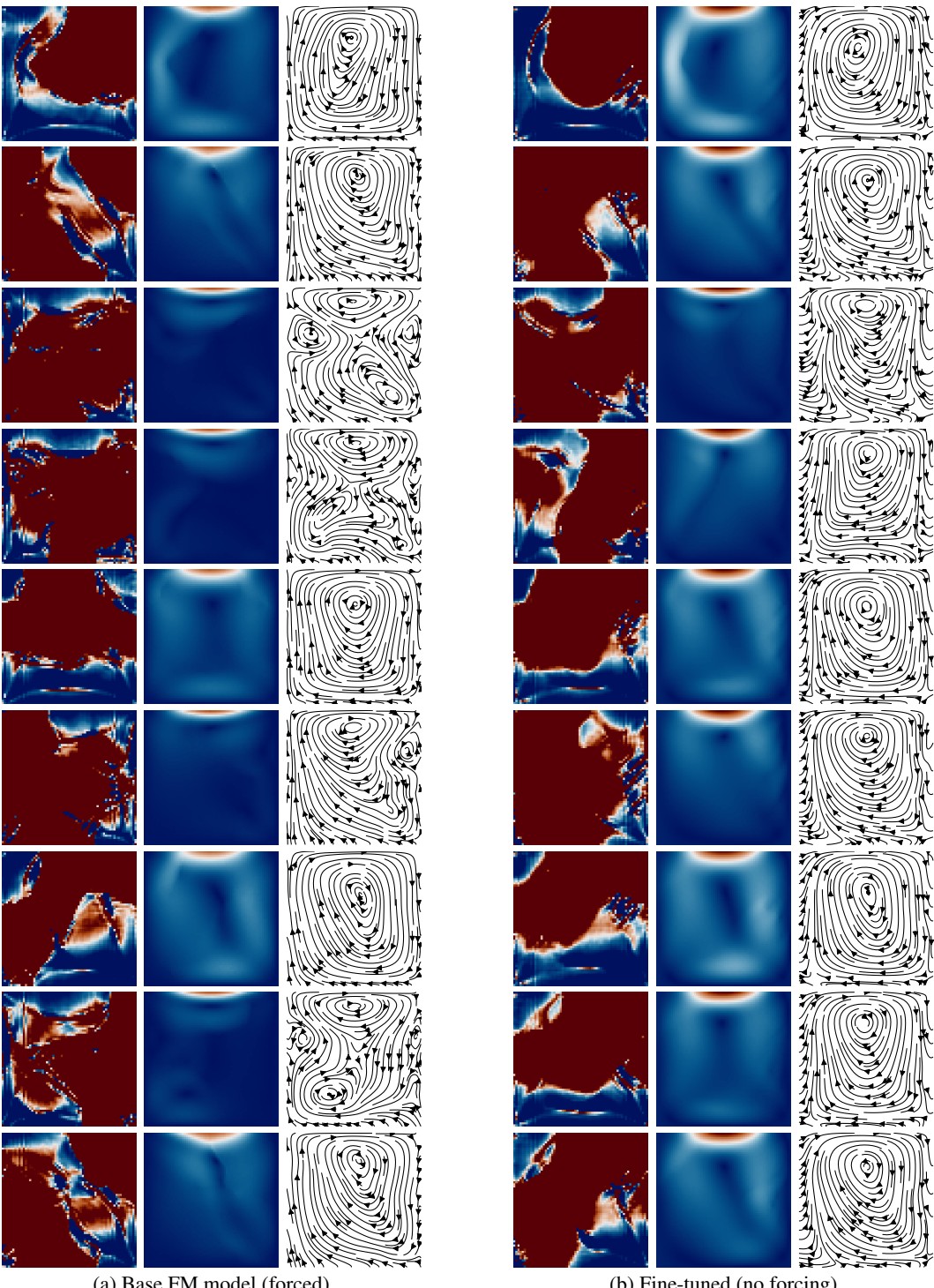

(a) Base FM model (forced)

(b) Fine-tuned (no forcing)

Figure 17: Non-curated Stokes samples. Each row displays: (i) the viscosity field $\nu$, (ii) the velocity magnitude $\|u\|$, and (iii) a streamplot of the velocity $(u_x, u_y)$.

F.3.5 GUIDANCE

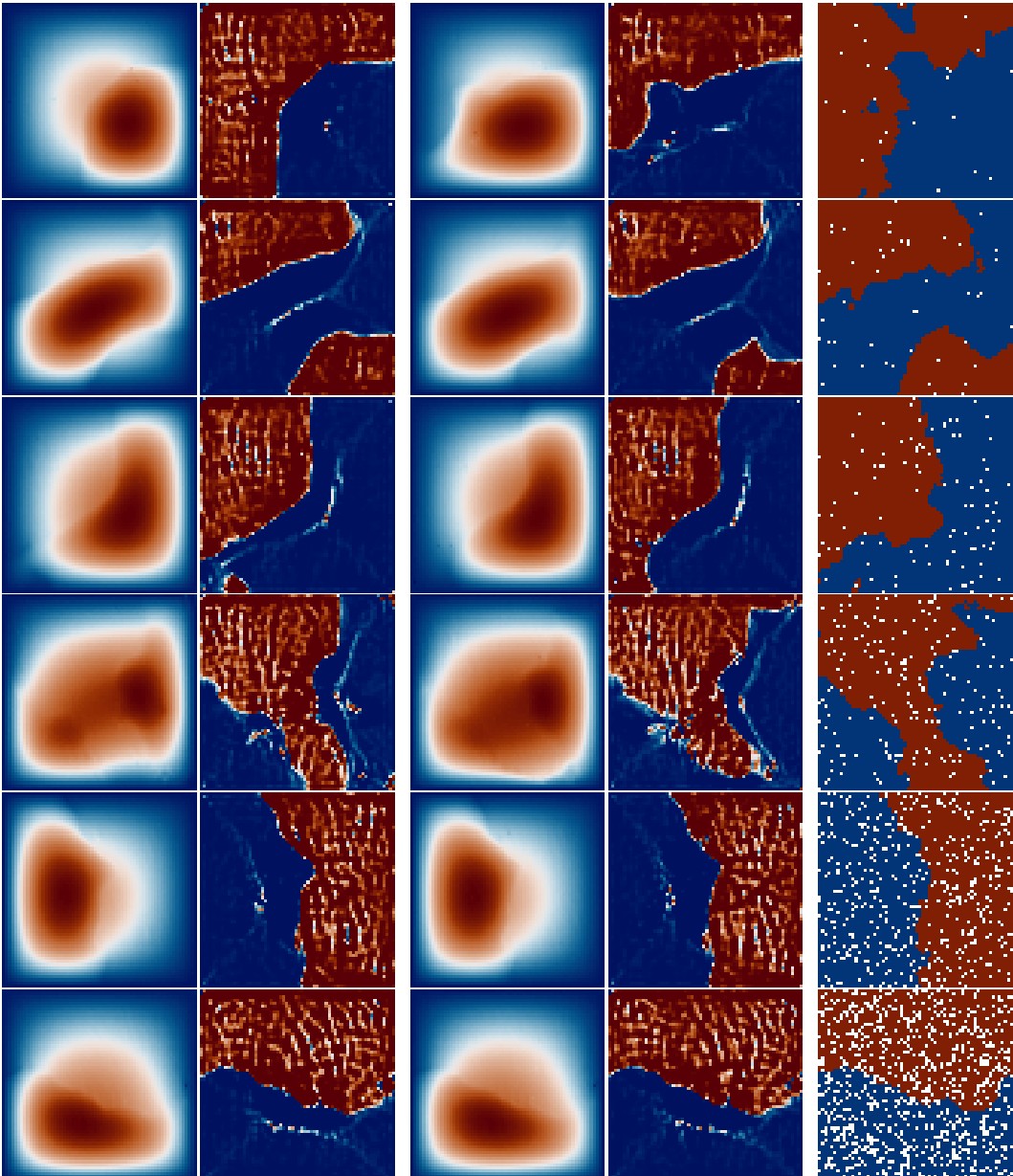

Figure 18: Guided samples with an increasing number of given observations, specifically $[25, 50, 100, 200, 300, 400, 500, 750, 1000]$. For each number of conditioning points, we generate two samples from independent noise and condition on the same sparse samples, indicated as white markers in the right column. As expected, with more points both $x$ and $\alpha$ become more constrained and less diverse.

F.3.6 Natural Images: Recoloring

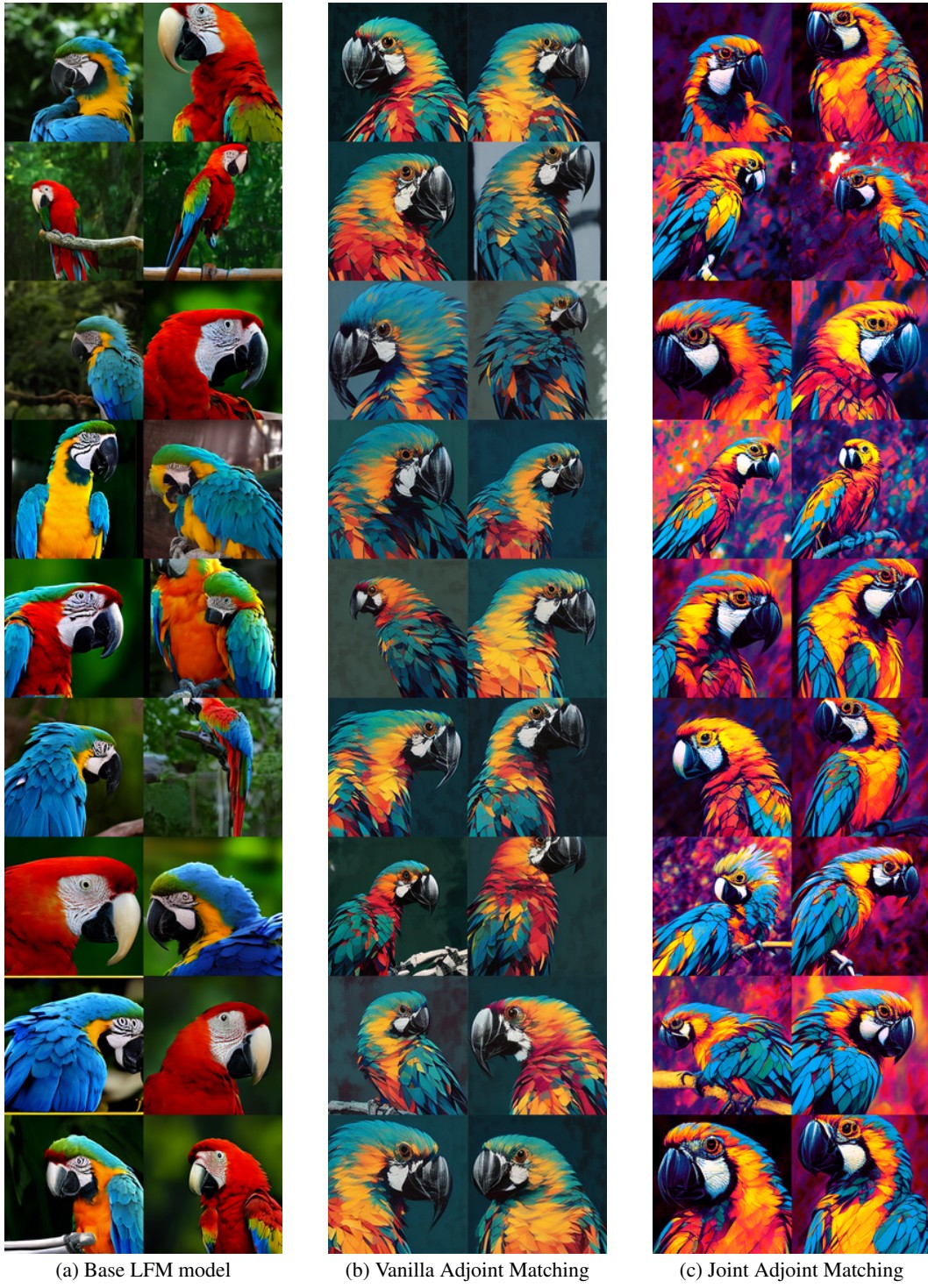

(a) Base LFM model     (b) Vanilla Adjoint Matching     (c) Joint Adjoint Matching

Figure 19: Non-curated independent samples from LFM model conditioned on class "macaw" and using guidance scale 4.0. Models were fine-tuned to maximize PickScore using the prompt "close-up pop art of a macaw parrot".

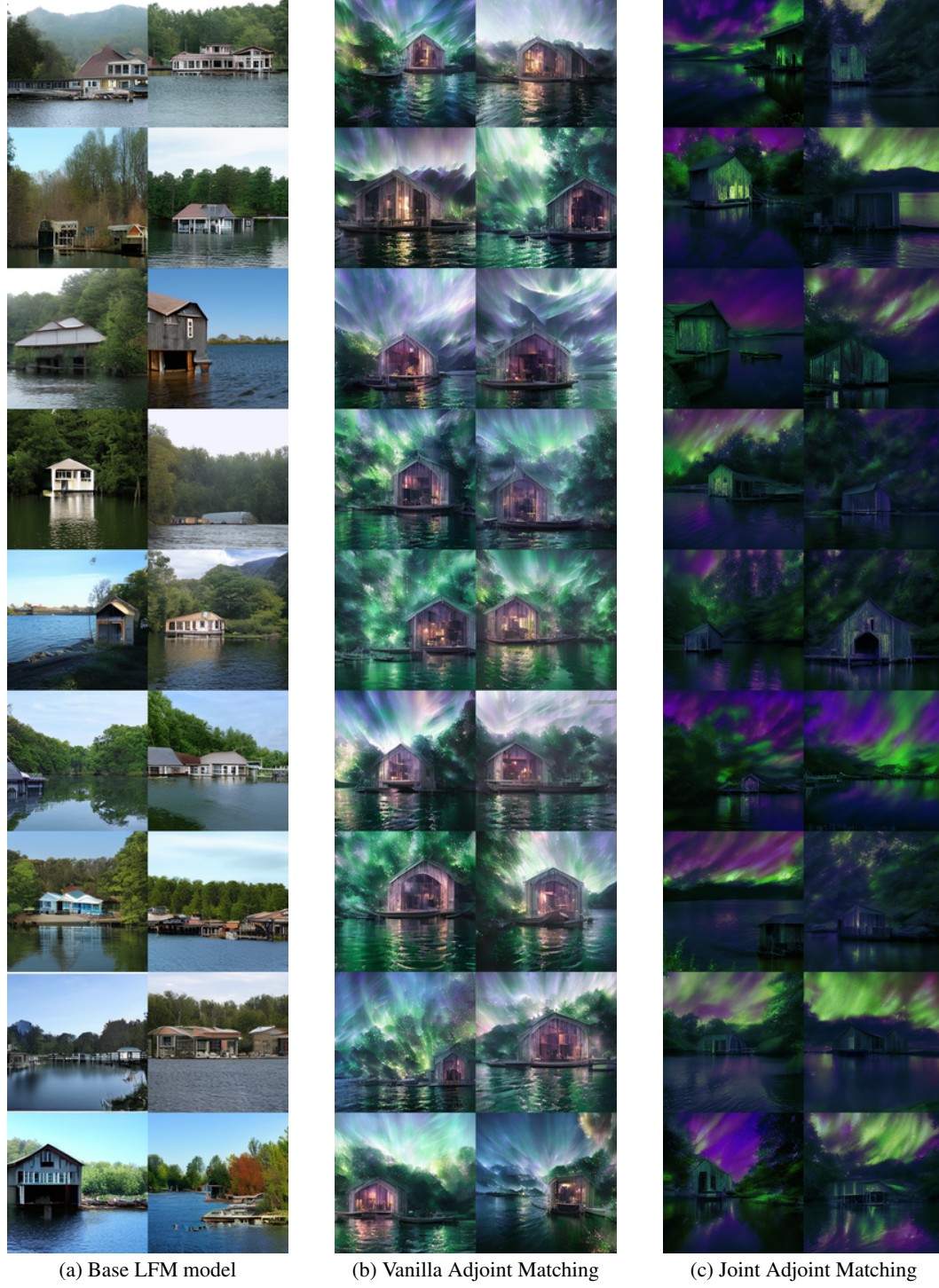

| (a) Base LFM model | (b) Vanilla Adjoint Matching | (c) Joint Adjoint Matching |

Figure 20: Non-curated independent samples from LFM model conditioned on class "boathouse" and using guidance scale 4.0. Models were fine-tuned to maximize PickScore using the prompt "boathouse with green and purple curtains of northern lights." Our joint model is able to generate the colors demanded in the prompt while retaining diversity in the generated boathouses.

