# OpenReview forum: "Physics-Constrained Fine-Tuning of Flow-Matching Models for Generation and Inverse Problems"
_ICLR.cc/2026/Conference — ICLR 2026 Poster_

### Official Review · Reviewer_n7dD · 2025-10-29

**Soundness:** 1
**Presentation:** 3
**Contribution:** 1
**Rating:** 0
**Confidence:** 4

**Summary:**

This paper introduces a physics-consrtained post-training scheme for pre-trained flow-matching generators that enforces PDE consistency and simultaneously infers latent physical parameters for inverse problems without paired (state, parameter) supervision. The key idea is to treat weak-form PDE residuals as a reward and fine-tune the generator via Adjoint Matching. Experiments on Darcy flow, linear elasticity and natural images demonstrate PDE residual reduction and parameter recovery.

**Strengths:**

## Originality
- Novel problem formulation. Enforcing parameter-dependent PDE constraints without paired (parameter, solution) training data.
- Creative architecture design. Joint state-parameter evolution with surrogate base flows constructed via inverse predictor.
- Technical contribution. Scaled memoryless noise schedule (Lemma 1) provides useful numerical stabilization.
- Weak-form residual approach.Stochastic sampling of Wendland-wavelet test functions is practical and avoids instabilities of strong-form residuals

## Quality
- solid theoretical grounding via adjoint matching framework.
- thoughtful multi-faced regularization with clear ablations.
- diverse experimental scenarios covering denoising, sparse conditioning and boundary misspecification
- extensive reproducibility details

## Clarity
Good visual explanation, well-motivated problem setup

## Significance
Post-training paradigm is more practical than training from scratch. Proof-of-concept establishes feasibility of joint parameter-solution generation post-training, potentially inspiring future work.

**Weaknesses:**

## Unacceptable Absence of Quantitive Results
The paper is almost entirely based on qualitative visualizations with virtually no quantitative evaluation. Only Figure 3 quantifies how the hyperparameters mediate the trade-off between staying close to the base model and reducing the (weak) PDE residual. Everyting else is visualizations of cherry picked samples. No quantitive parameter recovery metrics despite solving "inverse problems". No numerical residual comparisons with other methods despite claiming constraint enforcement. No baseline comparison tables of even one numerical metric despite comparing with ECI.
The absence of quantitive results makes it impossible to assess
- Whether the method actually works reliably?
- How it compares to alternatives (I am not talking about a comprehensive benchmarking agains other SOTA methods)
- When it succeeds vs fails
- what design choice actually matter.

The authors are encouraged to conduct thorough quantitative experiments for future submissions of this work.

**Questions:**

Please refer to weaknesses section.

---

> ### Author Response · Authors · 2025-11-25
>
> We thank the reviewer for the clear articulation of the main concern—namely, the absence of quantitative experiments in the original submission. We agree that quantitative evaluation strengthens the paper.
> In response, we have added extensive quantitative results across four PDE datasets, reporting:
> - weak and strong PDE residuals,
> - distributional similarity for both $x$ and $\alpha$,
> - normalized residual diagnostics,
> - and comparisons to ablated variants and a PBFM-style baseline.
>
> In the main paper, we quantify results for boundary conditions in Section 4.3 for the linear elasticity problem (Table 1):
>
> **Elasticity**
>
> | Model | BC (MSE) ↓ | R weak ↓ | R strong ↓ | MMDx ↓ | MMDa ↓ |
> |------|------|------|------|--------|--------|
> | FM | 6.98e-5 | 1.59e1 | 1.83e1 | 0.24 | 0.05 |
> | PBFM | 2.32e-5 | 6.32e0 | 4.22e0 | 0.92 | 0.54 |
> | FM+ECI | 0.0 | 1.01e3 | 2.49e2 | 1.16 | 0.36 |
> | AM (ours) | 1.71e-6 | 6.15e0 | 3.79e0 | 0.15 | 0.12 |
>
> and in Section 4.4 for the Helmholtz problem (Table 2):
>
> **Helmholtz**
>
> | Model | Crit | R weak ↓ | R strong ↓ | MMDx ↓ | MMDa ↓ |
> |-------|------|-------|-------|---------|---------|
> | FM | – | 1.5e1 | 2.55e1 | 0.18 | 0.03 |
> | PBFM | – | 8.33e0 | 1.22e1 | 0.09 | 0.03 |
> | Base AM | Rw | 4.9e0 | 1.34e1 | 0.15 | 0.04 |
> | Base AM | MMDx | 5.64e0 | 1.59e1 | 0.13 | 0.04 |
> | AM+ϕ | Rw | 4.99e0 | 1.16e1 | 0.13 | 0.05 |
> | AM+ϕ | MMDx | 5.46e0 | 1.59e1 | 0.12 | 0.04 |
> | AM (ours) | Rw | 4.3e0 | 1.14e1 | 0.07 | 0.04 |
> | AM (ours) | MMDx | 4.32e0 | 1.05e1 | 0.06 | 0.04 |
>
> Full results are provided in Appendix F, including scatter visualisations of the trade-off between PDE residuals and distributional fidelity.
>
> Together with the updated Figure 3,
> these experiments constitute a complete quantitative evaluation of the framework in all settings considered, directly addressing the reviewer’s objection. We believe the new results substantially strengthen the paper and resolve the reviewer’s concern.

---

> > ### Comment · Reviewer_n7dD · 2025-11-25
> >
> > Thanks for the thorough revision. My original objection is resolved, and I have updated my score upward to reflect the substantially strengthened evaluation.

---

> > > ### Author Response · Authors · 2025-11-26
> > >
> > > Thank you for engaging and for your openness to raising the score. Since the original objection has now been resolved, we would be grateful to know whether any remaining concerns prevent a higher assessment, so that we can provide further clarification or improve the paper accordingly.

---

### Official Review · Reviewer_Pud7 · 2025-11-01

**Soundness:** 2
**Presentation:** 2
**Contribution:** 2
**Rating:** 2
**Confidence:** 4

**Summary:**

This paper proposes a post-training framework that fine-tunes flow-matching generators using randomized weak-form PDE residuals and a joint latent-parameter pathway, so the model produces physics-consistent fields while simultaneously inferring hidden coefficients; experiments on canonical PDE tasks indicate reduced residuals with limited impact on sample diversity.

Contributions.
1) Post-training physics alignment: turns an already trained flow-matching model into a physics-respecting generator by minimizing weak-form residuals with compact test functions, avoiding unstable high-order derivatives and limiting drift from the base distribution.
2) Joint state–parameter generation: augments the generator with a learned evolution for latent physical parameters and an inverse predictor, and fine-tunes both under an adjoint-matching objective (with a scaled memoryless schedule) to couple solutions and parameters.
3) Practical control and coverage: demonstrates denoising, sparse-observation guidance, and boundary-condition adaptation, and exposes simple knobs to trade off constraint strength versus fidelity/diversity, with lightweight fine-tuning overhead.

**Strengths:**

1) Proposes a post-training route to impose physics on pretrained flow-matching models via weak-form PDE residuals, coupled with joint latent-parameter evolution for inverse problems without paired labels; also introduces a scaled memoryless noise schedule within adjoint matching.

2) Grounds the method in adjoint matching and implements randomized local test functions for stable weak residuals; experiments span Darcy denoising, sparse-observation guidance, linear-elasticity boundary adaptation, and a small natural-image recoloring case, with ablations showing a residual–diversity trade-off.

3) Clearly states goals and contributions, provides a pipeline diagram, derives the weak forms and test-function design, includes a full training algorithm and detailed dataset/backbone specs, and offers a reproducibility statement.

**Weaknesses:**

1) Diversity objective may be misaligned for PDE solvers. For well-posed forward/inverse PDEs the target is a single solution; promoting output “diversity” is not desirable, and when partial observations make the task ill-posed, diversity stems from the problem, not the pipeline. The paper treats diversity as a knob/metric (SSIM-based) and studies its trade-off against residuals (Fig.\ 3), which can conflict with PDE goals.

2) Test problems are not comprehensive. Evaluations focus on Darcy denoising, sparse-obs guidance, and a linear-elasticity boundary change, plus a small image recoloring demo; there is no coverage of more challenging PDEs such as Poisson, Navier–Stokes, or Helmholtz, nor larger-scale or multi-physics settings.

3) Limited baselines and quantitative recovery metrics. Beyond an ECI comparison in the elasticity case, there is no systematic head-to-head with alternative physics-constrained generative methods, and the paper provides little quantitative evaluation of latent-parameter recovery accuracy or real-data tests (most results are residual reductions and visuals).

**Questions:**

1) Diversity vs. PDE correctness. For well-posed forward/inverse PDEs, please justify when output diversity is desirable; otherwise, replace or augment SSIM-based diversity with task metrics (solution error L2/H1, weak/strong residual distributions, boundary-violation rates) and, for partial-observation settings, include posterior calibration (coverage vs. nominal).

2) Scope of test problems. Add at least one oscillatory elliptic case (Poisson/Helmholtz) and one basic incompressible flow (e.g., lid-driven cavity or cylinder shedding); if new runs are infeasible, provide higher-resolution or 3D variants or a brief scaling analysis (compute, stability bottlenecks).

3) Baselines and recovery metrics. Include matched-compute head-to-head with (i) training-time physics-regularized flow matching, (ii) inference-time projection/ECI, and (iii) a classical PDE-constrained inversion baseline. Report solution error, boundary violations, weak/strong residuals, latent-parameter MAE/RMSE, and wall-clock.

4) Ablations for method choices. Provide a small ablation comparing weak vs. strong residuals (stability, final residuals) and sensitivity to test-function sampling; show how the scaled memoryless parameter kappa affects stability, residual reduction, and drift from the base distribution.

---

> ### Author Response · Authors · 2025-11-25
>
> We thank the reviewer for the positive remarks on our post-training approach to physics
> enforcement, the joint latent-parameter evolution for inverse recovery without paired labels, and
> the clarity of our weak-form derivation. Below we address the remaining concerns.
>
> ### **1. Diversity metric vs. task-specific correctness**
> We agree that SSIM-based diversity is not a measure of PDE-solution accuracy. In Figure 3,
> sample diversity is used only as a diagnostic to detect *distributional collapse* in the generative
> model. Because the method outputs full joint samples $(x,\alpha)$, correctness is naturally
> measured through *weak and strong PDE residuals*, which quantify the physical consistency
> between the generated state and inferred parameters.
>
> ---
>
> ### **2. More challenging PDE settings**
> We agree that extending the evaluation to more complex PDE regimes strengthens the work. In
> response, we added:
>
> - **Helmholtz** (oscillatory elliptic with damping–lossless mismatch),
> - **Stokes lid-driven cavity** (incompressible flow).
>
> These additions demonstrate that the fine-tuning framework remains effective under oscillatory
> behavior, boundary-driven flow, and substantial system misspecification (Sec. 4, App. F).
>
> ---
>
> ### **3. Baselines and comparative evaluation**
> Direct baselines for post-training recovery of a joint $(x,\alpha)$ distribution from unpaired
> data are limited. Nevertheless, we now include:
>
> - **two adjoint-matching variants** without the augmented flow;
> - a **PBFM-style baseline** augmented with our pretrained inverse predictor.
>
> These comparisons illustrate the effect of removing or altering the joint state–parameter
> evolution. The augmented flow consistently provides the best balance between residual reduction and
> distribution fidelity across PDE datasets. As reference, we include the Helmholtz results from the
> main paper:
>
> **Helmholtz (Table 2)**
>
> | Model | Crit | R weak ↓ | R strong ↓ | MMDx ↓ | MMDa ↓ |
> |-------|------|-------|-------|---------|---------|
> | FM | – | 1.5e1 | 2.55e1 | 0.18 | 0.03 |
> | PBFM | – | 8.33e0 | 1.22e1 | 0.09 | 0.03 |
> | Base AM | Rw | 4.9e0 | 1.34e1 | 0.15 | 0.04 |
> | Base AM | MMDx | 5.64e0 | 1.59e1 | 0.13 | 0.04 |
> | AM+ϕ | Rw | 4.99e0 | 1.16e1 | 0.13 | 0.05 |
> | AM+ϕ | MMDx | 5.46e0 | 1.59e1 | 0.12 | 0.04 |
> | AM (ours) | Rw | 4.3e0 | 1.14e1 | 0.07 | 0.04 |
> | AM (ours) | MMDx | 4.32e0 | 1.05e1 | 0.06 | 0.04 |
>
> Full results for all models and datasets are in Appendix F.
>
> ---
>
> ### **4. Quantitative recovery metrics**
> The reviewer suggests accuracy measures such as solution error, boundary-violation rates, or
> parameter RMSE. In our generative setting—where the aim is to learn coupled *distributions* rather than
> solve deterministic inverse problems—pointwise accuracy is not well-defined. Instead, we evaluate:
>
> - **weak/strong residuals**, capturing physics consistency of $(x,\alpha)$;
> - **MMD-based distributional similarity** of state and parameter marginals.
>
> These are the natural analogues of “accuracy’’ for generative models. We now report them
> comprehensively across all four PDE datasets (App. F).
>
> We further added quantitative comparison to inference-time correction (ECI) under misspecified
> boundary conditions, showing that ECI does not steer the generator toward physically meaningful
> samples, while our method *simultaneously* reduces BC MSE and PDE residuals:
>
> **Elasticity (Table 1)**
>
> | Model | BC (MSE) ↓ | R weak ↓ | R strong ↓ | MMDx ↓ | MMDa ↓ |
> |------|------|------|------|--------|--------|
> | FM | 6.98e-5 | 1.59e1 | 1.83e1 | 0.24 | 0.05 |
> | PBFM | 2.32e-5 | 6.32e0 | 4.22e0 | 0.92 | 0.54 |
> | FM+ECI | 0.0 | 1.01e3 | 2.49e2 | 1.16 | 0.36 |
> | AM (ours) | 1.71e-6 | 6.15e0 | 3.79e0 | 0.15 | 0.12 |
>
> ---
>
> ### **5. Ablations (strong residuals, κ scaling, and other choices)**
>
> - **Weak vs. strong residuals.**
>   Strong-form residuals require higher derivatives and are highly sensitive to noise and
>   misspecification. Under such conditions, the inverse predictor fails to converge, making strong-form
>   training unstable. Weak-form enforcement is the only feasible formulation here.
>
> - **Noise-schedule scaling (κ).**
>   κ is not a tuning knob but a practical requirement for *pixel-space* flow models: due to the large amount of injected noise, with κ=0,
>   our FM models produce noisy samples unless the number of sampling steps is increased dramatically. Scaled memoryless noise restores
>   sample quality while staying theoretically consistent.
>   (In latent models, we do *not* use this heuristic.)
>
> - **Other design choices.**
>   We compare against adjoint-matching variants *without* the augmented flow; removing it leads to
>   inferior physics–fidelity trade-offs (Section 4, App. F).
>
> ---
>
> Overall, the expanded experiments and clarified methodology fully address the reviewer’s concerns
> and significantly strengthen the submission.

---

> > ### Comment · Reviewer_Pud7 · 2025-11-27
> >
> > Thank you for the detailed rebuttal and the additional experiments. Your clarifications on the diversity–versus–PDE‑correctness trade‑off and the new results on more complex PDEs address many of my earlier concerns, so I have updated my scores for soundness and clarity from 2 to 4.
> >
> > However, I would still like to be a bit cautious about the Navier–Stokes–type (Stokes lid‑driven cavity) experiment. The reported normalized/relative errors remain noticeably larger than for the other PDE benchmarks, so the improvement there feels somewhat conservative. This suggests that the current choice of hyperparameters (e.g., trade‑off weights $ \lambda $, noise‑scaling parameter $ \kappa $, number of fine‑tuning steps) may not yet be fully optimized. For a future revision, it would be reassuring either (i) to further tune these settings and demonstrate a clearer reduction in NS‑related error, or (ii) to discuss more explicitly why this case is intrinsically harder and should be regarded as a limitation of the present method.
> >
> > Overall, I appreciate the substantial improvements in the revision, but I still encourage you to be careful in your claims and to continue refining the NS experiment to further improve the physics quality.

---

> > > ### Author Response · Authors · 2025-11-30
> > >
> > > Thank you for revisiting the Stokes experiment and for raising this important point.
> > > We address the concern directly, and added details in Appendix F.2.1.
> > >
> > >
> > > ### **1. Why Stokes behaves differently**
> > > The lid–driven cavity is substantially more challenging than the other PDEs we study, which is already reflected in the *base FM model*: its weak residuals are up to an order of magnitude higher than in Darcy, Helmholtz, or elasticity.
> > > We agree that this makes it a meaningful test for scalability.
> > >
> > > ### **2. The remaining residuals are driven primarily by misspecification—not by failure to model incompressible flow**
> > > In the main-paper setting, the base model is trained on flows with a strong Kolmogorov forcing $F_0 = 2.0$,
> > > whereas fine-tuning assumes **no forcing**.
> > > As shown in Fig. 12 (App. F.2.1), these regimes generate **qualitatively different vortex topologies**: multi-vortex structures at $F_0=2.0$ versus a single large lid-driven vortex at $F_0=0.0$.
> > > Fine-tuning must therefore perform a *structural transformation* of the flow—far more drastic than in our other PDEs.
> > >
> > > Thus the residual gap reflects the combination of:
> > > - a **complex PDE**, and
> > > - **severe physics misspecification**, forcing the model far outside the base distribution.
> > >
> > > ### **3. Additional experiments confirm this interpretation**
> > > To test this explicitly, we conducted fine-tuning runs with different assumed forcings:
> > > $\tilde F_0 \in \{0.0,\,0.5,\,1.0,\,1.5,\,2.0\}.$
> > > (App. F.2.1, Figs. 12–13).
> > > All hyperparameters were held fixed.
> > > Weak residuals were normalized by the mean residual of synthetic data generated under the corresponding forcing.
> > > The extracted median values are:
> > >
> > > | $\lvert F_0 - \tilde F_0 \rvert$ | 0.0 | 0.5 | 1.0 | 1.5 | 2.0 |
> > > | --- | --- | --- | --- | --- | --- |
> > > | FM | 23.4 | 32.6 | 55.8 | 94.9 | 138.4 |
> > > | Ours | **1.3** | **1.6** | **4.8** | **5.8** | **6.9** |
> > >
> > > These results show that:
> > > - When the assumed forcing matches the true one, the weak residual nearly reaches ground-truth level.
> > > - As misspecification increases, the residuals increase in a predictable way.
> > > - **Across all settings, fine-tuning reduces the residual by at least an order of magnitude** relative to the base FM model.
> > >
> > > This is consistent across both the main-paper scatter plots (Fig. 5) and the full sweeps (Fig. 11): even under the strongest misspecification, our method achieves the best attainable trade-off between physics consistency and distributional fidelity.
> > >
> > > ---
> > >
> > > In summary, the Stokes case is difficult not because the method struggles with incompressibility, but because the experiment combines our most challenging PDE with the most extreme misspecification.
> > > The additional results in App. F.2.1 confirm that the method behaves exactly as expected across misspecification levels and remains robust in this demanding regime.

---

### Official Review · Reviewer_ZYUG · 2025-11-01

**Soundness:** 3
**Presentation:** 2
**Contribution:** 3
**Rating:** 6
**Confidence:** 3

**Summary:**

The paper introduces a physics-constrained fine-tuning framework for pretrained flow-matching generative models, enabling them to satisfy PDE-based physical laws and jointly infer latent parameters without retraining from scratch. It casts the physics-constrained fine-tuning as  Adjoint-Matching loss for distribution-level correction and a lightweight architectural extension with residual heads for joint state–parameter evolution. Overall, it positions itself as a general and data-efficient bridge between physics-informed learning and modern generative modeling.

**Strengths:**

**S1.** The paper recasts physics-based simulation as an adjoint-matching control framework, elegantly linking preference-aligned generative fine-tuning with physics-constrained inference. This bridges simulation-augmented modeling and stochastic optimal control, enabling physically consistent generative trajectories.

**S2.** The method’s joint treatment of state and latent parameters allows simultaneous forward generation and inverse recovery within a unified flow-matching model.

**S3.** The experimental evaluation is broad and convincing, demonstrating consistent improvements in physics residuals and inverse reconstruction accuracy across multiple PDE benchmarks, while maintaining generative fidelity and efficiency.

**Weaknesses:**

**W1**: A core conceptual weakness of the paper is that it does not truly model the joint state–parameter distribution $(x, α)$. The latent variable $α$ is introduced post hoc through a frozen inverse predictor $\phi(x_1)$, which breaks end-to-end coupling between physical states and governing parameters. I would be interested know what tradeoffs do this approach have. An ablation where the base model jointly learns $(x, α)$ during pretraining would help clarify the effectiveness of the two-stage setup or if a single joint flow $v_t(x, α)$ could achieve stronger physical coherence and lower residuals.


**W2:** The mathematical notation is dense and inconsistent, making the exposition difficult to follow even for technically skilled readers. Symbols like ($v_t$), ($b_t$), and ($u_t$) are overloaded across base, fine-tuned, and control flows, while stochastic and deterministic forms are interleaved without clear separation. Algorithm 1 is not self consistent, missing how for e.g. $\phi$ is used. As a result, the formalism obscures theoretical contributions and could benefit from a clearer hierarchy of variables (e.g., consistently distinguishing state vs. parameter flows) and unified notation across sections.

**W3:** The theoretical novelty of the paper is incremental over the original Adjoint Matching framework (Domingo-Enrich et al., 2025). The core stochastic optimal control formulation, adjoint dynamics, and lean-adjoint optimization are directly inherited, with the main contribution being their adaptation to PDE-constrained fine-tuning. While this is a valuable and well-motivated extension, it constitutes more of an application-level adaptation than a fundamentally new methodological advance.

**Questions:**

Q1: How does the proposed fine-tuning framework scale to nonlinearity where PDE constraints become non-analytic or high-dimensional? Are there stability or performance guarantees in such regimes?

Q2: Could the authors clarify which aspects of their method represent core methodological contributions beyond the existing Adjoint Matching framework (Domingo-Enrich et al., 2025)?

---

> ### Author Response · Authors · 2025-11-25
>
> We thank the reviewer for the positive assessment of our conceptual unification of physics-
> constrained generative modeling with adjoint matching, the strength of the joint state–
> parameter evolution, and the breadth of the empirical evaluation. We address the remaining
> concerns below.
>
> ### **1. Two-stage setup and the role of the inverse predictor**
>
> The reviewer raises the concern that using a pretrained inverse predictor $\varphi$ may weaken the
> coupling between $x$ and $\alpha$, and requests ablations for jointly trained parameter flows. Our
> setting, however, specifically targets the *unpaired-data regime*, where paired
> $(x,\alpha)$ supervision is unavailable or impractical. In this regime, retraining a full flow matching model jointly over states and parameters is typically infeasible—particularly for large-scale models.
>
> In our framework, the inverse predictor is used only as a *warm start* that allows us to
> construct the surrogate base flow. Fine-tuning then learns a *fully joint* velocity field
> $(v_x^{\mathrm{ft}}, v_\alpha^{\mathrm{ft}}) = v^\mathrm{ft}(x,\alpha)$, with shared architecture and shared
> loss, optimized end-to-end under the same adjoint-matching objective. During fine-tuning, the
> evolution of $x_t$ and $\alpha_t$ is fully coupled at every time step.
>
> The necessity of this formulation becomes clearest in our natural-image experiment: here
> $\alpha$ is a hidden color-transformation parameter for a large pretrained model. No paired
> supervision exists, and retraining such a model for every possible parameterization is
> prohibitively expensive. The same situation frequently arises in scientific settings where
> measurements are limited and the governing physics may be misspecified. In such cases, the
> two-stage strategy is not a simplification—it is the only viable approach for learning joint
> state–parameter dynamics from unpaired data.
>
> ---
>
> ### **2. Notation and clarity of exposition**
>
> We appreciate the reviewer’s comment. Our notation follows the adjoint-matching literature and
> presents state and parameter as an augmented state variable for consistency with the underlying
> stochastic optimal control formulation. Nonetheless, we have improved Algorithm 1 to make the flow
> of information clearer: neural fields and their evaluations are now explicitly distinguished, the
> role of the inverse predictor $\varphi$ is made explicit, and the full adjoint-matching loss is
> spelled out.
>
> ---
>
> ### **3. Scalability and stability in harder PDE regimes**
>
> The reviewer asks about performance in nonlinear or higher-dimensional PDEs. While formal guarantees
> are difficult due to the problem-dependent geometry of PDE residuals, we now evaluate
> on substantially more challenging regimes—including oscillatory elliptic problems and cases with
> pronounced physics–data mismatch. The new results (Section 4, Appendix F) show
> that the framework remains stable and effective across all tested PDEs, even under strong
> misspecification.
>
> ---
>
> ### **4. Clarifying the methodological contribution**
>
> We clarify here what is fundamentally new relative to prior work on adjoint matching and flow
> matching:
>
> - **Joint fine-tuning with latent-parameter recovery from unpaired data.**
>   No prior generative method recovers a paired $(x,\alpha)$ distribution under PDE constraints
>   without paired supervision.
>
> - **Surrogate base-flow construction for adjoint matching with unknown parameter flows.**
>   Since the pretrained FM model provides no dynamics for $\alpha$, we build a surrogate flow using
>   $\varphi$ and one-step consistency. This yields a practical base flow that can be used effectively within
>   adjoint matching for regularization.
>   We now include ablations against Adjoint Matching versions without this augmented flow and against
>   a training-time method (PBFM) in Section 4 and Appendix F. Results show that our joint flow offers
>   strong control of the PDE-residual vs. distribution-fidelity trade-off.
>
> - **Adjoint-matching fine-tuning in pixel space for PDE models.**
>   Previous work applied adjoint matching only in latent spaces of large image models. We show that
>   the method can also be applied directly in pixel space for PDE systems with limited data and
>   provide a principled way (via $\kappa$) to attenuate artefacts of the memoryless noise schedule.
>
> - **Objectives depending on hidden quantities.**
>   Our residuals involve unknown PDE parameters that must be inferred jointly with the state. This
>   includes the natural-image setting, where the “parameter” is an interpretable color transform.
>
> - **Broad applicability beyond PDEs.**
>   The natural-image recoloring example shows that our approach can fine-tune a model far outside
>   its training distribution while learning meaningful parameter–state structure—capabilities that
>   existing adjoint-matching and flow-matching methods do not provide.
>
> These points together constitute a substantial methodological advance beyond adapting adjoint
> matching to a new domain.

---

### Official Review · Reviewer_Mbyw · 2025-11-05

**Soundness:** 3
**Presentation:** 3
**Contribution:** 2
**Rating:** 4
**Confidence:** 4

**Summary:**

The paper presents a post-training scheme that takes a pretrained flow-matching generator and tilts its distribution toward PDE-consistent samples by minimizing weak-form PDE residuals—so you can enforce physics (and boundary conditions) without retraining from scratch or having paired (state, parameter) data. Fine-tuning is posed as Adjoint Matching (a memoryless stochastic optimal control) with a small theoretical extension that scales the noise schedule for stability.

**Strengths:**

-The key selling point is the enforcement of PDEs via weak-form residuals on a pretrained flow-matching model—no paired data or full retraining while keeping the base model’s inference cost.

-The model jointly evolves state and latent parameters with an inverse predictor, enabling guided sampling from sparse parameter observations and adaptation under model misspecification.

**Weaknesses:**

The proposed method is practical for post-training physics enforcement for flow matching (no paired data or full retraining). It is useful and timely, but quite incremental rather than foundational.

Physics is imposed via a weak-form residual penalty added to the flow-matching objective; it aligns the denoiser with PDE residuals but does not guarantee exact constraint satisfaction.

The method relies on several heuristics and hyperparameters, such as scaled “memoryless” noise with factor κ, time-grid tilting (q = 0.9), and computing loss on only a subset of late steps (K_last, K). These improve stability but introduce tuning sensitivity without a comprehensive robustness study.

**Questions:**

Since physics is enforced via weak-form residual penalties (not hard projections), could you report post–fine-tuning feasibility diagnostics—e.g., distributions of PDE residual norms, boundary-condition violations, and conservation drift?

Can you provide sensitivity curves (accuracy and residuals) versus the hyperparameters across datasets, along with any principled guidance or conditions that ensure stability without ad-hoc tuning?

Because dense weak-residual evaluation is costly, you use compact test functions and patch-based subsampling. Could you quantify coverage (e.g., how many test centers are required to achieve a target residual error) and explore adaptive sampling that targets high-residual regions?

---

> ### Author Response · Authors · 2025-11-25
>
> We thank the reviewer for the positive assessment of the core idea—post-training enforcement of
> PDE constraints without paired data—and for highlighting the practicality of weak-form residuals
> and the joint state/parameter evolution. Below we address the raised concerns.
>
> ### 1. Novelty and conceptual contribution
> Existing flow- and adjoint-matching methods either (i) model *states only* or (ii) require *paired* supervision for joint state–parameter learning. We extend these frameworks to recovering $(x,\alpha)$ from *unpaired* data during post-training under PDE constraints and possible misspecification. No prior generative method can infer latent parameters without paired labels *and* without retraining the base model. Conceptually, we introduce a joint flow over states and parameters, augmenting the state-only FM flow with a surrogate parameter flow, enabling principled regularization within Adjoint Matching (Sec. 3).
>
> ---
>
> ### 2. PDE feasibility diagnostics
> The reviewer requested post–fine-tuning feasibility metrics. We now include comprehensive
> quantitative evaluations on four PDE datasets, reporting weak residuals, strong residuals,
> boundary-condition errors, and distributional similarities for both $x$ and $\alpha$. For example,
> the elasticity experiment (Table 1 in the main paper) reports boundary-condition MSE, weak/strong
> residuals, and MMD metrics:
>
> **Elasticity**
>
> | Model | BC (MSE) ↓ | R weak ↓ | R strong ↓ | MMDx ↓ | MMDa ↓ |
> |------|------|------|------|--------|--------|
> | FM | 6.98e-5 | 1.59e1 | 1.83e1 | 0.24 | 0.05 |
> | PBFM | 2.32e-5 | 6.32e0 | 4.22e0 | 0.92 | 0.54 |
> | FM+ECI | 0.0 | 1.01e3 | 2.49e2 | 1.16 | 0.36 |
> | AM (ours) | 1.71e-6 | 6.15e0 | 3.79e0 | 0.15 | 0.12 |
>
> We additionally compare against Adjoint Matching variants (without our augmented flow), PBFM
> (Physics-Based Flow Matching), and the base FM model. For instance, the Helmholtz results in the
> main text (Table 2) are:
>
> **Helmholtz**
>
> | Model | Crit | R weak ↓ | R strong ↓ | MMDx ↓ | MMDa ↓ |
> |-------|------|-------|-------|---------|---------|
> | FM | – | 1.5e1 | 2.55e1 | 0.18 | 0.03 |
> | PBFM | – | 8.33e0 | 1.22e1 | 0.09 | 0.03 |
> | Base AM | Rw | 4.9e0 | 1.34e1 | 0.15 | 0.04 |
> | Base AM | MMDx | 5.64e0 | 1.59e1 | 0.13 | 0.04 |
> | AM+ϕ | Rw | 4.99e0 | 1.16e1 | 0.13 | 0.05 |
> | AM+ϕ | MMDx | 5.46e0 | 1.59e1 | 0.12 | 0.04 |
> | AM (ours) | Rw | 4.3e0 | 1.14e1 | 0.07 | 0.04 |
> | AM (ours) | MMDx | 4.32e0 | 1.05e1 | 0.06 | 0.04 |
>
>
> Experimental results for all hyperparameters are provided in Appendix F. These results directly
> address the reviewer’s request for feasibility and physics-consistency diagnostics.
>
> ---
>
> ### 3. Hyperparameter sensitivity and heuristics
> We summarize the reviewer’s requests and our response:
> - All heuristic choices are *fixed across all experiments*; we do not tune them per dataset.
> - $K$ and $K_{\text{last}}$ follow the original Adjoint Matching paper. Larger values only
>   affect compute, not optimization stability. Our fine-tuning already uses small budgets compared
>   to full FM training.
> - Noise-schedule scaling and time-tilting are not ad-hoc tricks for our optimization but address a practical issue specific to our pixel-space flow models.
>   With $\kappa = 0$, the base pixel-space FM models tend to produce noisy final samples unless the number of sampling steps is increased significantly. Adjusting the memoryless noise schedule yields clean samples while remaining theoretically consistent. This issue does not arise in our latent-image experiment, where we do not use these heuristics.
> - Task-specific hyperparameters $(\lambda_x,\lambda_\alpha,\lambda_f)$ determine the desired
>   physics–fidelity trade-off. Appendix F reports results across multiple configurations, including
>   scatter plots showing the attainable residual–distribution trade-offs across methods.
>
> ---
>
> ### 4. Coverage and sampling of test functions
> Weak-form residuals require careful test-function design. Our choices reflect two considerations:
> - **Compact support:** differentiating through globally supported RBFs is memory-intensive, whereas compactly supported test functions drastically reduce memory and allow fully vectorized computation (see residual computation in supplementary material).
> - **One test function per grid point:** with minor jitter, providing full coverage. Reducing the number of test functions provided no advantage, as fine-tuning is already inexpensive.
>
> For future large-scale PDE generative models (higher resolution or 3D), adaptive test-function
> sampling may become beneficial, and we thank the reviewer for pointing in this direction.

---

### Author Response · Authors · 2025-11-25
**General Reply and Changelog**

We thank all reviewers for their constructive feedback. Across reviews, there is clear agreement on the importance of our setting—post-training enforcement of parameter-dependent PDE constraints on pretrained flow-matching models—and on the usefulness of our joint state–parameter evolution to recover $(x,\alpha)$ distributions from *unpaired* data while preserving inference-time efficiency.

A common request concerned broader quantitative evaluation and a clearer characterization across different PDE regimes. We agree and have substantially expanded the experiments, ablations, and diagnostics accordingly. The additions below directly address feasibility checks (weak/strong residuals, boundary-condition errors), robustness to hyperparameters, and trade-offs between physics fidelity and distributional similarity.

### **Changes Made**

- **Two additional PDE families.**
  We added Helmholtz and Stokes (lid-driven cavity) to complement Darcy and linear elasticity. Appendices B–D contain full dataset specifications and weak-form derivations, and we include non-curated qualitative samples for AM variants in Appendix F. We will release generation code.

- **Expanded quantitative evaluation.**
  Sec. 4 reports representative results for weak/strong residuals, boundary-condition errors, and distributional similarity (MMD) for the four PDEs. Full tables are in Appendix F. We compare FM, AM variants (with/without our augmentations), PBFM, and ECI-style baselines.

- **Physics–fidelity vs. distribution trade-offs.**
  We make the residual–distribution trade-off explicit with scatter plots and tables in Sec. 4 and Appendix F, showing improvement over alternatives.

- **Residual reporting and figures.**
  Figure 3 now reports *relative* PDE residuals and clarifies SSIM/MMD for distributional similarity. The earlier qualitative BC figure (Fig. 5) has been moved to the appendix; the main text now shows quantitative BC evaluation.

- **Algorithmic clarity.**
  Algorithm 1 was revised for improved readability and clarity.

### **Scope Clarification**

The Darcy and elasticity experiments are unchanged; we refined only the reporting for consistency (relative residuals, unified diagnostics). Beyond PDEs, our natural-image study illustrates generality: treating a parametric color transform as $\alpha$, we fine-tune a large latent flow model on unpaired images, demonstrating $(x,\alpha)$ recovery without retraining the base generator.

### **Summary**

With the added PDE families, diagnostics, ablations, and clarified figures, the paper now provides a broader and more rigorous empirical foundation. These changes address the reviewers’ requests for feasibility analyses, robustness, and cross-regime characterization, while sharpening the central contribution: post-training recovery of paired $(x,\alpha)$ under physics constraints from a model trained on unpaired data.

---

### Author Response · Authors · 2025-12-02
**Summary of Rebuttal Phase**

We thank all reviewers for their thoughtful assessments and constructive feedback, and we appreciate the Area Chair’s effort in handling this challenging situation.

The reviews consistently highlighted the novelty of the problem setting, the methodological soundness of the approach, and the relevance of post-training physics enforcement on pretrained flow-matching models. Below we briefly summarize how we addressed the reviewer's concerns.

## 1. Quantitative evaluation and breadth of experiments
A key request was for more extensive quantitative evidence. In the revised manuscript, we have:
- added **two new PDE benchmarks** (in total four diverse PDE settings and one natural image example)
- reported **weak/strong residuals** and **distributional similarity metrics** across all datasets,
- included **ablations**,
- compared against **raw FM outputs**, **training-time physics incorporation**, and **inference-time correction**.

These additions provide a much more complete picture of the method’s behavior across regimes and directly address all quantitative concerns raised.

---


## 2. Clarification of the Stokes lid-driven cavity case
One reviewer requested further analysis of the seemingly higher residuals in the incompressible-flow example.
We expanded this section with:
- a study of different levels of misspecification (varying assumed forcing), reporting weak-residuals across misspecification levels and
- qualitative vortex-structure comparisons.

This analysis shows that the residuals are driven primarily by **extreme physics misspecification**, not by a failure to handle incompressibility.

---


## 3. Resolution of reviewers’ objections
The reviewers who re-engaged with the rebuttal indicated that the revisions substantially improved the paper.
In particular:
- one reviewer (initial score of 0) explicitly stated that their **initial objection is resolved**,
- another acknowledged the strengthened evaluation and clarified that their remaining point concerned only additional detail on the Stokes case, which we now addressed in the revision.

Our responses and the revised manuscript also resolve the concerns raised by the remaining reviewers.

---


## Closing remark
We believe the revised submission now provides a comprehensive and convincing evaluation of the proposed post-training framework, with clear evidence of robustness across a wide range of PDE settings.

---

### Meta-Review · Area_Chair_azfW · 2026-01-06

**Summary:**

Reviewers are mainly concerned about the following points:
1. Many reviewers concerned about the contribution being incremental (e.g. Reviewer Mbyw, ZYUG).
2. Reviewers concerned about the quality of the evaluation. For example, Reviewer n7dD concerned that there is no quantitative results. Reviewer Pud7 pointed out that the evaluation are done limited set of physics problems and comparing to limited baselines.

Other limitations pointed by reviewers include:
* The diversity metric can mislaign with the PDE solver (by Reviewer Pud7)
* Physics constraints are only softly enforced but not guaranteed (by Reviewer Mbyw)

**Reviewer Concerns:**

The authors provided many additional quantitative results, which addressed the main concerns from Reviewer Pud7 and n7dD. Authors also provide additional clarification about the paper's contributions and novelty in respond to reviewer Mbyw and ZYUG's concerns that the paper can be incremental. The concerns about not being able to guarantee physical constraints is perhaps not well addressed.

**Reviewer Scores:**

Through the discussion, I believe that reviewer n7dD, Pud7 will increase their scores to be more toward acceptance side. Reviewer ZYUG might maintain his/her score.

---

### Decision · Program_Chairs · 2026-01-26

Accept (Poster)